# Ana1/CEP295 is an essential player in the centrosome maintenance program regulated by Polo kinase and the PCM

Ana Pimenta-Marques [ID] [1,3,5][✉], Tania Perestrelo[1,5], Patricia Reis-Rodrigues [ID] [1,4], Paulo Duarte [ID] [1], Ana Ferreira-Silva [ID] [2], Mariana Lince-Faria [ID] [1] & Mónica Bettencourt-Dias [ID] [1][✉]

## Abstract

Centrioles are part of centrosomes and cilia, which are microtubule organising centres (MTOC) with diverse functions. Despite their stability, centrioles can disappear during differentiation, such as in oocytes, but little is known about the regulation of their structural integrity. Our previous research revealed that the pericentriolar material (PCM) that surrounds centrioles and its recruiter, Polo kinase, are downregulated in oogenesis and sufficient for maintaining both centrosome structural integrity and MTOC activity. We now show that the expression of specific components of the centriole cartwheel and wall, including ANA1/CEP295, is essential for maintaining centrosome integrity. We find that Polo kinase requires ANA1 to promote centriole stability in cultured cells and eggs. In addition, ANA1 expression prevents the loss of centrioles observed upon PCM-downregulation. However, the centrioles maintained by overexpressing and tethering ANA1 are inactive, unlike the MTOCs observed upon tethering Polo kinase. These findings demonstrate that several centriole components are needed to maintain centrosome structure. Our study also highlights that centrioles are more dynamic than previously believed, with their structural stability relying on the continuous expression of multiple components.

**Keywords** Centrosome Integrity Maintenance; Oogenesis; Cytoskeleton; Homeostasis; Centriole
**Subject Categories** Cell Adhesion, Polarity & Cytoskeleton; Development

## Introduction

An important feature for cell homeostasis is how its structures are maintained. This is the case of the centrosome, the most studied microtubule-organizing center (MTOC) in eukaryotic cells. This organelle is composed of two centrioles, surrounded by a multi-protein matrix called the pericentriolar material (PCM) (Brito et al, 2012; Conduit et al, 2015). The PCM is indispensable for centriole biogenesis and for nucleating and anchoring MTs at the centrosome (Nabais et al, 2021; Pimenta-Marques et al, 2016). Centrioles are cylindrical structures, made of microtubule (MT) doublets or triplets arranged in a 9-fold radial symmetry, together with several conserved proteins, that build the centriole wall (Fig. 1B). The most proximal part of centrioles features a cartwheel structure consisting of a central hub and nine radially arranged spokes along their length, which serves as a template for the assembly of new centrioles (Callaini et al, 1997; Guichard et al, 2018). At the distal end of the centriole, lies the centriole cap, a complex of conserved proteins crucial for regulating centriole biogenesis and growth (Fu and Glover, 2012; Kleylein-Sohn et al, 2007).

Historically, centrosomes have been regarded as exceptionally stable structures. They are resistant to drug- and cold-induced depolymerization (Kochanski and Borisy, 1990) and to forces and MT destabilisation at the entrance of mitosis (Belmont et al, 1990). Furthermore, experiments in *C. elegans* demonstrated that centrioles contributed by the sperm, persist for several embryonic cell cycles (Balestra et al, 2015). This finding strongly suggests that centrioles are stably inherited through numerous cell divisions.

Despite their inherent stability, centrosomes are lost from the oocytes of most metazoan species (Manandhar et al, 2005; Werner et al, 2017) and are known to be inactivated (i.e. loss of their MTOC capacity) in some cell types that undergo differentiation, such as neuronal, muscle and epithelial cells (Muroyama and Lechler, 2017; Sanchez and Feldman, 2017). Several studies in different proliferating cell types have uncovered the pathways regulating how centrosomes mature and become active MTOCs. However, far less is known on the pathways regulating the maintenance of mature centrosomes, as well as their inactivation or elimination, which are critical for centrosome number control in different cell types.

We have previously identified what we named a *centrosome maintenance mechanism*, which operates in the *Drosophila* germline and somatic cells (Pimenta-Marques et al, 2016). This process

[1]Instituto Gulbenkian de Ciência, Rua da Quinta Grande, 6, 2780-156 Oeiras, Portugal. [2]iNOVA4Health | NOVA Medical School | Faculdade de Ciências Médicas, Universidade Nova de Lisboa, Lisbon, Portugal. [3]Present address: iNOVA4Health | NOVA Medical School | Faculdade de Ciências Médicas, Universidade Nova de Lisboa, Lisbon, Portugal. [4]Present address: Institute of Science and Technology Austria, 3400 Klosterneuburg, Austria. [5]These authors contributed equally: Ana Pimenta-Marques, Tania Perestrelo.
[✉]E-mail: ana.pmarques@nms.unl.pt; mdias@igc.gulbenkian.pt

   

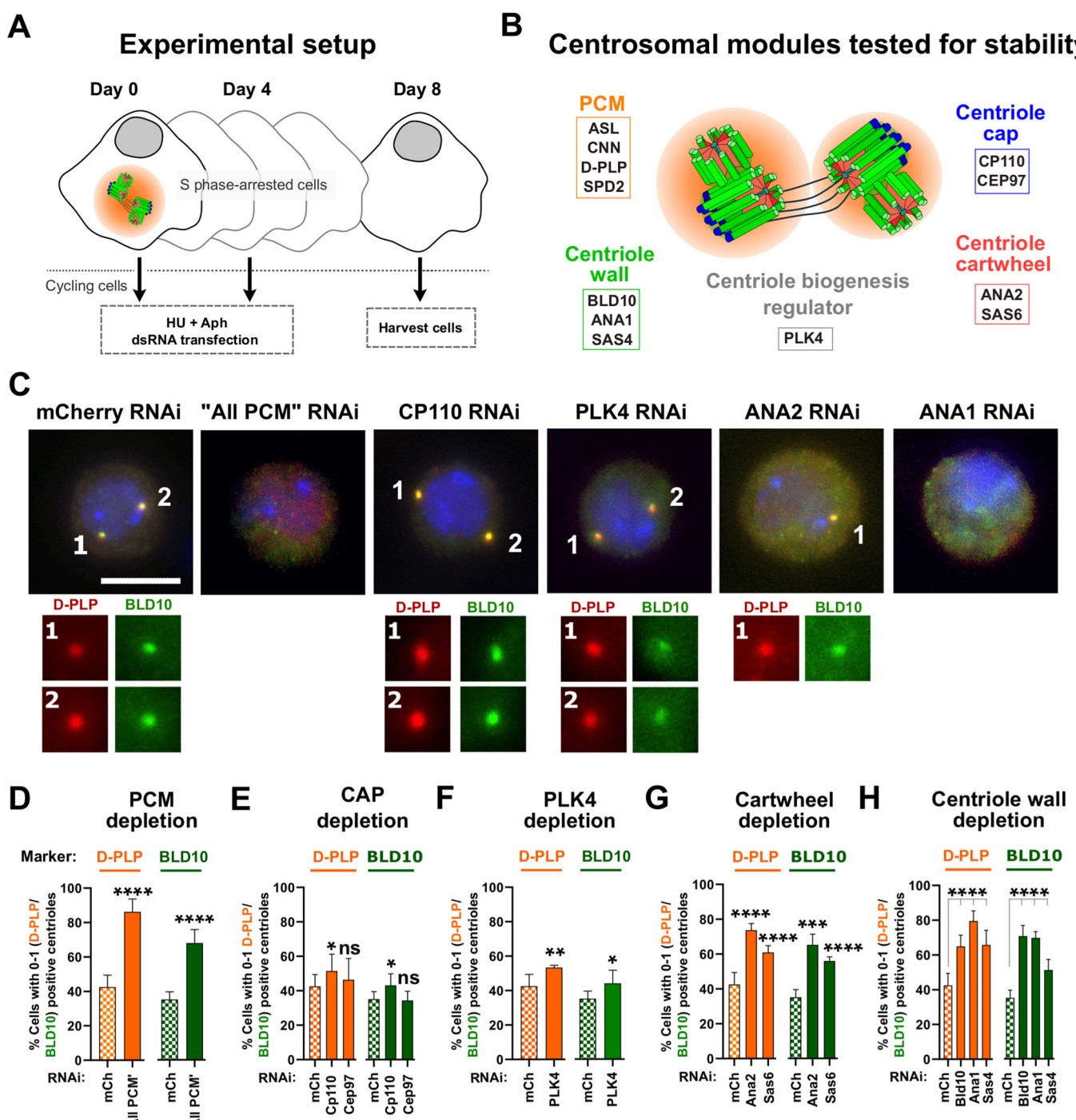

**A** Experimental setup

**B** Centrosomal modules tested for stability

**C** mCherry RNAi · "All PCM" RNAi · CP110 RNAi · PLK4 RNAi · ANA2 RNAi · ANA1 RNAi

**D** PCM depletion · **E** CAP depletion · **F** PLK4 depletion · **G** Cartwheel depletion · **H** Centriole wall depletion

is led by Polo (PLK1 in Humans), a conserved kinase that is critical for PCM recruitment (Dobbelaere et al, 2008; Haren et al, 2009; Lane and Nigg, 1996) and PCM maintenance (Cabral et al, 2019; Pimenta-Marques et al, 2016; Singh et al, 2014) in proliferating cells. This mechanism is shut down in the female germline, with loss of Polo and consequently of the PCM, from centrosomes. This is followed by centrosome functional inactivation and centriole loss. Induced depletion of Polo and the PCM (depletion of four major PCM components) in S-phase arrested *Drosophila* cultured

cells, also leads to centriole loss (Pimenta-Marques et al, 2016). These observations demonstrate that Polo and the PCM are critical players in the maintenance of centriole integrity, and that the mechanism by which they promote centriole stability may be conserved in different cell types.

How Polo and the PCM promote centriole structural integrity is not known. There is some evidence supporting the dynamicity of PLK1, the PCM and centriole components (Bahmanyar et al, 2010; Conduit et al, 2015; Keller et al, 2014; Mahjoub et al, 2010; Novak

◄  **Figure 1.  The PCM, the cartwheel and the centriole wall are critical to maintain centriole structural integrity.**

(**A**) Schematic representation of the "centriole stability assay": cells were transfected at day 0 with double-stranded RNA (dsRNA), and simultaneously arrested in S-phase with HU (hydroxyurea) and Aph (aphidicolin). On day 4, cells were subjected to a second round of dsRNA transfection and treatment with HU (hydroxyurea) and Aph (aphidicolin). On day 8, cells were harvested and assayed for centriole numbers by immunofluorescence. (**B**) Schematic representation of molecules depleted in the screen. (**C**) Representative images of centrosomes stained with D-PLP (red) and BLD10 (green) in control cells (mCherry RNAi) and cells with dsRNA transfection for "All PCM" (ASL, CNN, D-PLP and SPD2; positive control as previously described (Pimenta-Marques et al, 2016)), PLK4, CP110, ANA2, or ANA1. DNA (blue). Enlargements of centrosomes present in each cell are shown. Scale bar, 5 μm. (**D-H**) Centriolar numbers were assessed considering the staining for the PCM marker D-PLP (orange bars) or centriole wall marker BLD10 (green bars). Quantification of the percentage of cells with abnormally low centriole numbers (0-1 centrioles) upon depletion of (**D**) "All PCM", (**E**) CP110 and CEP97, (**F**) PLK4 kinase, (**G**) SAS6 and ANA2, and (**H**) BLD10, ANA1 and SAS4. Data Information: Data are the mean ± SEM of three independent experiments (for each biological replicate in each experimental condition, $n \geq 100$ cells). Statistical significance was determined by performing a bimodal regression test. The impact of the different RNAi treatments on the number of cells with 0–1 centrioles was estimated based on the number of cells that present a reduced number of centrioles with the different markers. Estimates indicate the log odds ratio that the indicated treatment increases the number of cells with 0–1 centrioles. $*p < 0.05$; $**p < 0.01$; $***p < 0.001$; $****p < 0.0001$; ns, not significant. See also Fig. EV1 for other markers and Materials and Methods for details on statistical methods. Source data are available online for this figure.

et al, 2014; Woodruff et al, 2014), suggesting that replenishment of those molecules may be important for centriole function and integrity. Here we conduct an RNAi screen to test which centrosome components have a role in centriole structural integrity and show that the cartwheel and the centriolar wall are critical. We further identify an essential role for the centriolar wall protein ANA1 in centriole integrity, with its removal leading to the disappearance of fully matured centrosomes, and with its overexpression rescuing centriole loss both in cultured cells and in oocytes.

## Results

### The PCM, the centriole wall and the cartwheel are critical for centriole integrity

We conducted an RNAi screen to identify which centriole components are important for its integrity. We used a "centriole stability assay" that we previously developed and validated (Pimenta-Marques et al, 2016) (Fig. 1A). In this assay, *Drosophila* tissue culture cells (DMEL) are arrested in S-phase, halting both the cell and the centriole biogenesis cycles, thus keeping centriole numbers constant (Dzhindzhev et al, 2010a). The dsRNA is given at the same time as cells are arrested to ensure the assay addresses centriole maintenance and not biogenesis. Given that the dsRNA takes at least 24 h to have an effect and the cell cycle of DMEL cells is ~24 h, the targeted proteins should not be depleted before the drugs arrest the cells (Pimenta-Marques et al, 2016).

We targeted different centriole substructures such as the cartwheel (SAS6 (Cottee et al, 2015; Kitagawa et al, 2011) and ANA2/STIL (Cottee et al, 2015; Dzhindzhev et al, 2014)), the wall (BLD10/CEP135 (Kleylein-Sohn et al, 2007; Roque et al, 2012), SAS4/CPAP (Gopalakrishnan et al, 2011; Kleylein-Sohn et al, 2007), and ANA1/CEP295 (Chang et al, 2016; Fu et al, 2016)) and the cap (CP110 and CEP97 (Fu and Glover, 2012; Kleylein-Sohn et al, 2007)). We also tested PLK4 kinase as it is a major regulator of centriole biogenesis, known to regulate several centriolar components (Bettencourt-Dias et al, 2004, 2005; Habedanck et al, 2005; Zitouni et al, 2014). We targeted the PCM as a positive control, given its known role in centrosome maintenance (Pimenta-Marques et al, 2016). Four major PCM proteins were simultaneously depleted (ASL, CNN, D-PLP and SPD-2 ("ALL PCM") (Dzhindzhev et al, 2010a; Fu et al, 2016; Fu and Glover, 2012;

Martinez-Campos et al, 2004; Mennella et al, 2012, 2014), as individual depletion was previously shown not to be sufficient to induce centriole loss (Pimenta-Marques et al, 2016). To infer which parts of the centrosome were disturbed upon RNAi, we used markers of the PCM (D-PLP), the centriolar wall (BLD10, ANA1, and SAS4), and the centriole distal cap (CP110) (Fig. 1B–H, markers D-PLP and BLD10; Fig. EV1, markers SAS4, ANA1 and CP110). As previously shown by us (Pimenta-Marques et al, 2016), "All PCM" depletion induced a strong reduction of centriole number, confirming the importance of the PCM for the maintenance of centrosome integrity (Figs. 1C,D and EV1B). Depletion of cartwheel or centriolar wall proteins also led to a strong decrease in centriole number, as seen with several markers (Figs. 1C,G,H and EV1E,F).

Cells depleted of cap proteins, despite showing a reduction in SAS4 foci numbers (Fig. EV1C), did not show a strong reduction in the other markers (Figs. 1C,E and EV1C). It is possible that the loss of SAS4 upon cap protein depletion reflects the specific interaction of SAS4 with CP110 (Galletta et al, 2016). Similarly to the cap proteins, PLK4 depletion did not lead to substantial centriole loss (Figs. 1C,F and EV1D). This result is in line with other experiments where cells were subjected to PLK4 inhibition for a very long time and loss of centriole integrity was not observed (Nabais et al, 2021; Wong et al, 2015). The observed loss of SAS4 foci upon Plk4 depletion (Fig. EV1D), suggests that PLK4 might be involved in SAS4 maintenance, additionally to its known role in human SAS4 recruitment to centrioles (Moyer and Holland, 2019). Together our results suggest that turnover of the PCM, the wall, and the cartwheel contributes to centriole structural integrity.

From all the candidates tested, although we cannot rule out the possibility that differences in RNAi efficiency lead to the observed differences in phenotype severity, depletion of ANA1 led to the strongest effect on different centrosomal markers (Figs. 1C,H and EV1F), similar to "All PCM" depletion (Figs. 1C,D and EV1B).

Interestingly, ANA1/CEP295 has been shown to function as a centriolar bridge, connecting the centriole wall with the PCM (Fu et al, 2016; Tian et al, 2021; Tsuchiya et al, 2016). The direct interaction in *Drosophila* between ANA1 and the PCM protein ASL (Fu et al, 2016), as well as in Humans between CEP295 and both Cep152 (Fu et al, 2016) and Cep192 (Tsuchiya et al, 2016) is required for centriole-to-centrosome conversion (Fu et al, 2016; Izquierdo et al, 2014). This process is important to stabilise centrioles after cartwheel loss at the exit of mitosis in human cells (Izquierdo et al, 2014). In addition, phosphorylation of ANA1 at

S-S/T motifs was recently shown to prompt the recruitment of Polo to mother centrioles, contributing to PCM maturation and centriole elongation (Alvarez-Rodrigo et al, 2021). Also, in *Drosophila*, ANA1 is one of the last proteins to be lost from centrioles before they are eliminated in oogenesis (Pimenta-Marques et al, 2016), and from the ommatidia of the eye (Riparbelli et al, 2018). Similarly, it is one the last proteins to be lost upon centrosome reduction in the sperm basal body (Blachon et al, 2009; Khire et al, 2016). Altogether this evidence suggests that ANA1 may be an important player in the PCM/Polo centrosome maintenance mechanism. Therefore, we further investigated the role of ANA1 in centrosome maintenance, in particular its role in centriole structural integrity in fully matured centrosomes, exploring its function in vivo, as well as its mechanism of action in relation to Polo and the PCM.

### The centriolar wall protein ANA1 is required for centrosome maintenance in vivo

The female germline is a great system to study the maintenance of centrosomes as these structures are progressively eliminated throughout oogenesis. In early stages, oocytes are specified from cysts of 16 interconnected cells. The centrosomes from 15 cells migrate into the oocyte forming a large MTOC, which is active up to mid stages of oogenesis (stages 6–7). At these stages, centrosomes start first losing Polo and the PCM, followed by their progressive elimination before meiotic metaphase I (Pimenta-Marques et al, 2016). Therefore, we investigated if ANA1 is important for centrosome maintenance in vivo in this system.

We depleted a large portion of existing ANA1 using the deGradFP system (Fig. 2). This system targets GFP-tagged proteins for degradation (Caussinus et al, 2012). We specifically induced the expression of deGradFP in oocytes from stages 3/4 onward, when most centrosomes should have already duplicated and migrated from the nurse cells to the oocyte (Mahowald and Strassheim, 1970). Here we expressed ANA1-GFP under its endogenous promoter in the genetic background of *ANA1* mutant (Blachon et al, 2008). Centrioles were analysed at stage 10 of oogenesis, when they are still present in control conditions (Fig. 2A) (Pimenta-Marques et al, 2016). Expression of deGradFP led to approximately a 80% decrease in the total levels of ANA1-GFP on centrioles (Degron condition), when compared to ANA1-GFP flies without deGradFP expression (control, Fig. 2B), showing that a significant pool of ANA1 is being depleted. Given that centrioles are densely packed at late stages of oogenesis, and thus very difficult to count individually, we measured the total intensity of different centrosome markers as a proxy for centrosome content as previously validated and performed (Pimenta-Marques et al, 2016). We used the robust markers D-PLP and γ-tubulin for the PCM, and CP110 for the centriole. ANA1 depletion led to a strong reduction in all tested markers, suggesting centrioles are being eliminated in those conditions (Fig. 2B). Collectively, our observations show that ANA1 is important for maintaining the integrity of centrioles.

### ANA1 is a player in the Polo mediated centrosome maintenance program

We have previously found that Polo kinase and the PCM are critical for the maintenance of centrosome integrity. Tethering Polo to centrosomes using the PACT domain of the PCM protein D-PLP (Gillingham and Munro, 2000; Martinez-Campos et al, 2004)

rescues the loss of centrosomes, both in culture cells depleted of PCM, and in oogenesis where the PCM is naturally lost (Pimenta-Marques et al, 2016). We asked whether ANA1 would play a role in the Polo-mediated centrosome maintenance program.

We first asked whether Polo requires ANA1 to prevent centriole loss. We used the "centriole stability assay" in *Drosophila* culture cells, where we depleted the PCM to trigger centriole loss as observed in Fig. 1, and expressed GFP-Polo-PACT or GFP-PACT (control ; Fig. 3A–C). As expected, cells depleted of "All PCM" and expressing GFP-PACT have abnormally low numbers of centrioles (0-1 centrioles) (Figs. 3C and EV2A using another centriole marker). In this context, as previously shown, expression of GFP-Polo-PACT partially rescues centriole loss, when compared to cells expressing GFP-PACT only (Figs. 3C and EV2A) (Pimenta-Marques et al, 2016). It is unlikely that the observed partial rescue results from retaining the PCM as there is no rescue of the PCM markers upon Polo-PACT expression (Fig. EV2B–H). Indeed, most centrioles have no PCM marker (SPD2 and D-PLP) upon both GFP-Polo-PACT and GFP-PACT expression, as compared with mcherry RNAi control (Fig. EV3). However, the percentage of ANA1 positive centrioles is increased upon Polo-PACT expression (Fig. EV3B), suggesting ANA1 is a critical component in rescuing centriole structural integrity upon Polo-PACT expression. Indeed, upon "All PCM" and ANA1 depletion, GFP-Polo-PACT expression was no longer capable of rescuing centriole numbers (Fig. 3C). These results suggest that the partial rescue provided by GFP-Polo-PACT upon PCM depletion is dependent on ANA1 expression. Consistent with this, GFP-Polo-PACT cannot also rescue centriole numbers in cells depleted of ANA1 (Fig. EV2I).

In oogenesis, tethering Polo to the oocyte centrioles leads to their maintenance, as well as recruitment of γ-tubulin (Pimenta-Marques et al, 2016). Under these conditions, centrioles are maintained beyond meiotic metaphase I, a stage where they are naturally absent. These centrioles contain ANA1 and are active MTOCs (Pimenta-Marques et al, 2016). We thus asked if ANA1 is required for Polo-promoted centriole maintenance in oocytes. In stages 12 of oogenesis, when the majority of centrioles are normally lost in control conditions (GFP-PACT expressing oocytes), tethering Polo to the centrioles maintains centrosomes (Pimenta-Marques et al, 2016). Here we expressed GFP-Polo-PACT while ANA1 synthesis was prevented by RNAi. We observed that the levels of both γ-tubulin and BLD10 were significantly reduced, and there was a clear reduction of the percentage of stage 12 egg chambers showing the presence of centrioles (Fig. 4A–C). It's worth noting that, at stages 12, 60% of the eggs expressing GFP-Polo-PACT and subjected to RNAi for ANA1, retained their centrioles, whereas typically, at this stage, centrioles are only identified in 30% of the total number of eggs (Fig. 4C). This partial rescue likely results from a not fully efficient RNAi-mediated depletion of ANA1 in oogenesis (Fig. EV4A,B). This is consistent with previous reports showing that ANA1 at centrosomes in fly embryos has a dynamic and a stable pool (Saurya et al, 2016). Nonetheless, our results strongly suggest that, as observed in cultured cells, Polo-induced centrosome maintenance in vivo depends on the presence of ANA1.

### ANA1 is sufficient for maintaining centriole structural integrity

One plausible scenario to explain our findings is that ANA1 serves as a critical centriole structural element that is slowly exchanged

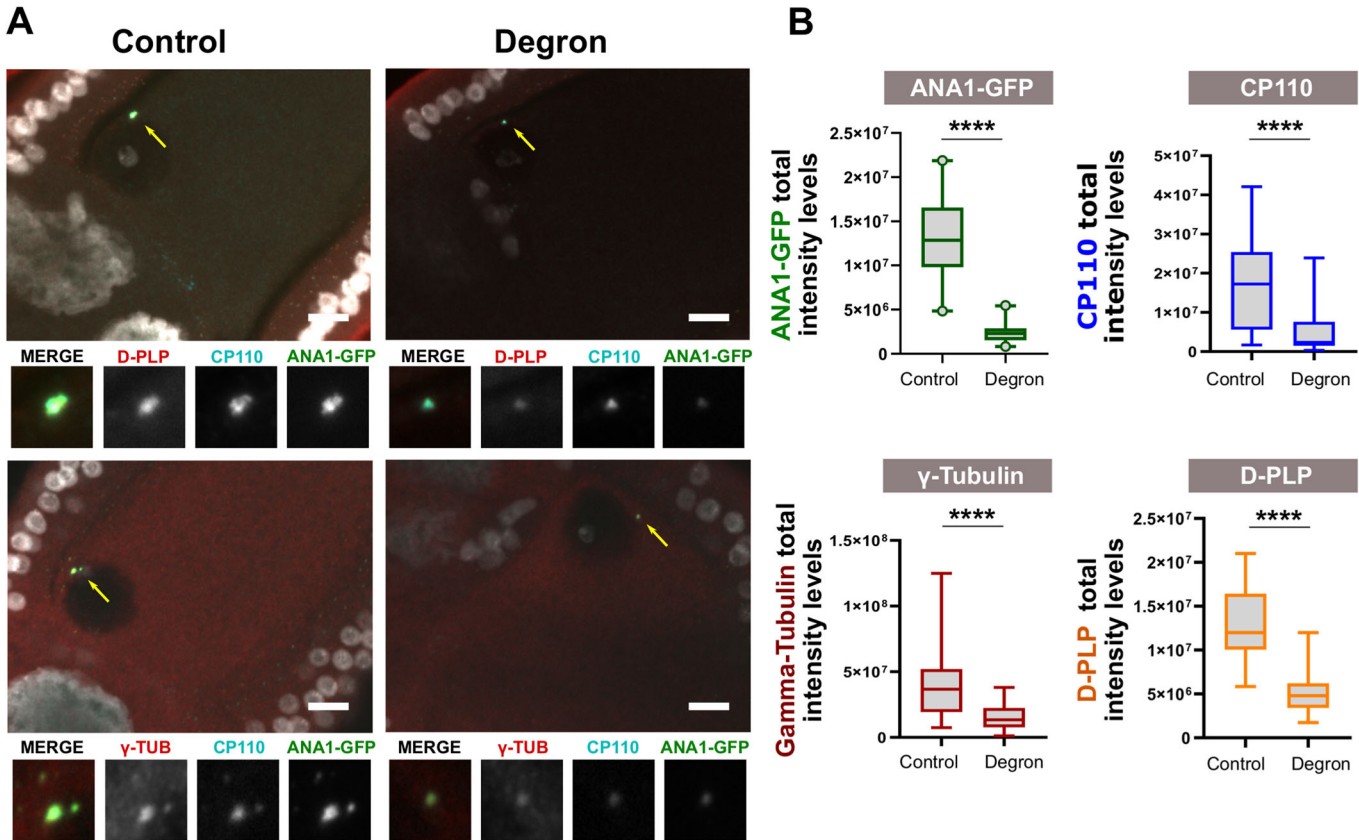

**Figure 2.  The centriolar wall protein ANA1 is required for centrosome maintenance in vivo.**

(A,B) Expression of the deGradFP tissue-specific system in oogenesis (using UAS/Gal4) leads to tissue-specific degradation of ANA1-GFP (expressed under the control of its endogenous promoter). deGradFP (Degron Condition) was induced in the germ line after stages 3/4 onward, a stage where centrosomes have duplicated and migrated to the oocyte. ANA1-GFP flies without the deGradFP system were used as controls. (A) Stage 10 oocytes were immunostained for the centriole marker CP110 (cyan), and PCM markers, D-PLP (red) or alternatively γ-TUB (red). Arrows point to centrosomes in the different egg chambers, which are enlarged in the figure insets. (B) The degron-induced reduction of ANA1-GFP levels in oogenesis leads to a strong reduction in the levels of PCM (γ-TUB and D-PLP) and CP110. Quantification of total intensity levels of the different markers. Data information: Box-and-whisker plot (2.5th and 97.5th percentiles) of the total integrated intensities of the different markers analysed. For ANA1-GFP, for control, $n = 41$, box minimum = 4,834,508, box maxima = 21,879,966, box median = 12,865,901, box 25% percentile = 9,801,948, box 75% percentile = 16,583,028; for degron, $n = 41$, box minimum = 821,394, box maxima = 5,478,289, box median = 2,305,398, box 25% percentile = 1,533,454, box 75% percentile = 2,893,825. For CP110, for control $n = 30$ box minimum = 1,677,893, box maxima = 42,124,559, box median = 17,226,784, box 25% percentile = 17,226,784, box 75% percentile = 25,378,539; for degron, $n = 31$, box minimum = 318,155, box maxima = 23,915,065, box median = 2,341,007, box 25% percentile = 1,462,800, box 75% percentile = 7,620,560. For D-PLP, for control, $n = 32$, box minimum = 5842624, box maxima = 21,013,732, box median = 11,984,641, box 25% percentile = 10,052,899, box 75% percentile = 16,432,314; for degron, $n = 28$, box minimum = 1,740,947, box maxima = 11,981,605, box median = 4,797,745, box 25% percentile = 3,404,878, box 75% percentile = 6,221,396. For γ-TUB, for control, $n = 30$, box minimum = 7,432,147, box maxima = 124,943,411, box median = 36,687,129, box 25% percentile = 19,354,882, box 75% percentile = 52118277; for degron, $n = 31$, box minimum = 1,203,453, box maxima = 38,005,878, box median = 13,440,215, box 25% percentile = 7,740,412, box 75% percentile = 22,380,976. For ANA1-GFP four independent biological replicates were performed. For CP110, D-PLP and γ-TUB, 3 independent biological replicates were performed. Statistical significance was determined by performing a bimodal regression test. For all the statistical tests used in this figure: ****$p < 0.0001$; ns, not statistically significant. See Statistical methods for details on statistics. Source data are available online for this figure.

and thus needs to be replenished at the centriole via continuous synthesis and recruitment. We thus tested if overexpressing and tethering ANA1 to the oocyte centrioles is sufficient to maintain centrioles until later stages (stages 12), when most egg chambers have already lost their centrioles (Pimenta-Marques et al, 2016).

To tether ANA1 and other proteins to the centriole, we used a more generalised strategy. We developed a nanobody trapping experiment, using the anti-GFP single domain antibody fragment (vhhGFP4) (Caussinus et al, 2012; Saerens et al, 2005) fused to the PACT domain of D-PLP (Gillingham and Munro, 2000) to predominantly trap GFP tagged proteins to the oocyte centrioles.

We analysed egg chambers in stages 10 of oogenesis, a stage where centrioles are normally present, and observed that the PACT::vhhGFP4 construct is efficient for tethering GFP-tagged proteins to centrioles (Figs. 5A,B,F and EV4C,D). As a positive control, expressing GFP-Polo led to an increase in both γ-tubulin and CNN levels on the oocyte centrioles in comparison with the control (tethering of GFP alone) (Fig. 5A,C and EV4C,E), as previously observed for GFP-Polo-PACT (Pimenta-Marques et al, 2016). The levels of ANA1-GFP at the centrioles of stage 10 oocytes were also significantly increased by co-expressing PACT::vhhGFP4 (Figs. 5A,B and EV4C,D), showing that ANA1 is more efficiently

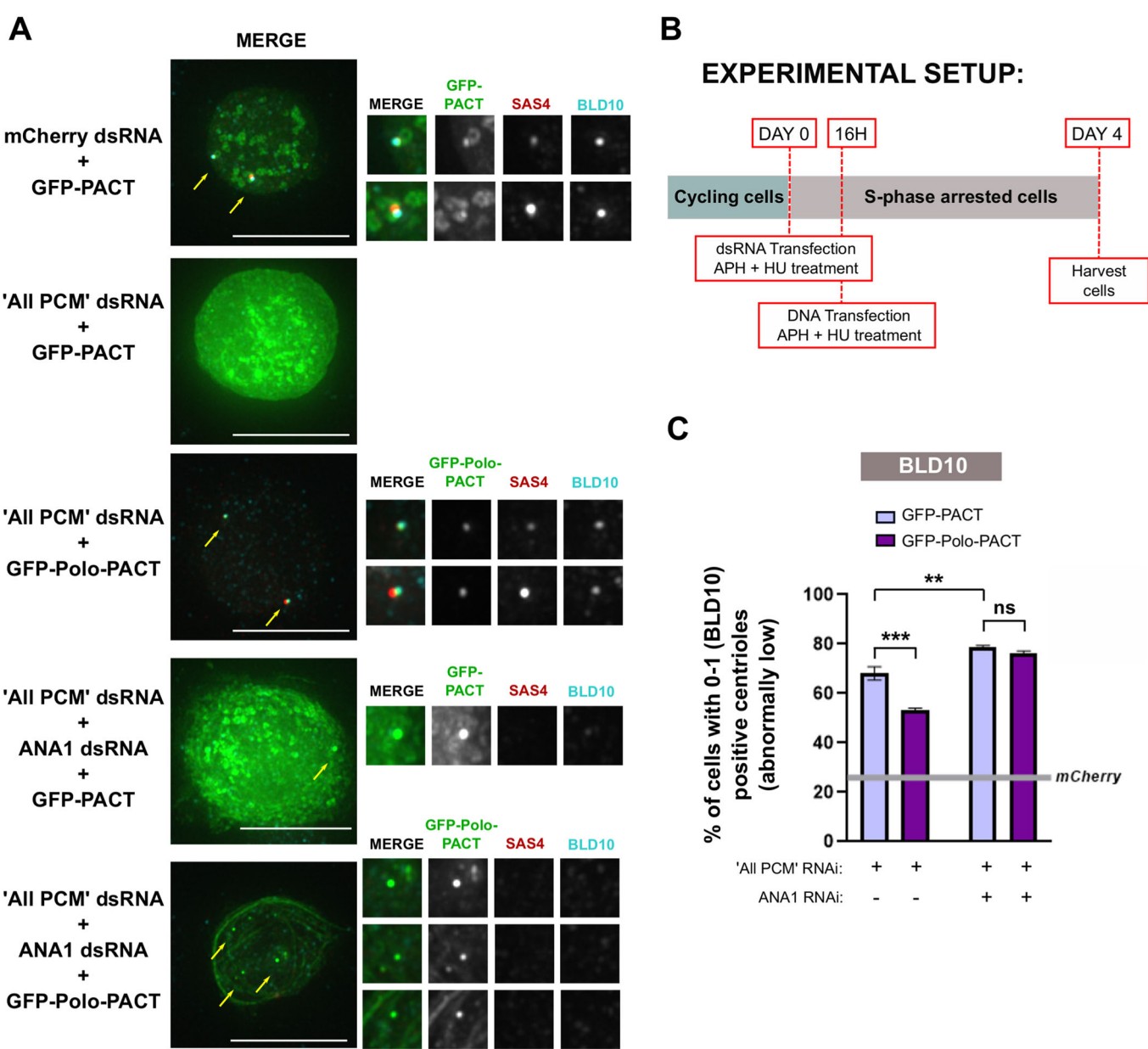

**Figure 3. Centrosome maintenance promoted by Polo kinase in cells is dependent on ANA1.**

(**A**) DMEL cells arrested in S-phase (treatment with hydroxyurea (HU) and aphidicolin (Aph)) were depleted of "All PCM" (simultaneous depletion of ASL, CNN, D-PLP and SPD-2), or simultaneous depletion of "All PCM" and ANA1 and transfected with either GFP-PACT or GFP-Polo-PACT. mCherry dsRNA was used as a negative control. Cells were immunostained for BLD10 (cyan) and SAS4 (red). Representative images are shown. All conditions were acquired with the same exposure. Arrows point to centrosomes in the different cells, which are enlarged in the figure insets. Note that although GFP aggregates are present when "All PCM" and ANA1 are co-depleted, they do not correspond to centrioles as there is no colocalization between GFP aggregates and BLD10 and/or SAS4. Scale bar, 10 μm. (**B**) Schematic representation of the experimental setup: DMEL cells were subjected to dsRNA transfection and treatment with Aph (aphidicolin) and HU (hydroxyurea) at day 0. Cells were depleted of "All PCM" as before, or simultaneously depleted of "All PCM" and ANA1. After 16 h, cells were transfected (GFP-PACT or GFP-Polo-PACT) in medium with Aph and HU. Note that RNAi in *Drosophila* cultured cells takes 3–4 days to have an effect (Bettencourt-Dias and Goshima, 2009), so it is very likely that GFP-Polo-PACT expression occurs before ANA1 or PCM RNAi have an effect. Cells were harvested and assayed for centriole numbers by immunofluorescence at day 4. (**C**) Quantification of the percentage of cells with abnormally low numbers of centrioles (i.e. 0–1). Centrioles were identified by considering the positive staining in each cell for the centriolar wall protein BLD10. The grey line represents the percentage of cells with 0–1 centrioles in the control (cells transfected with mCherry dsRNA and expressing GFP-PACT). Bars represent the mean ± SEM of three independent experiments. For each replicate in each experimental condition, $n \geq 100$ cells. A Two-way ANOVA, with Tukey's multiple comparisons test was used to test statistical significance. Source data are available online for this figure.

**A**

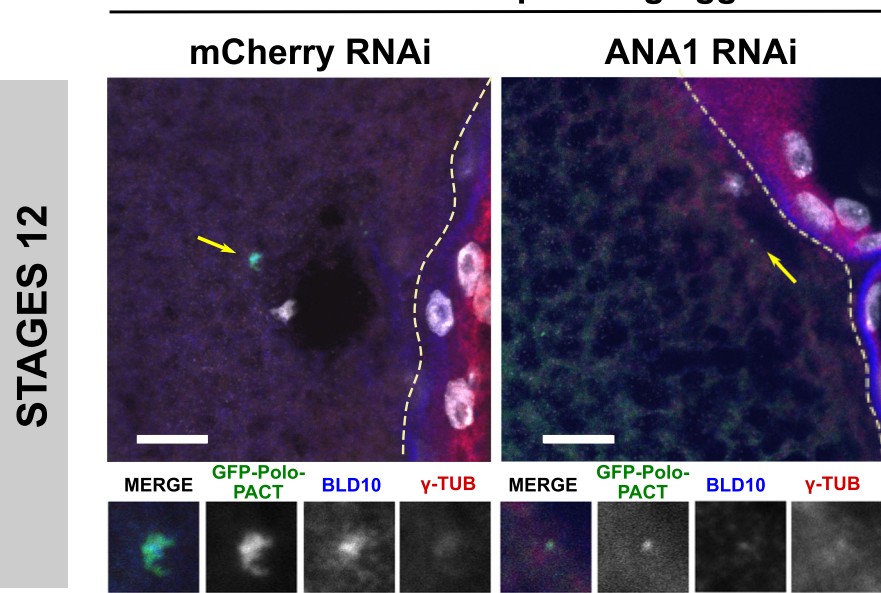

**B**

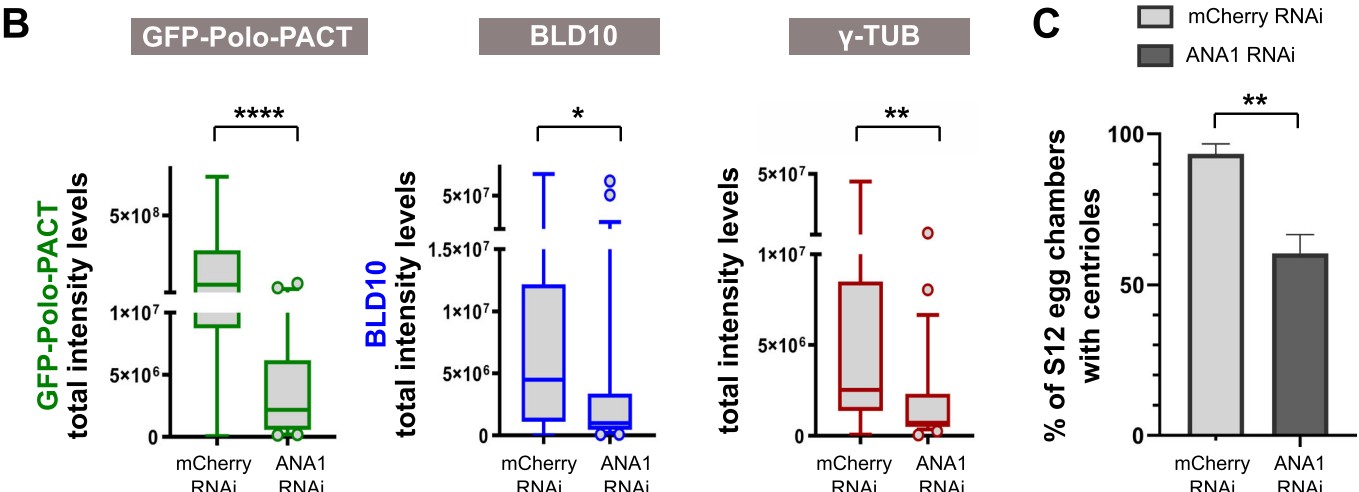

**Figure 4.  ANA1 expression is required for Polo´s activity in promoting centrosome maintenance in stage 12 egg chambers.**

(**A**) Immunostaining of egg chambers with different highlighted markers. GFP-Polo-PACT was expressed in the context of ANA1 or mCherry (negative control) RNAi. Arrows point to centrosomes in the different egg chambers, which are enlarged in the figure insets. Scale bars, 10 µm. (**B**) Quantification of total intensity levels of different centrosomal proteins (GFP-Polo-PACT, BLD10 and γ-TUB) in stages 12 of oogenesis, in GFP-Polo-PACT expressing oocytes. (**C**) ANA1 RNAi reduces the number of stage 12 egg chambers containing centrioles in the context of the expression of GFP-Polo-PACT. Shown are mean ± SEM. Data information: For (**B**) Box-and-whisker plots (2.5th and 97.5th percentiles) of the total integrated intensities of the different markers analysed. For quantification of GFP-Polo-PACT, for mCherry RNAi, $n = 29$, box minimum = 67,761, box maxima = 745,004,192, box median = 60,485,400, box 25% percentile = 8,751,230, box 75% percentile = 278,836,957; for ANA1 RNAi, $n = 27$, box minimum = 148,410, box maxima = 68,079,037, box median =  2,170,045, box 25% percentile = 564,124, box 75% percentile = 6,170,666. For BLD10 quantification, for mCherry RNAi, $n = 29$, box minimum = 22,016, box maxima = 72,457,040, box median = 4,477,697, box 25% percentile = 1,102,764, box 75% percentile = 12,170,274; for ANA1 RNAi, $n = 27$, box minimum = 89,923, box maxima = 65,099,874, box median = 999,490, box 25% percentile = 464,308, box 75% percentile = 3,366,269. For γ-TUB quantification, for control $n = 29$ box minimum = 68,956, box maxima = 45,034,891, box median = 2,534,961, box 25% percentile = 1,380,990, box 75% percentile = 8,508,945; for ANA1 RNAi, $n = 27$, box minimum = 57,831, box maxima = 11,097,411, box median = 730,827, box 25% percentile = 492,962, box 75% percentile = 2,314,277. Three independent biological replicates were performed. Statistical significance was determined by performing a bimodal regression test. See Statistical methods for details. For (**C**), $n = 28$ for mCherry RNAi and $n = 27$ for ANA1 RNAi. Three independent biological replicates were performed. Statistical significance was determined by performing a two-tailed Fisher exact test. For all the statistical tests used in this figure: *$p < 0.05$; **$p < 0.01$; ***$p < 0.001$. Source data are available online for this figure.

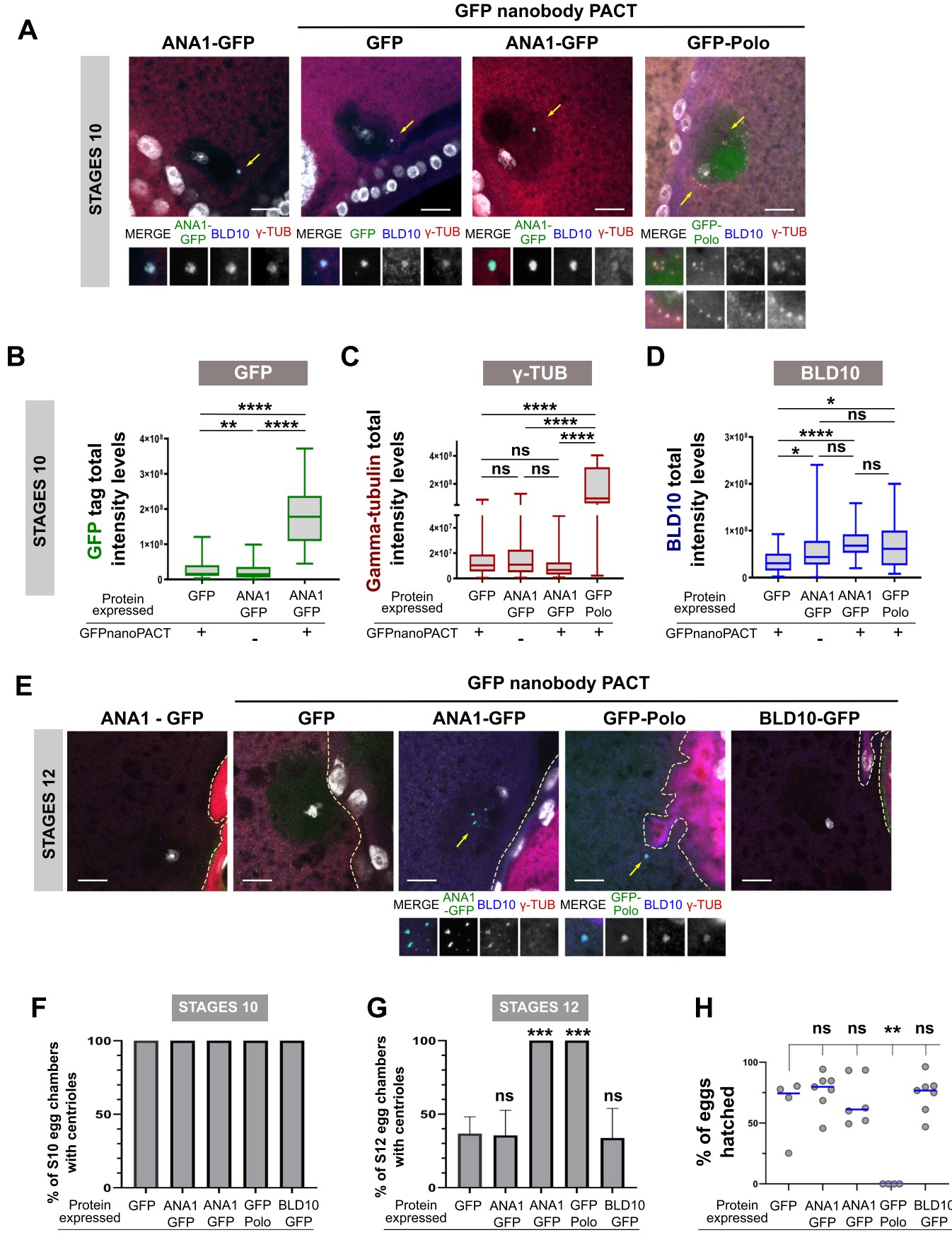

**Figure 5. Ectopic tethering of ANA1 to the oocyte centrioles, increases its levels in that structure and prolongs centriole presence.**

(A–D) Analysis of stage 10 oocytes upon tethering different centrosomal proteins to the oocyte centrioles by expressing a GFP nanobody construct fused to the PACT domain (PACT::vhhGFP4) that targets molecules to the centriole. Enlargements of the indicated areas (with centrioles, yellow arrows) are shown. Note that at this stage the oocyte is supposed to have ~64 clustered centrioles, known to be scattered when Polo-PACT is expressed (Pimenta-Marques et al, 2016). Scale bars, 10 μm. (B) Quantification of the total intensities of GFP and ANA1-GFP tethered to centrioles by PACT::vhhGFP4, as well as ANA1-GFP without tethering to the oocyte centrioles (without PACT::vhhGFP4 expression) in stages 10. Note that tethering of ANA1 to centrioles with PACT::vhhGFP4 leads to an increase in the levels of ANA1 on these structures. (C,D) Quantification of the total intensities of (C) γ-tubulin and (D) BLD10 in stage 10 egg chambers expressing either GFP, GFP-Polo or ANA1-GFP in combination with PACT::vhhGFP4. Quantifications were also performed for stage 10 egg chambers expressing ANA1-GFP without PACT::vhhGFP4 expression. Note that tethering GFP-Polo leads to an increase in the total levels of γ-tubulin on the oocyte centrioles in stages 10, which is not observed upon forced localization of ANA1 on the oocyte centrioles. (E) Representative images of each tested condition. Enlargements of the indicated areas (yellow arrows) are shown. Scale bars, 10 μm. Percentage of egg chambers from (F) stages 10 and (G) stages 12 showing the presence of centrioles (centrioles identified by the colocalization of the GFP signal with BLD10 staining). *P*-values were Bonferroni corrected for a Family Wise Error Rate of 5%. Shown are the mean ± SEM. (H) Quantification of the number of eggs hatched from the total number of eggs laid by females expressing GFP; GFP-Polo; ANA1-GFP, and BLD10-GFP in the presence of PACT::vhhGFP4 and females expressing ANA1-GFP alone without PACT::vhhGFP4. Each dot in the plot represents the percentage of eggs hatched from the total number of eggs laid by a single female (Kruskal–Wallis test). Data Information: For (B–D) Box-and-whisker plots (whiskers extend to the 2.5th and 97.5th percentiles) of the total integrated intensities of the different markers analysed. For (B), *n* = 27 for GFP + GFPnanoPACT, box minimum = 2,744,060, box maxima = 120,599,300, box median = 15,936,907, box 25% percentile = 10,030,880, box 75% percentile = 40,282,531; for ANA1-GFP, *n* = 32, box minimum = 1,172,971, box maxima = 98,563,643, box median = 14,766,941, box 25% percentile = 5,873,495, box 75% percentile = 35,560,508; for ANA1-GFP + GFPnanoPACT, *n* = 30, box minimum = 44,653,158, box maxima = 371,867,276, box median = 178,282,415, box 25% percentile = 108,858,510, box 75% percentile = 237,563,037. For (C), *n* = 27 for GFP + GFPnanoPACT, box minimum = 463,715, box maxima = 87,939,710, box median = 10,284,987, box 25% percentile = 5,509,216, box 75% percentile = 18,986,233. for ANA1-GFP, *n* = 32, box minimum = 618,947, box maxima = 129,786,121, box median = 10,900,274, box 25% percentile = 4,942,630, box 75% percentile = 22,715,755; for ANA1-GFP + GFPnanoPACT, *n* = 30, box minimum = 786,335, box maxima = 49,501,547, box median = 6,598,892, box 25% percentile = 3,098,434, box 75% percentile = 12,625,516; for GFP-Polo + GFPnanoPACT, *n* = 15, box minimum = 2,163,567, box maxima = 403,625,032, box median = 96,500,853, box 25% percentile = 61,587,773, box 75% percentile = 319,041,427. For (D), *n* = 27 for GFP + GFPnanoPACT, box minimum = 2,213,291, box maxima = 92710894, box median = 30,614,157, box 25% percentile = 15,128,141, box 75% percentile = 51,031,508. for ANA1-GFP, *n* = 32, box minimum = 618,947, box maxima = 240,703,745, box median = 43,841,602, box 25% percentile = 27,894,165, box 75% percentile = 78,486,010; for ANA1-GFP + GFPnanoPACT, *n* = 30, box minimum = 20,202,764, box maxima = 158,799,484, box median = 68,183,341, box 25% percentile = 53,042,973, box 75% percentile = 92,272,040; for GFP-Polo + GFPnanoPACT, *n* = 15, box minimum = 8,106,514, box maxima = 200,112,044, box median = 61,314,772, box 25% percentile = 26,259,130, box 75% percentile = 100,309,251. Statistical significance was determined by performing a bimodal regression test. See Statistical methods for details. For (G), *n* = 26 for GFP tethered with GFPnanoPACT, *n* = 30 for ANA1-GFP, *n* = 30 for ANA1-GFP tethered with GFPnanoPACT, *n* = 14 for GFP-Polo tethered with GFPnanoPACT and *n* = 29 for BLD10 tethered with GFPnanoPACT). Statistical significance was determined by performing a Two-tailed Fisher exact test. For (H), a minimum of four independent biological replicates were performed for each condition. significance was determined by performing a Kruskal–Wallis test. For all the statistical tests used in this figure: *p < 0.05; **p < 0.01; ***p < 0.001; ****p < 0.0001; ns, not statistically significant. Source data are available online for this figure.

recruited/maintained at centrioles with this approach. Interestingly, expressing ANA1 (tethered or not to centrioles) increases mildly the levels of BLD10, suggesting that ANA1 may contribute to stabilise the centriole structure by maintaining the levels of other important wall proteins (Fig. 5A,D). Tethering Polo also leads to a mild increase in BLD10 levels, which is consistent with its role in centriole maintenance. BLD10 levels are not different in oocytes expressing ANA1 (either tethered or not to centrioles) and Polo. However, in contrast to tethering of Polo, ANA1 did not promote additional recruitment/maintenance of the PCM components, γ-tubulin (Fig. 5A,C) and CNN (Fig. EV4C, E). Importantly, the observed increase in these PCM components upon tethering of Polo, is not due to an increase in centriole content, as the levels of BLD10 are not different in oocytes expressing ANA1 (either tethered or not to centrioles) and Polo (Fig. 5A,D). Consistent with this, while tethering of Polo to centrioles led to a mild but significant increase in the levels of a reporter for MTs (Jupiter-mCherry, a MT associated protein (Lowe et al, 2014)), the same was not observed when ANA1-GFP was tethered (Fig. EV4C,F).

We then analysed stages 12 of oogenesis, when centrioles normally start to be eliminated and asked whether ANA1 would be sufficient to keep their structural integrity, even if not capable of retaining their PCM. In the control condition, where GFP was tethered to the centriole, only ~30% of stage 12 oocytes showed the presence of centrioles (colocalization of GFP with BLD10) (Fig. 5E,G). As expected, GFP-Polo/PACT::vhhGFP4 expressing flies showed 100% of stage 12 oocytes with the presence of centrioles (Fig. 5E,G). When tethering ANA1 to the oocyte centrioles (ANA1-GFP/PACT::vhhGFP4 expressing oocytes),

100% of stage 12 oocytes showed the presence of centrioles (Fig. 5E,G). This data demonstrates that ANA1 can maintain centrioles in a context when both Polo and the PCM are naturally down-regulated (Jambor et al, 2015; Pimenta-Marques et al, 2016; Xiang et al, 2007). Importantly, this is likely to be a phenotype specific to ANA1, as tethering another centriolar wall protein, BLD10, was not sufficient to prevent normal centriole elimination (Fig. 5E,G). Given these centrioles do not recruit additional amounts of PCM, our data also shows that ANA1 is critical for maintaining centriole integrity, independently of its known role in PCM recruitment during centriole-to-centrosome conversion in cycling cells (Fu et al, 2016; Izquierdo et al, 2014).

Interestingly, tethering of ANA1 to centrioles did not lead to any obvious defects in meiosis, in contrast to tethering of Polo, (Pimenta-Marques et al, 2016). Moreover, flies expressing ANA1-GFP/PACT::vhhGFP4 were fertile and laid eggs which hatched at a comparable rate to control flies (expression of GFP/PACT::vhhGFP4) (Fig. 5H). These centrioles, which are structurally maintained are most likely inactive as they do not recruit PCM (Figs. 5C and EV4E) and do not nucleate MTs as discussed above (Fig. EV4F). This inactivity is significant, as it prevents potential interference with the meiotic spindle and subsequent embryonic nuclear divisions, which were observed when Polo was tethered to the centriole (Pimenta-Marques et al, 2016). Our observations in vivo suggest that maintaining a given threshold of ANA1 at the centriole provides stability to this structure. All together, these findings show that ANA1 is a critical player in providing integrity to fully assembled and matured centrioles.

The PCM may exert its function in centriole stability by recruiting and/or stabilising components, such as ANA1, within the

# A

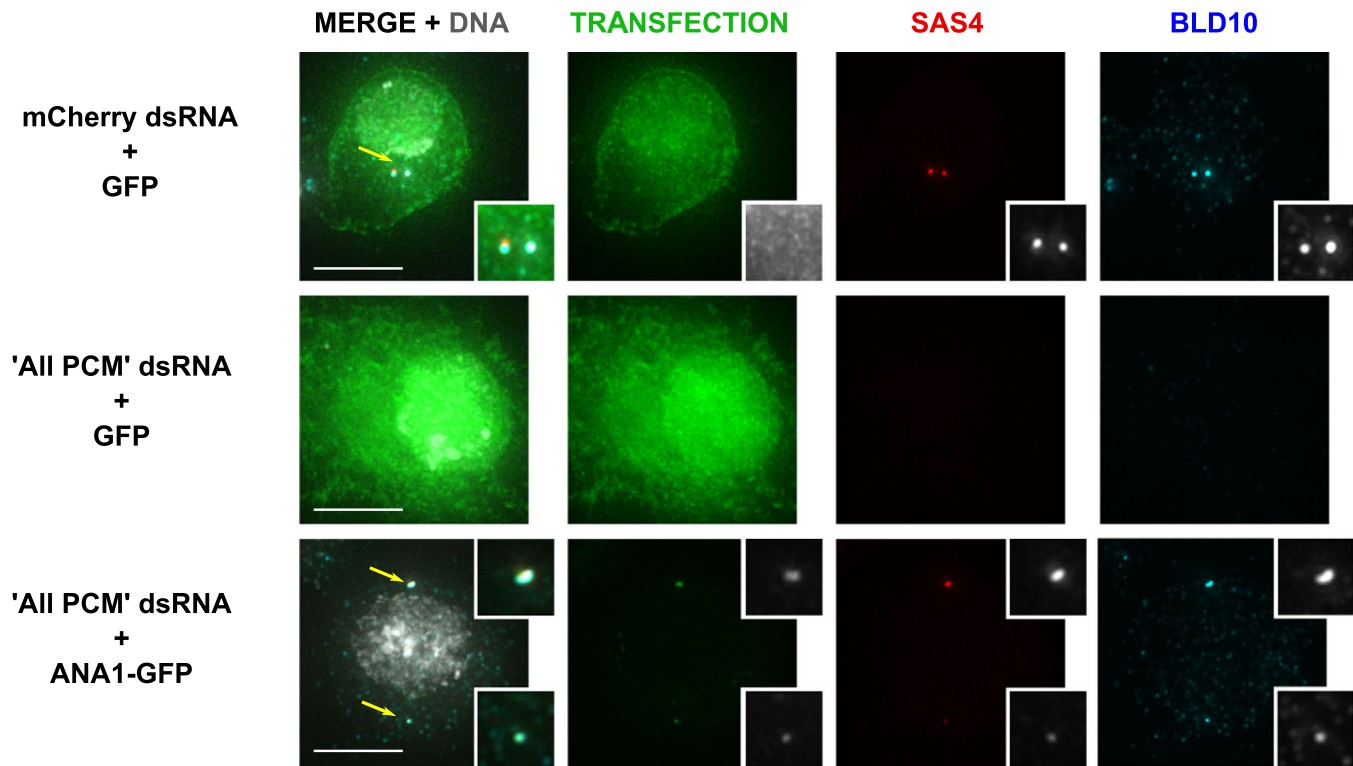

# B

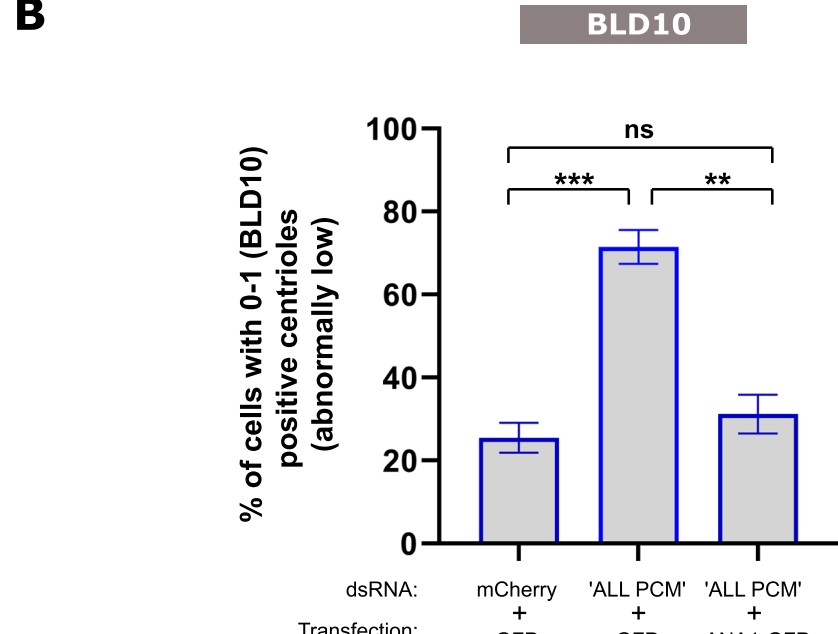

**Figure 6. ANA1 rescues the loss of centrioles induced by PCM depletion.**

(A,B) DMEL cells were subjected to dsRNA transfection and treatment with Aph (aphidicolin) and HU (hydroxyurea) at day 0. Cells were depleted of "All PCM" or mCherry as control. After 16 h, cells were transfected (GFP or ANA1-GFP) in medium with Aph and HU. (A) Cells were stained for SAS4 (red), BLD10 (cyan) and DNA (grey). Representative images are shown. Arrows point to centrosomes in the different cells, which are enlarged in the figure insets. Scale bar, 5 µm. (B) Histogram shows the percentage of cells with abnormally low numbers (i.e. 0–1). Centrioles were identified by considering the positive staining in each cell for the centriolar wall protein BLD10. For each replicate in each experimental condition, n > 40 cells. Bars represent the mean ± SEM of three independent experiments. One-way ANOVA, with Tukey's multiple comparisons test (**$p < 0.01$; ***$p < 0.001$; ns, not statistically significant). Note that expression of ANA1-GFP rescues centriole loss in the context of PCM depletion. Source data are available online for this figure.

centriole structure. If that is the case, we would expect that overexpression of ANA1 should at least partially rescue the loss of centrioles induced by PCM depletion. To test this hypothesis, we used the "centriole stability assay" in culture cells and over-expressed ANA1 in PCM-depleted cells (Fig. 6A). Expressing ANA1 rescued centriole loss observed in the "All PCM" RNAi condition, as assessed through both BLD10 (Fig. 6A,B) and SAS4 markers (Fig. 6A and EV5A,C). This rescue appears to be independent of residual PCM components (Fig. EV5B), as there was no increase in the proportion of centrioles that also had the PCM marker, SPD2 (Fig. EV5C). Altogether, our data suggests that ANA1 is important for maintaining the integrity of the centriole structure and that the PCM reinforces that role, perhaps through facilitating the incorporation and/or maintenance of ANA1 at the centriole, a need which is likely overridden by ANA1 over-expression. Notably, simultaneous depletion of PCM ("All PCM" RNAi) and ANA1 has a significantly stronger effect on centriole loss, in comparison with PCM depletion alone (Fig. 3C). This result suggests that the PCM may facilitate the incorporation and/or maintenance of other centriolar components which are also required for centriole integrity. Such components may encompass ANA2 and SAS6, which emerged as strong candidates for a role in centriole integrity (Fig. 1G).

## Discussion

Our work suggests that the turnover, likely by slow exchange, of critical centrosome components is needed to maintain the integrity of the structure. Such components include proteins from the centriole wall and the cartwheel, as well as the PCM and Polo kinase. Amongst those components, we show that the conserved centriolar wall protein ANA1 plays a critical role in the maintenance of fully matured centrioles in cultured cells and in the female germline. In fact, tethering of ANA1 to the oocyte centrioles is sufficient for their maintenance up to stages where they are normally absent. Moreover, overexpression of ANA1 in S-phase arrested culture cells can rescue the centriole loss that is normally observed upon "All PCM" depletion. We further show that Polo depends on ANA1 for its function in centrosome maintenance, as in both germline and tissue culture cells, ANA1 RNAi impairs Polo-induced centrosome maintenance. Altogether our data suggests that the centriole is more dynamic than previously thought.

Our data also separates pathways. It shows that PLK4 is not critical for the maintenance of centrosome integrity, suggesting that centriole biogenesis and maintenance pathways are differentially regulated. Moreover, in contrast to Polo tethering phenotype, centrioles which are maintained after ANA1 tethering are not

functional, suggesting that Polo has a more comprehensive function, which includes both centriole integrity and centrosome function.

Here we show that ANA1 is not just necessary for centriole biogenesis (Blachon et al, 2008, 2009; Dobbelaere et al, 2008) and for centriole-to-centrosome conversion (Fu et al, 2016), it is also an essential player in the maintenance of centriole integrity and therefore, a critical protein for the entire lifespan of the centrosome. Very recently, the use of several superresolution techniques revealed that ANA1 localises from the pinheads to the outer edge of the doublet microtubules (Tian et al, 2021). ANA1 contains multiple predicted coiled-coil regions (Saurya et al, 2016), which could potentially promote the interaction with different centrosomal players. One possibility is that ANA1 maintains the microtubule doublets "linked" to each other, promoting the integrity of the wall. This can possibly occur either through direct binding of ANA1 to the microtubule doublets (Chang et al, 2016), or indirectly by its interaction with proteins which provide such links such as BLD10 (Carvalho-Santos et al, 2012), or through both pathways. In addition, ANA1 may also promote cartwheel stability as it was found to interact with SAS6 and ANA2 by yeast two-hybrid (Galletta et al, 2016). This central role would also explain the observations made in our candidate screen (Fig. 1), where depletion of ANA1 had one of the strongest effects on centrosome maintenance.

How can the PCM contribute to centriole integrity? In vitro work in *C. elegans* showed that the PCM can function as a condensate with self-assembling properties, allowing the selective concentration of different components. This was enhanced by Polo/PLK1 (Woodruff et al, 2017). Moreover, in *Drosophila* egg extracts, γ-tubulin was proposed to concentrate components to promote de novo centriole biogenesis (Nabais et al, 2021). Given that ANA1 overexpression rescues the centriole instability phenotype resulting from PCM loss, we hypothesise that the PCM promotes a stable concentration of centriolar proteins at the centrosome, and/or regulates their exchange with a cytoplasmic pool. Overexpression of ANA1 would rescue its presence at the centrosome.

How does Polo contribute to centriole integrity? It is possible that Polo participates in this pathway by increasing the local PCM pool. However, this cannot be the only mechanism, as centriole-tethered Polo can partially rescue centriole loss, even in the absence of PCM (Figs. 3C and EV2). Given that ANA1 phosphorylation by Polo at predicted Polo-box binding domains is not required for centriole structural integrity (Alvarez-Rodrigo et al, 2021), it is possible that Polo helps to recruit ANA1 directly through phosphorylation on other sites, or by direct interaction with parts of the molecule independently of phosphorylation (Pimenta-Marques et al, 2016). Alternatively, Polo may indirectly promote ANA1 recruitment through interactions and/or phosphorylation of

other centriole components. Future work will aim at further clarifying this mechanism.

Oocytes lose their centrioles before fertilisation. Non cycling cells, such as neurons, muscle, and epithelial cells, often attenuate the activity of the centrosome as a MTOC via loss of PCM (Muroyama and Lechler, 2017; Sanchez and Feldman, 2017; Tillery et al, 2018). In these cells it is not clear whether the centrioles are eventually lost or remain inactive throughout their whole lifespan. Moreover, it has been suggested that centrosome inactivation is a mechanism used by cancer cells to silence supernumerary centrosomes that otherwise could lead to cell death (Sabino et al, 2015). Our work shows that while misregulation of Polo in eggs can lead to infertility in flies, that is not the case for ANA1, as ectopic expression of ANA1 is not sufficient for retaining centrosome function. In future studies it will be of critical importance to address how deregulation of proteins such as ANA1, Polo and the PCM might differentially change the integrity/activity of centrosomes in these contexts and possibly contribute to disease, in particular infertility and cancer.

# Methods

## *Drosophila* stocks and genetics

### Fly stocks

The following fly stocks were used in this study: from Bloomington Stock Centre: UASp-DegradFP (y[1] w*; M{UASp-Nslmb.vhhGFP4}ZH-51D; #58740); Df(3R)Exel7357 (W[1118]; Df(3 R)Exel7357/TM6B, Tb[1]; a deficiency for the *ANA1* locus #7948); Ubi-GFP (y[1] w[67c23] P{Ubi-GFP.D}ID-1; #1681); UAS-ANA1-RNAi (y[1] v[1]; P{TRiP.HMJ23356}attP40; #61867, UAS-mCherry-RNAi (y[1] v[1] sc* sev[21];; P{VALIUM20-mCherry}attP2 #35785) and matalpha4-Gal4 (w*; P{matalpha4-GAL-VP16}V2H; #7062). Furthermore, W; ANA1-GFP/CyO, W; ANA1-td-Tomato / CyO and W; BLD10-GFP/CyO (Blachon et al, 2008); *ana1*[mecB] mutant flies (Blachon et al, 2008); the maternal germline-specific G302-Gal4/TM6B and endogenous Jupiter-mCherry (Lowe et al, 2014) kindly provided by Daniel St. Johnston; Gordon Institute, UK); w; UASp-GFP-PACT; w; UASp-GFP-Polo-PACT; w ;UASp-GFP-Polo and w, PUbq-GFP PACT (Pimenta-Marques et al, 2016).

The following transgenes were generated for this study:

**Nanobody construct**: The coding sequences of PACT (Gillingham and Munro, 2000; Pimenta-Marques et al, 2016) and vhhGFP4 ((pGEX6P1-GFP-Nanobody was kindly provided by Kazuhisa Nakayama (Addgene plasmid # 61838)) were PCR amplified (primers used are provided on Table EV3). PCR products were purified, excised independently, ligated into the pSpark®-TA Done vector (Canvax), and then transformed into Escherichia coli 'DH5α' competent cells. Inserts from at least two different clones were sequenced by the Sanger method. The generated construct (pSpark_GFPnanobody-PACT) was linearized with the restriction enzyme KpnI and cloned into the pDONR™ 221 Vector (Thermo-Fisher #12536017) Using the gateway® system to generate an entry clone. To create the expression vectors, recombination reactions were achieved using the created entry vector and the destination vector pUbq-phi31. Transgenic flies were generated via plasmid injection (BestGene, INC) using the pUbq-phi31_GFPnanobody-PACT.

The following combination of fly genotypes were generated for this study:

*ana1*[mecB] was recombined to G302-Gal4, generating flies W; ; ana1[mecB] G302-Gal4

Ana1-GFP; ana1[mecB] G302-Gal4

UASp-DegradFP; Df(3R)Exel7357

Matalpha4-Gal4; UAS-mCherry-RNAi

UAS-ANA1-RNAi; UASp-GFP-PACT

UAS-ANA1-RNAi; UASp-GFP-Polo-PACT

Ubi-GFPnanobodyPACT; G302-Gal4

Ubi-GFPnanobodyPACT; G302-Gal4::mCherry-Jupiter

ANA1-td-Tomato:: PUbq-GFP-PACT/CyO; G302-Gal4/TM6B, Tb

### Fly husbandry

To degrade ANA1-GFP specifically in the female germ-line at stages 3–4 of oogenesis, flies of the genotype Ana1-GFP; ana1[mecB] G302-Gal4 were crossed to flies of the genotype UASp-DegradFP; Df(3R)Exel7357. As control, flies of the genotype Ana1-GFP; ana1[mecB] G302-Gal4 were crossed to Df(3R)Exel7357.

To deplete ANA1 specifically in the female germ-line in the context of Polo-mediated forced maintenance of centrioles, the following crosses were performed: Flies of the genotypes UAS-ANA1-RNAi; UASp-GFP-Polo-PACT and UAS-ANA1-RNAi; UASp-GFP-PACT were crossed to Matalpha4-Gal4. As controls, Matalpha4-Gal4; UAS-mCherry-RNAi were crossed to either UASp-GFP-Polo-PACT or UASp-GFP-PACT.

To tether different GFP-tagged proteins to the oocytes centrioles in the female germ-line, flies of the genotype Ubi-GFPnanobody-PACT; G302-Gal4 were crossed to flies of the following genotypes: ANA1-GFP/CyO; UASp-GFP-Polo; BLD10-GFP and Ubi-GFP as control.

All strains were raised on standard medium at 25 °C, using standard techniques.

### Egg laying and hatching

Single well-fed virgin females with 1-day old were mated with two w[1118] males in cages with agar plates supplemented with apple juice. The number of eggs laid was counted for 6 days. Each plate was kept at 25 °C for 3 extra days and examined for the number of larvae hatched. Egg hatching rates were calculated as the percentage of larvae hatched from the total number of eggs laid by each female. More than 4 independent crosses were performed for each phenotype.

### Ovaries immunostaining

Ovary stainings were performed as previously described (Pimenta-Marques et al, 2016). Briefly, females were transferred to pre-warmed (25 °C) BRB80 buffer (80 mM Pipes pH 6.8, 1 mM MgCl2, 1 mM EGTA) supplemented with 1× protease inhibitors (Roche), and their ovaries were extracted with pre-cleaned forceps. Individualized ovaries were then incubated for 1 hour (h) at 25 °C in BRB80 with 1% Triton X-100 without agitation, followed by a 15 minutes (min) fixation step at −20 °C in chilled methanol. 3 wash steps of 15 min each and overnight permeabilization were done in PBST (1× PBS with 0.1% Tween). Blocking for 1 h was done in PBST with 2% BSA (Gibco). Primary antibodies were incubated overnight at 4 °C in PBS with 1% BSA (PBSB) followed by 3 wash steps. Secondary antibodies were diluted in PBSB and

incubated for 2 h at room temperature (RT). Ovaries were washed in PBS and DNA was counterstained with DAPI.

### Imaging, analysis, and quantification

*Drosophila* egg chambers were imaged as Z-series (0.3 μm z-interval) on a Zeiss LSM 980, using confocal mode. All images were acquired with the same exposure. Images were processed as sum-intensity projections and intensity measurements were performed using ImageJ software (NIH). We have previously found that in stages 10 of oogenesis, centrosomes are still present, with centriolar loss being more pronounced at stages 12 (Pimenta-Marques et al, 2016). In the stages of oogenesis investigated in this manuscript, the centrosomes of the oocyte are clustered, precluding the individualization of centrosomes. Quantification was performed on the clustered centrosomes, identified based on the colocalization of different centrosomal markers, as previously validated and performed (Pimenta-Marques et al, 2016). To assess the background level, the intensity of three different regions was measured and subtracted to the centrosomal region. After background subtraction, intensity of different centrosomal markers was quantified by measuring the sum intensity on the areas containing centrosomes. In experiments where presence/absence of signal was evaluated, presence of signal was defined as a significant signal above oocyte background. Image panels were assembled using QuickFigures (Mazo, 2021).

### Protein depletion in DMEL cells

*Drosophila melanogaster* culture cells (DMEL; ATCC CRL-1963) were maintained in Express5 SFM medium (Gibco, UK), supplemented with 2 mM L-Glutamine (ThermoFisher Scientific, UK). Double-stranded RNA (dsRNA) were performed as previously described (Bettencourt-Dias et al, 2004). 10 million cells were used for dsRNA transient transfection. dsRNA amounts used in the screen: 20 μg individual *CNN, ASL, D-PLP*, and *SPD2* for "All PCM"; 40 μg *PLK4*, 40 μg *CP110*, 40 μg *CEP97*,40 μg *ANA2*, 40 μg *SAS6*, 40 μg *ANA1*, 40 μg *BLD10*, 40 μg *SAS4*, 80 μg *mCherry* dsRNA. dsRNA combination amounts used for Fig. 3: "All PCM" combined with 20 μg of *mCherry* dsRNA or 20 μg of *ANA1* 5'-3'-UTR dsRNA; 100 μg of *mCherry* dsRNA. Primers used for dsRNA production are listed in Table EV1. Sequence used to generate dsRNA for *ANA1* 5'-3'-UTR is presented in Table EV2.

### Centriole stability assay

Centriole stability assay was performed as previously described (Pimenta-Marques et al, 2016). Briefly, DMEL cells were S-phase-arrested with 10 μM aphidicolin (Aph), a specific eukaryotic DNA polymerase inhibitor, and 1.5 mM hydroxyurea (HU), which reduces deoxyribonucleotide production, 1 h after dsRNA transfection. In case the "centriole stability assay" (Fig. 1) lasted for 8 days, cells were subject to a second round of dsRNA transfection and Aph+HU treatment after 4 days. In case the assay lasted for 4 days (Figs. 3 and 5), cells were collected after 4 days of dsRNA and Aph + HU treatment.

### Plasmid transfections

DMEL cells were transiently transfected with GFP-PACT, GFP-POLO-WT-PACT, GFP, or ANA1-GFP after transfection with dsRNA either *mCherry* dsRNA (control), "All PCM" dsRNA or "All PCM" + *ANA1* dsRNA. Since GFP-PACT and GFP-POLO-WT-

PACT constructs contain an UASp promoter, each of these constructs were simultaneously co-transfected with an Actin5C-Gal4 plasmid. Plasmid transfections were performed as previously described (Pimenta-Marques et al, 2016).

### Immunostaining and imaging of D. melanogaster culture cells

DMEL cells were plated into glass coverslips and allowed to adhere for 1 h at 25 °C. Cells were then fixed for 10 min at room temperature with a solution containing 4% paraformaldehyde, 60 mM PIPES pH 6.8, 30 mM HEPES pH7.0, 10 mM EGTA pH 6.8, 4 mM MgSO$_4$. After 3 washes, cells were permeabilized and blocked with PBSTB (a PBS solution containing 0.1% Triton X-100 and 1% BSA). Cells were incubated with primary antibodies diluted in PBSTB overnight at 4 °C. After 3 washes, cells were incubated with secondary antibodies and DAPI (Santa Cruz Biotechnology) diluted in PBSTB for 2hH at RT. Cells were mounted with Vectashield Mounting Medium (Vector laboratories). Cell imaging was performed in Deltavision OMX (Deltavision) microscope with a PCO Edge 5.5 sCMOS 2560 × 2160 camera, or with a Nikon High Content Screening (Nikon) microscope with an Andor Zyla 4.2 sCMOS 4.2Mpx. All images were acquired with the same exposure. Images were acquired as Z-series (0.2 μm z-interval) and analysed as maximum intensity projections. Image panels were assembled using QuickFigures (Mazo, 2021).

### Antibodies

Primary antibodies and dilutions used: chicken anti-PLP (1:1000 for DMEL cells immunostaining; 1:500 for ovary immunostaining), kindly provided by David Glover,University of Cambridge, UK (Bettencourt-Dias et al, 2005); rabbit anti-Bld10 (1:2000, kindly provided by Timothy Megraw, The Florida State University, USA (Mottier-Pavie and Megraw, 2009); rat anti-ANA1 (1:500, kindly provided by Jordan Raff, University of Oxford, UK (Saurya et al, 2016); mouse anti-γ-tubulin (1:50 dilution; clone GTU88, Sigma, cat. n° T5326, RRID AB_532292); rabbit anti-SAS4 (1:500, Metabion); rabbit anti-CP110 (1:10,000 for DMEL cells immunostaining; 1:5000 for ovary immunostaining, Metabion) (Nabais et al, 2021); rabbit anti-CNN (1:500), kindly provided by Thomas Kaufman. Secondary antibodies (Jackson Immunoresearch Europe) were used at 1:1000 for Dmel cells immunostaining, and 1:250 for ovaries immunostaining.

### Statistics

For all experiments at least three independent biological replicates were performed for each condition. The number of cells and egg chambers analysed for each experiment are detailed in the respective figure legends.

**Candidate screen in S2 cells (Figs. 1 and EV1)**: A minimum of 100 cells was counted for each biological replicate of each condition. Cells were classified into classes 0–1, 2 and >2 according to the number of centrioles observed. To quantify the impact and (statistical) significance of each RNAi treatment on the stability of centrioles, we performed a binomial regression on the number of cells with 0–1 centrioles, out of all cells measured, for each marker (BLD10, D-PLP, ANA1, CP110, and SAS4), with replicate as a categorical covariate. This allows quantification of the effect of the RNAi treatment (vs mCherry RNAi), controlling for replicate variation, and provides more statistical power than a simple ANOVA on the fractions of each replicate.

**Experiments in oogenesis:** In all experiments, we conducted three independent biological replicates for each condition. In oogenesis it is difficult to provide large datasets, therefore, we pooled together the replicates from each condition. To control for differences between the different replicates and to properly quantify differences between different conditions, we regressed the log-transformed intensity values of each condition, using an intercept per block/replicate to control for block variability. Total "n" for each condition in each experiment are presented in the respective figure legend.

## Data availability

This study includes no data deposited in external repositories.

## Peer review information

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

## Acknowledgements

We thank all members of the Cell Cycle and Regulation Lab for the discussions and for the critical reading of the manuscript. We thank Tomer Avidor-Reiss (University of Toledo, Toledo, OH), Daniel St. Johnston (The Gurdon Institute, Cambridge, UK), David Glover (University of Cambridge, Cambridge, UK), Jingyan Fu (Agricultural University, Beijing, China) Jordan Raff (University of Oxford, Oxford, UK) and Timothy Megraw (Florida State University, Tallahassee, FL) for sharing tools. We acknowledge the technical support of Instituto Gulbenkian de Ciência (IGC)'s Advanced Imaging Facility, in particular Gabriel Martins, Nuno Pimpão Martins and José Marques. We also thank Tiago Paixão from the IGC's Quantitative & Digital Science Unit and Marco Louro from the CCR lab for the support provided on statistical analysis. IGC's Advanced Imaging Facility (AIF-UIC) is supported by the national Portuguese funding ref# PPBI-POCI-01-0145-FEDER -022122. We thank the IGC's Fly Facility, supported by CONGENTO (LISBOA-01-0145-FEDER-

022170). This work was supported by an ERC grant (ERC-2015-CoG-683258) awarded to MBD and a grant from the Portuguese Research Council (FCT) awarded to APM (PTDC/BIA-BID/32225/2017).

## Author contributions

**Ana Pimenta-Marques**: Conceptualization; Data curation; Formal analysis; Supervision; Funding acquisition; Validation; Investigation; Visualization; Methodology; Writing—original draft; Writing—review and editing. **Tania Perestrelo**: Conceptualization; Resources; Data curation; Formal analysis; Validation; Investigation; Visualization; Methodology; Writing—original draft; Writing—review and editing. **Patricia Reis-Rodrigues**: Data curation; Formal analysis; Validation; Investigation; Visualization; Methodology; Writing—review and editing. **Paulo Duarte**: Resources; Validation. **Ana Ferreira-Silva**: Formal analysis; Validation; Investigation. **Mariana Lince-Faria**: Data curation; Formal analysis; Validation; Investigation; Writing—review and editing. **Monica Bettencourt-Dias**: Conceptualization; Supervision; Funding acquisition; Validation; Methodology; Project administration; Writing—review and editing.

## Disclosure and competing interests statement

The authors declare no competing interests.

# Expanded View Figures

**Figure EV1. Candidate screen in *Drosophila* cultured cells for centrosome maintenance.**                                                                                     ▶

(A) Schematic representation of the different proteins that were depleted in each of the centrosome modules tested for maintenance. These include: "ALL PCM" proteins (simultaneous depletion of four major PCM proteins: ASL, CNN, D-PLP and SPD-2); centriole cap proteins (CEP97 and CP110); the major regulator of centriole biogenesis, PLK4; cartwheel proteins (ANA2 and SAS6) and the centriolar wall proteins (BLD10, SAS4 and ANA1). (B–F) Centriolar numbers were assessed considering the positive staining in each cell for different centrosome markers. These include: the PCM marker D-PLP (orange bars); the centriole wall markers SAS4 (green bars), BLD10 (dark green bars) and ANA1 (light green bars) and the distal cap protein CP110 marker (blue bars). Histograms represent the percentage of cells with abnormally low centriole numbers (i.e. 0-1) (B) Depletion of "All PCM" (C) Depletion of the centriolar cap proteins CP110 or CEP97; (D) Depletion of the centriolar biogenesis regulator PLK4, (E) Depletion of the cartwheel proteins ANA2 or SAS6, and (F) depletion of the centriolar wall proteins BLD10, ANA1, or SAS4. Data information: Bars represent the mean ± SEM of three independent experiments ($n \geq 100$ cells per replicate in each condition). Statistical significance was determined by performing a bimodal regression test. The impact of the different RNAi treatments on the number of cells with 0-1 centrioles was estimated on the number of cells that present a reduced number of centrioles. Estimates indicate the log odds ratio of the indicated treatment on increasing the number of cells with 0-1 centrioles. See Statistical methods for more details. *$p < 0.05$; **$p < 0.01$; ***$p < 0.001$; ****$p < 0.0001$; ns, not significant (see also Fig. 1). Source data are available online for this figure.

                                                                                            

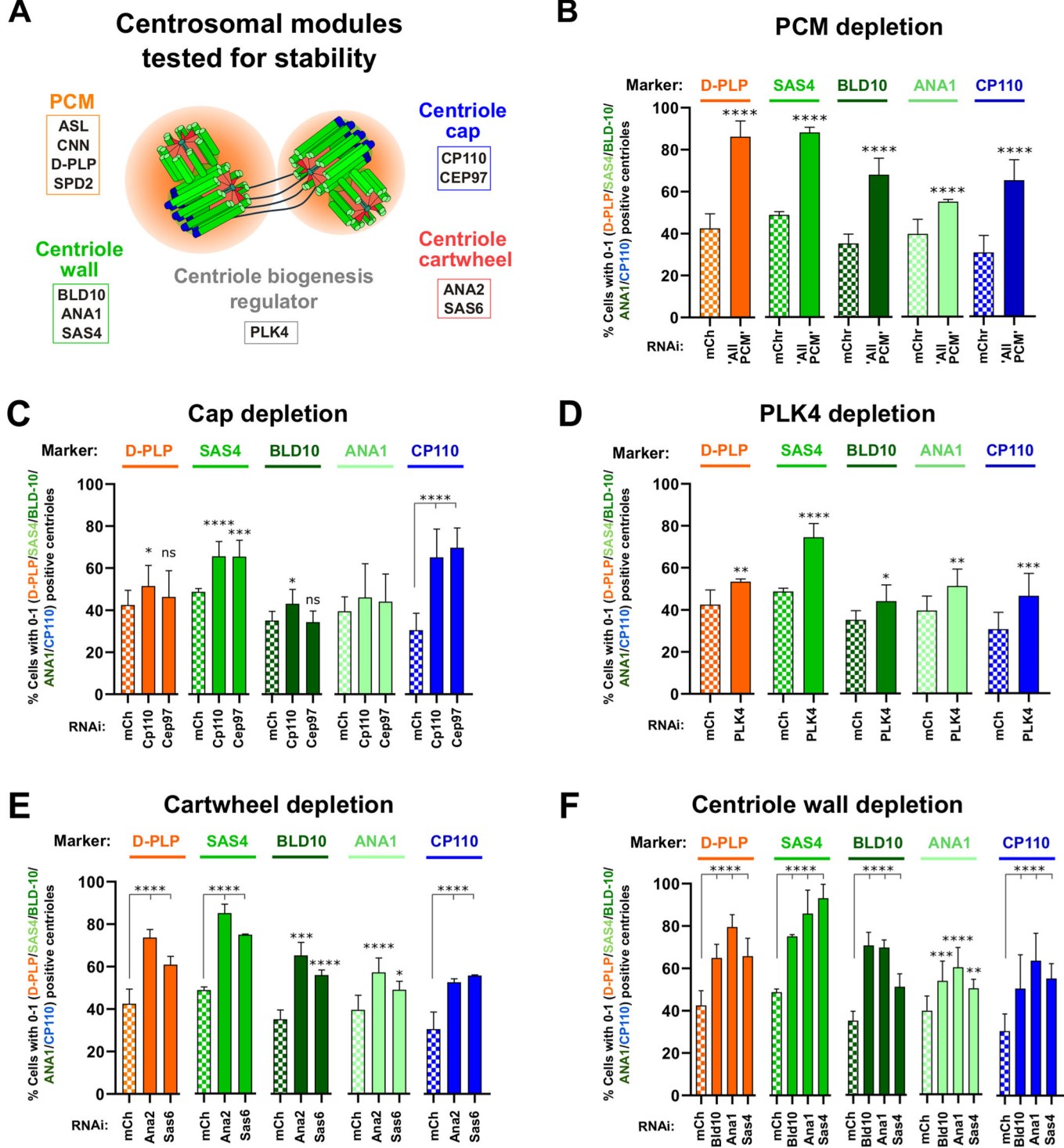

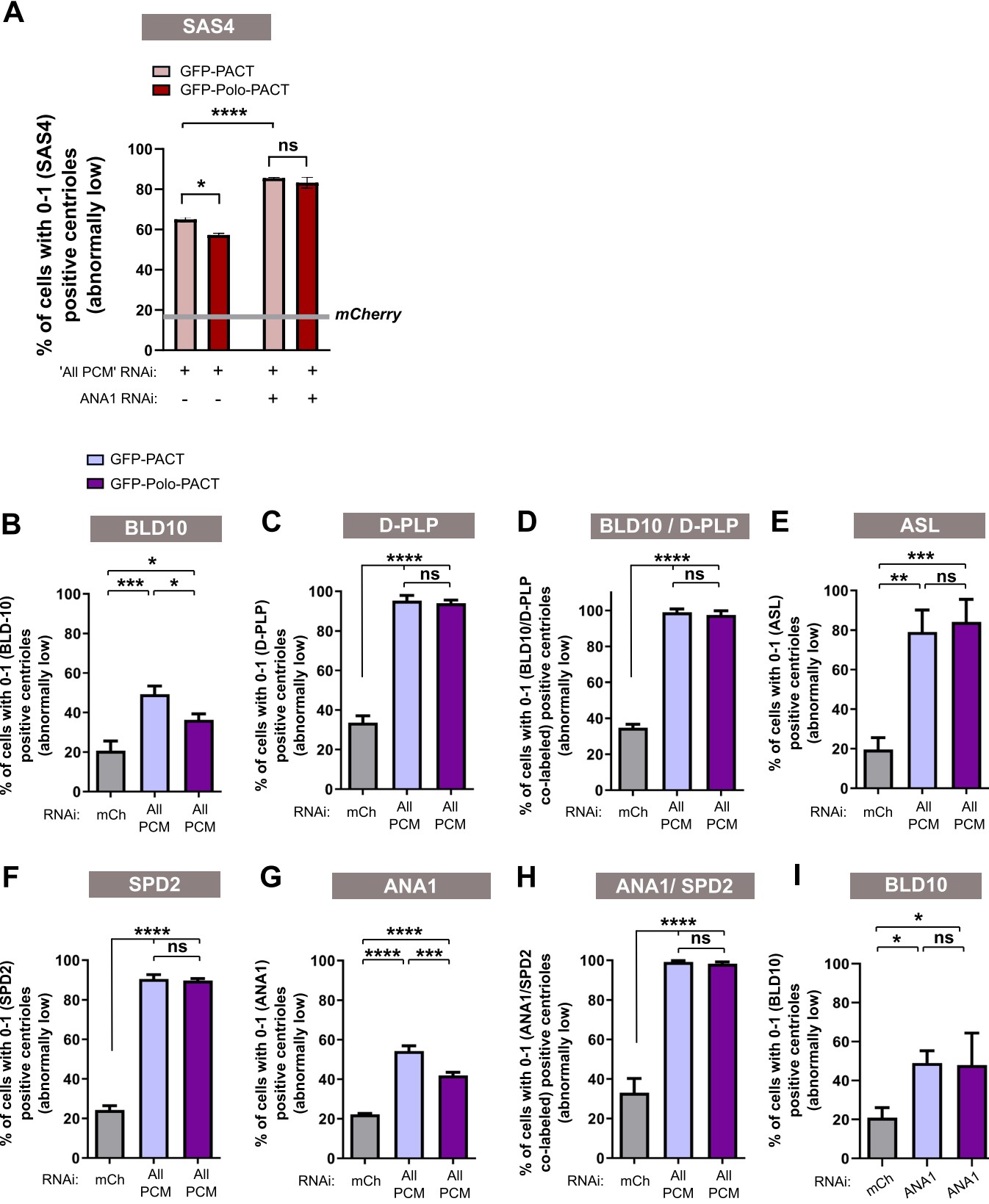

◀ **Figure EV2. Polo kinase induced centriole stability is dependent on ANA1.**

(A–I) DMEL cells were subjected to dsRNA transfection and treatment with Aph (aphidicolin) and HU (hydroxyurea) at day 0. Cells were harvested and assayed for centriole numbers by immunofluorescence at day 4. (A) Depletion of ANA1 prevents Polo-PACT induced centriole rescue, in the context of PCM RNAi. Histogram shows the percentage of cells with abnormally low numbers (i.e. 0-1) of centrioles labelled by SAS4. (B–H) Cells were depleted of "All PCM" or mCherry (control). After 16 h, cells were transfected (GFP-PACT or GFP-Polo-PACT) in medium with Aph and HU. Cells were harvested and assayed for centriole numbers by immunofluorescence at day 4. Quantification of the percentage of cells with abnormally low numbers of centrioles (i.e. 0-1). Centrioles were identified by considering the positive staining in each cell for BLD10 (B), D-PLP (C), ASL (E), SPD-2 (F) and ANA1 (G). Cells with co-staining for BLD10 and D-PLP (D) as well as ANA1 and SPD2 (H) were also quantified. Note that upon "All PCM" RNAi there is a large increase in the percentage of cells with abnormally low PCM foci per cell (zero or one) in the case of D-PLP, ASL or SPD2, which is not reduced upon GFP-Polo-PACT expression. (I) Cells were depleted of ANA1 or mCherry (control). After 16 h, cells were transfected (GFP-PACT or GFP-Polo-PACT) in medium with Aph and HU. Cells were harvested and assayed for centriole numbers by immunofluorescence at day 4. Quantification of the percentage of cells with abnormally low numbers of centrioles (i.e. 0-1). Centrioles were identified by considering the positive staining for the centriolar wall protein BLD10. Data information section: Bars represent the mean ± SEM of three independent biological replicate experiments. For (A), $n > 100$ cells per replicate, per condition in each experiment. For (B–H) $n > 80$ cells per condition in each experiment. For (I) n between 74–128 cells per condition in each experiment. For all the data in this figure, statistical significance was determined by a two-way ANOVA, with Tukey's multiple comparisons test. For all the statistical tests $*p < 0.05$; $**p < 0.01$; $***p < 0.001$; $****p < 0.0001$; ns, not statistically significant. Source data are available online for this figure.

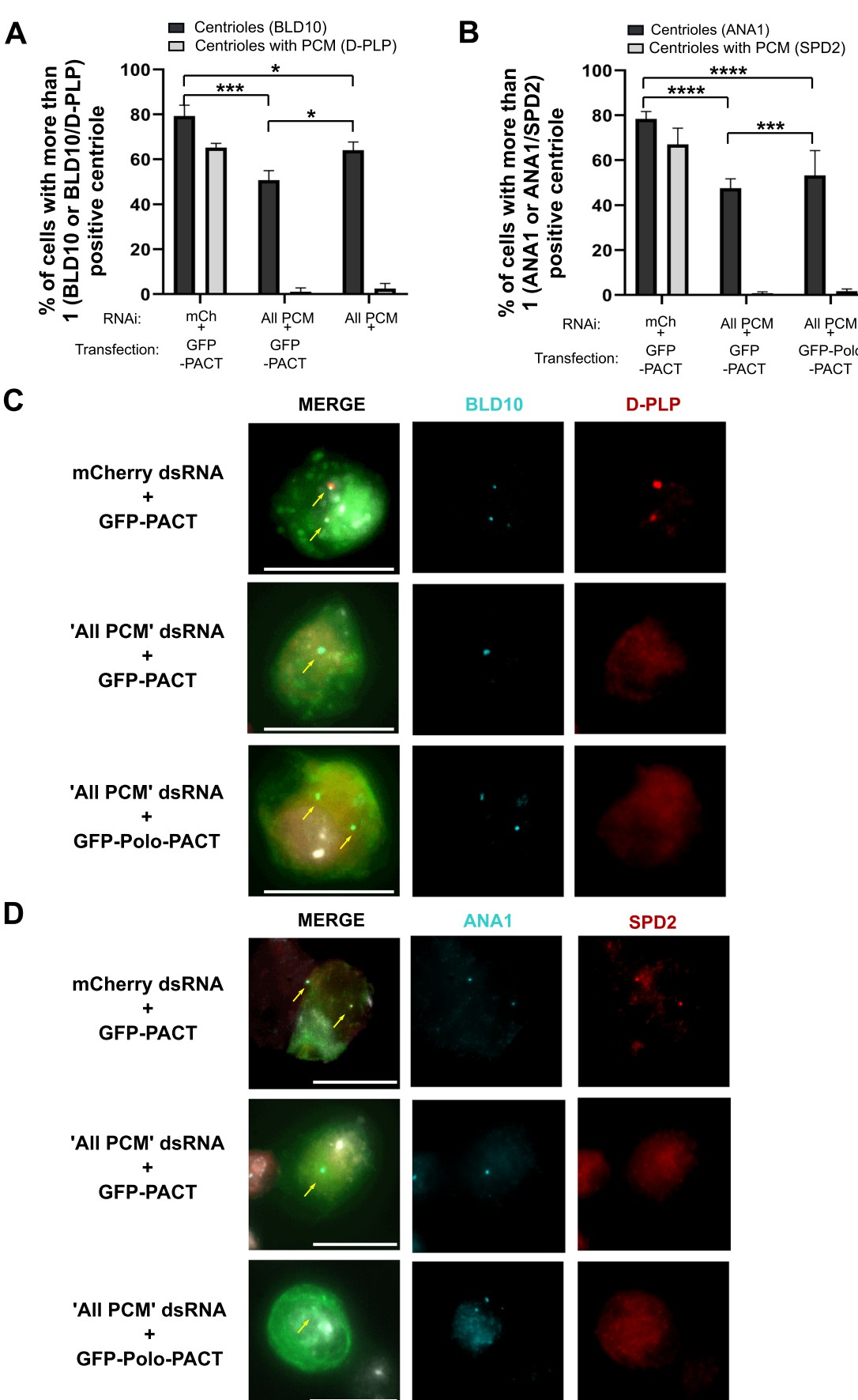

◄ **Figure EV3.   Rescued centrioles by Polo kinase do not retain PCM markers.**

The data in this figure is the same data as in Figure EV2B-D, F-H, analysed in different manner to investigate presence of PCM in centrioles. (A,B) DMEL cells were subjected to dsRNA transfection and treatment with Aph (aphidicolin) and HU (hydroxyurea) at day 0. Cells were depleted of "All PCM" or mCherry (control). After 16 h, cells were transfected (GFP-PACT or GFP-Polo-PACT) in medium with Aph and HU. Cells were harvested and assayed for centriole numbers by immunofluorescence at day 4. Quantification of the percentage of cells with abnormally low numbers of centrioles (i.e. 0-1). For single markers see Fig. EV2. (A) Quantification of the percentage of cells with normal centriole numbers (more than 1 centriole in each cell). Centrioles were quantified by identifying in each cell centrioles positive for BLD10 (dark grey bars), and centrioles which contained PCM, by co-staining for BLD10 and D-PLP (light grey bars). (B) Quantification of the percentage of cells with normal centriole numbers (more than 1 centriole in each cell). Centrioles were quantified by identifying in each cell centrioles positive for ANA1 (dark grey bars), and centrioles which contained PCM, by co-staining for ANA1 and SPD2 (light grey bars). (C) Representative images of A) are shown. All conditions were acquired with the same exposure. Arrows point to centrosomes in the different cells. MERGE shows the merge of the transfection with the GFP constructs, BLD10, D-PLP and DNA. Scale bar, 10 μm. (D) Representative images of B) are shown. All conditions were acquired with the same exposure. Arrows point to centrosomes in the different cells. MERGE shows the merge of the transfection with the GFP constructs, ANA1, SPD2 and DNA. Scale bar, 10 μm. Note data while in mCherry controls most centrioles contain PCM, this is not the case after "All PCM" RNAi, even upon GFP-Polo-PACT expression and increase in centriole number. Indeed, in cells depleted of PCM and transfected with GFP-Polo-PACT only 2,5% and 1,7% of cells contain the PCM markers D-PLP (A) and SPD2 (B), respectively. Data Information: Bars represent the mean ± SEM of three independent biological replicate experiments. For (A,B) "n">80 cells per condition in each experiment. A Two-way ANOVA, with Tukey's multiple comparisons test was used to test statistical significance. For all the statistical tests used in this figure: *$p < 0.05$; ***$p < 0.001$; ****$p < 0.0001$. Source data are available online for this figure.

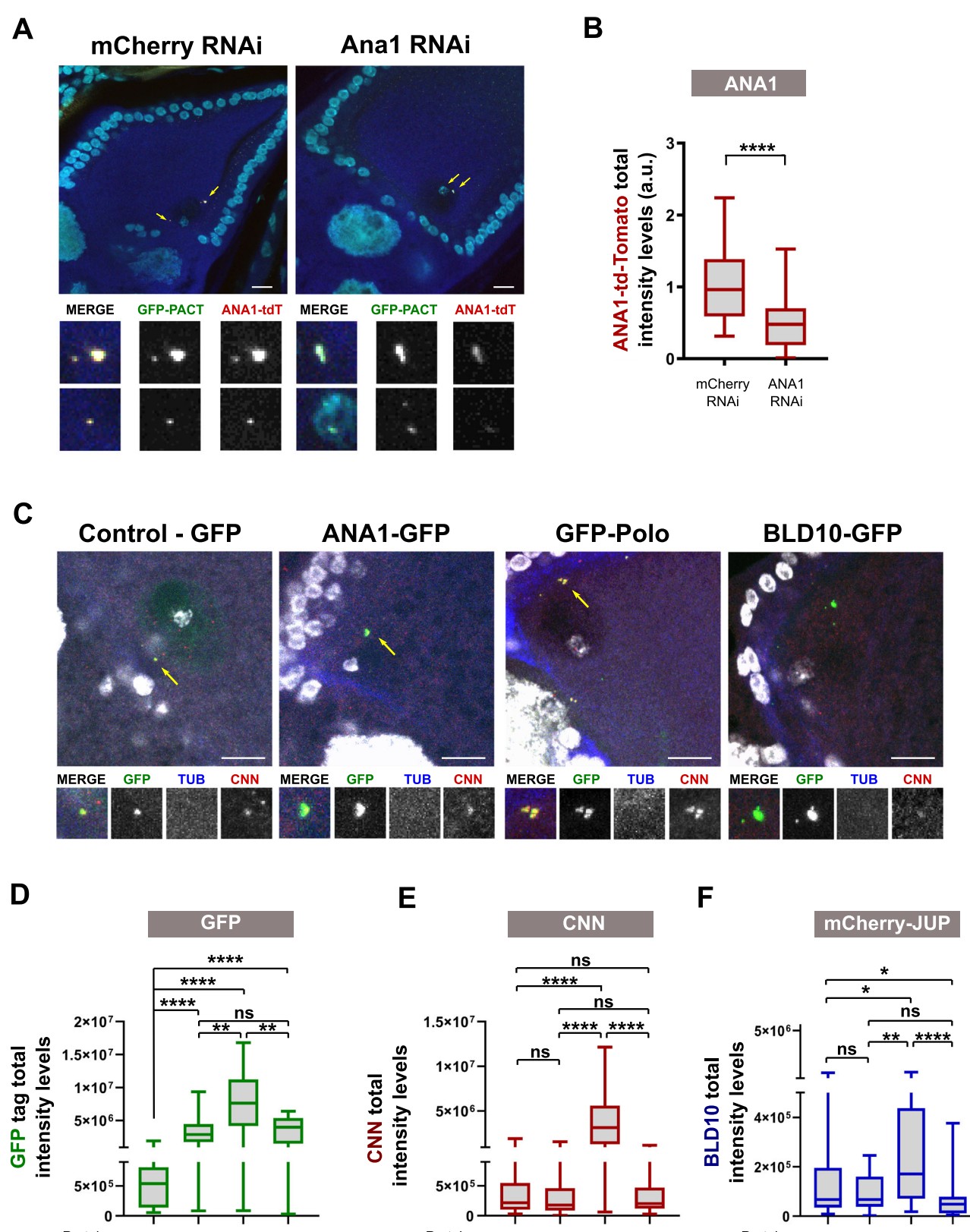

◄

**Figure EV4. Down-regulation of Ana1 by RNAi in oogenesis is not fully efficient and Ectopic tethering of ANA1 to the oocyte centrioles does not increase the PCM protein CNN and microtubule nucleation capacities.**

(A) Depletion of Ana1 by RNAi. *mCherry*-RNAi (control) and *ANA1*-RNAi were expressed in the germ line using a driver that only expresses after stages 3/4 (i.e., after oocyte specification). Expression of both GFP-PACT [under polyubiquitin promoter; PACT is the centriolar targeting domain of PLP] and ANA1-tdTomato (under endogenous promoter) were used as robust centriolar markers. ANA1-tdTomato was used as a readout to address the efficiency of the RNAi in the depletion of ANA1 protein. Enlargements of the indicated areas (yellow arrows) are shown. All images were acquired with the same exposure. Scale bars, 10 μm. (B) Quantification of total intensity levels of ANA1-tdTomato in stage 10 egg chambers for *mCherry*-RNAi and *ANA1*-RNAi expressing oocytes. Box-and-whisker plot (2.5th and 97.5th percentiles) of the total integrated intensity of Ana1-tdTomato. (C) Representative images of the analysis of stage 10 oocytes upon tethering different centrosomal proteins to the oocyte centrioles by expressing a GFP nanobody construct fused to the PACT domain (PACT::vhhGFP4) that targets molecules to the centriole. Enlargements of the indicated areas (with centrioles, yellow arrows) are shown. Note that at this stage the oocyte is supposed to have ~64 clustered centrioles, known to be scattered when Polo-PACT is expressed (Pimenta-Marques et al, 2020). Scale bars, 10 μm. (D) Quantification of the total intensities of GFP, ANA1-GFP, GFP-Polo and Bld10-GFP tethered to centrioles by PACT::vhhGFP4 in stages 10. (E,F) Quantification of the total intensities of (C) CNN and (D) mCherry-Jupiter (a proxi for microtubules as Jupiter is a MAP (Lowe et al, 2014) in stage 10 egg chambers expressing either GFP, ANA1-GFP, GFP-Polo or Bld10-GFP in combination with PACT::vhhGFP4. Note that tethering GFP-Polo leads to an increase in the total levels of CNN and mCherry-Jupiter to the oocyte centrioles in stages 10, which is not observed upon forced localization of ANA1 to the oocyte centrioles. Data Information: For (B), n = 30 for *mCherry*-RNAi, box minimum = 0.3129, box maxima = 2.240, box median = 0.9621, box 25% percentile= 0.5877, box 75% percentile = 1.388. For ANA1-RNAi, n = 31, box minimum = 1.528, box maximum = 1.528 box median = 0.4783, box 25% percentile = 0.1884, box 75% percentile = 0.7012. Box-and-whisker plot (2.5th and 97.5th percentiles) of the total integrated intensity of Ana1-tdTomato. Statistical significance was tested by performing a Unpaired Mann–Whitney test; ****p < 0.0001. For (D,E) Box-and-whisker plots (whiskers extend to the 2.5th and 97.5th percentiles) of the total integrated intensities of the different markers analysed. For (D), n = 31 for GFP + GFPnanoPACT, box minimum = 54,072, box maxima = 1,887,836, box median = 537,902, box 25% percentile = 139,239, box 75% percentile = 812,266; for ANA1-GFP + GFPnanoPCT, n = 30, box minimum = 85,748, box maxima = 9,350,192, box median = 2,850,538, box 25% percentile = 1,727,802, box 75% percentile = 4,443,137; for GFP-Polo + GFPnanoPACT, n = 26, box minimum = 87,484, box maxima = 16,806,005, box median = 7,631,831, box 25% percentile = 4,156,380, box 75% percentile = 11,234,152. for BLD10-GFP + GFPnanoPACT, n = 26, box minimum= 30,420, box maxima = 6,405,887, box median = 3,945,207, box 25% percentile = 1,477,427, box 75% percentile = 5,383,259. For (E), n = 31 for GFP + GFPnanoPACT, box minimum = 30,687, box maxima = 1,868,551, box median = 216,522, box 25% percentile = 103,936, box 75% percentile = 546,986. for ANA1-GFP + GFPnanoPACT, n = 30, box minimum = 8052, box maximum = 1,525,037, box median = 180,856, box 25% percentile = 86600, box 75% percentile = 460,606; for GFP-Polo + GFPnanoPACT, n = 26, box minimum = 786,335, box maxima = 49,501,547, box median = 6,598,892, box 25% percentile = 3,098,434, box 75% percentile = 12,625,516; for BLD10-GFP + GFPnanoPACT, n = 26, box minimum = 28,468, box maxima = 1,132,352, box median = 204,727, box 25% percentile = 120,895, box 75% percentile = 472,851. For (F), n = 31 for GFP + GFPnanoPACT, box minimum = 8875, box maxima = 1,015,035, box median = 66,987, box 25% percentile = 35,287, box 75% percentile = 195,555. for ANA1-GFP + GFPnanoPACT, n = 30, box minimum = 2516, box maximum = 245,586, box median = 67,146, box 25% percentile = 38,265, box 75% percentile = 160,067; for GFP-Polo + GFPnanoPACT, n = 26, box minimum = 18,307, box maxima = 1,071,604, box median = 170,734, box 25% percentile = 71,301, box 75% percentile = 437,557; for BLD10-GFP + GFPnanoPACT, n = 26, box minimum = 4708, box maxima = 376,286, box median = 48,386, box 25% percentile = 12,032, box 75% percentile = 78,575. 3 independent biological replicates were performed for each condition. significance was determined by performing a bimodal regression test. For statistical tests used in this figure: *p < 0.05; **p < 0.001; ****p < 0.0001; ns, not statistically significant. Source data are available online for this figure.

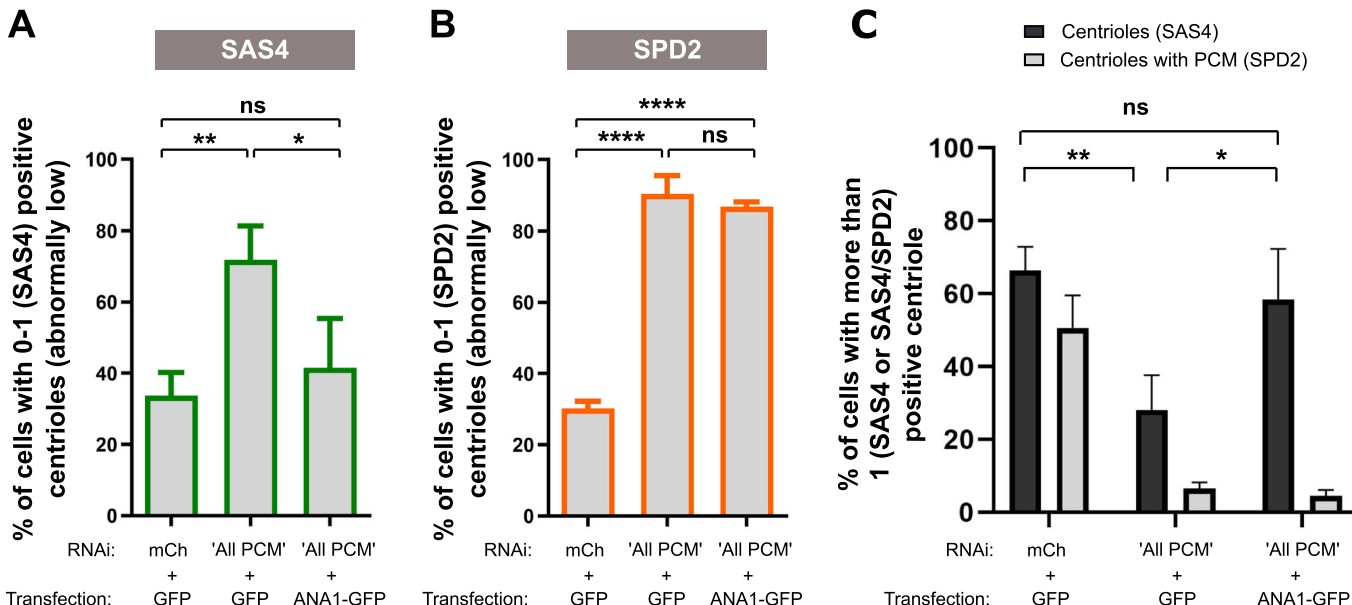

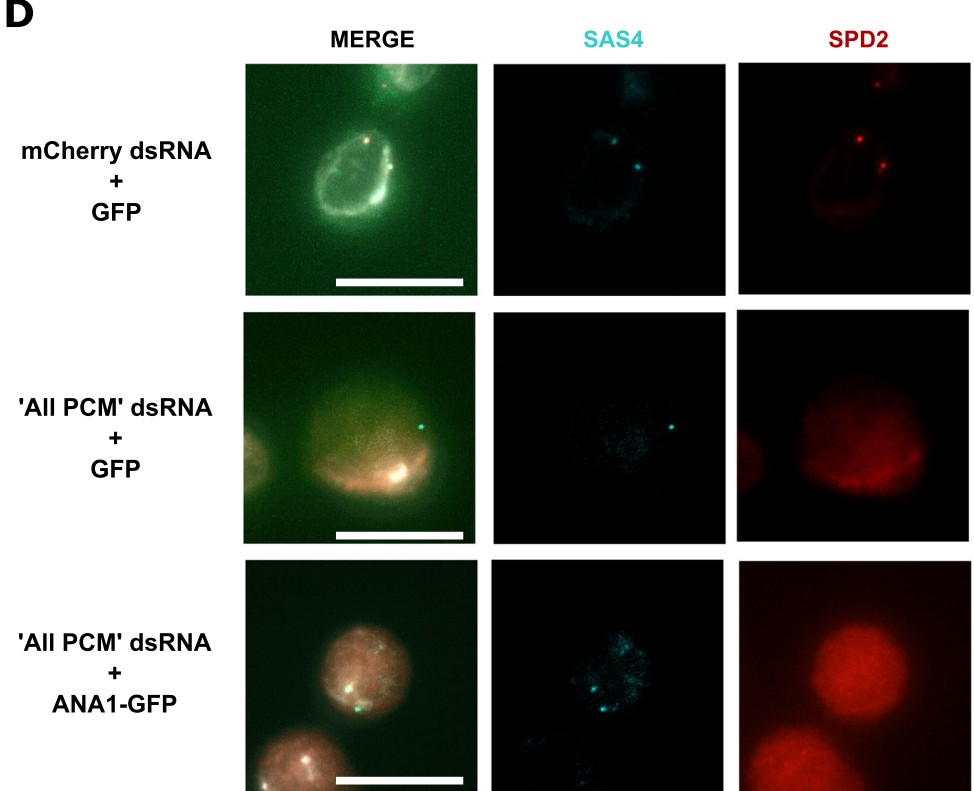

◀ **Figure EV5. ANA1 rescues the loss of centrioles induced by PCM depletion.**

(A–D) DMEL cells were subjected to dsRNA transfection and treatment with Aph (aphidicolin) and HU (hydroxyurea) at day 0. Cells were depleted of "ALL PCM" or mCherry (control). After 16 h, cells were transfected (GFP or ANA1-GFP) in medium with Aph and HU. Cells were harvested and assayed for centriole numbers by immunofluorescence at day 4. Centrioles were identified by considering the positive staining in each cell for the centriolar wall protein SAS4 and presence of PCM was investigated using the marker, SPD2. (A) Quantification of the percentage of cells with abnormally low numbers of centrioles (i.e. 0-1) labelled by SAS4. (B) Quantification of the percentage of cells with abnormally low numbers of foci with PCM marker SPD2 (i.e. 0-1). (C) Quantification of the percentage of cells with more than 1 centriole in each cell. Centrioles were quantified by identifying in each cell centrioles positive for SAS4 (dark grey bars), and centrioles which contained PCM, by co-staining for SAS4 and SPD2 (light grey bars). (D) Representative images of (A–C) are shown. All conditions were acquired with the same exposure. Arrows point to centrosomes in the different cells. MERGE shows the merge of the transfection with the GFP or ANA1-GFP and SAS4, SPD2 and DNA. Scale bar, 10 μm. Data information: Bars represent the mean ± SEM of three independent biological replicates ("n" between 47-100 cells per replicate, per condition). Two-way ANOVA, with Tukey's multiple comparisons test. For all the statistical tests used in this figure: $*p < 0.05$; $**p < 0.001$; $****p < 0.0001$; ns, not statistically significant. Source data are available online for this figure.

