## [Peer Review File · EMBO Reports]

Ana1/CEP295 is an essential player in the centrosome maintenance program regulated by Polo kinase and the PCM

Ana Pimenta-Marques, Tania Perestrelo, Patricia Reis-Rodrigues, Paulo Duarte, Mariana Lince-Faria, Monica Bettencourt-Dias, and Ana Ferreira-Silva

DOI: [10.15252/embr.202255667](https://doi.org/10.15252/embr.202255667)

Corresponding author(s): Ana Pimenta-Marques (ana.pmarques@nms.unl.pt), Monica Bettencourt-Dias (mdias@igc.gulbenkian.pt)

Review Timeline:

Transfer from Review Commons:	28th Jun 22
Editorial Decision:	5th Jul 22
Appeal Received:	12th Jul 22
Editorial Decision:	19th Jul 22
Revision Received:	23rd Aug 23
Editorial Decision:	25th Oct 23
Revision Received:	14th Nov 23
Accepted:	22nd Nov 23

Editor: Esther Schnapp

Transaction Report: This manuscript was transferred to EMBO reports following peer review at Review Commons.

Review
COMMONS

Review #1

1. Evidence, reproducibility and clarity:

Evidence, reproducibility and clarity (Required)

Summary:

In this manuscript Pimenta-Marques build on their previous work addressing how centrioles are stabilized and maintained or destabilized and disassembled, depending on the cell type and developmental context. Using *Drosophila* cell culture and oogenesis as an *in vivo* model for centriole destabilization, they identify the centriole wall protein Ana1 as a central player in centriole stability. Its presence is required for the maintenance even of mature centrioles, suggesting that there is continued turnover of centriole structural components.

Major comments:

1. The experiments and results are very well described and most of the conclusions are supported by the data. One aspect needs clarification though. It is not clear to this reviewer how the authors envision the regulation and mechanism by which Ana1 functions in centriole stability. The data suggest that it can stabilize centrioles independent of PCM (Fig. 3B, 5B), yet the authors claim in the results and discussion that it functions downstream of PCM. As presented, this does not make sense. I would argue the opposite, it may function upstream or in parallel to the PCM. Related to the above, the last sentence of the intro states: "Finally, we found that both Polo and the PCM require ANA1 to promote centriole structural integrity." This is shown for Polo, but where is the data showing that PCM requires ANA1 for promoting centriole stability?
2. I have a concern regarding the number *n* used for statistics in the quantifications. In many cases it seems that the number *n* of cells etc. was used (e.g. $n > 100$ cells) rather than the number of experiments (e.g. $n = 3$). The statistics should measure variability between experimental repetitions, not between cells etc. If statistics were indeed not done on experiments and would have to be changed, some of the observed effects may not be statistically significant and would require additional experimental replicates, which would increase the time needed for revision.

Minor comments:

1. I would advice the authors to improve the presentation of the figures. In particular

the labels are in many cases very small and difficult to read. Readability is also reduced by the use of bold font in the labels and a mix of various font sizes within single figure panels.

2. The result section could be shortened/become more readable by moving several paragraphs to the intro or discussion.

3. The introduction is quite long and some parts read more like an introduction of a review on the topic.

2. Significance:

Significance (Required)

This is a nice, focused study on the requirements underlying centriole stability and maintenance. The first part identifies the cartwheel, the centriole wall, and the PCM as important for centriole maintenance. The remaining parts identify and focus on the essential role of ANA1 in this process. This is an important finding, since the mechanisms underlying centriole stability and maintenance are poorly understood, yet highly relevant. Some cell types inactivate and/or disassemble centrioles during differentiation and this is likely important to their function. Providing more mechanistic insight, for example, regarding the relationship between ANA1 and PCM recruitment or the regulation of ANA1's centriole function by Polo, would have further strengthened the study.

The audience interested in this work will be cell and developmental biologists. My expertise is in centrosome biology and microtubule organization.

****Referees cross-commenting****

I agree with the additional points raised by the other reviewers. I still think that overall the paper is fine and most things could be addressed in a reasonable time frame. The work does not provide much mechanism though. In this regard, the confusing placement of ANA1 downstream of PCM, would be the only mechanistic aspect, and it seems the authors got it wrong, at least based on the provided data. Here, additional experiments could elucidate these relationships further, but if this is not the goal, text changes could also address this and it would remain a smaller, more focused study.

3. How much time do you estimate the authors will need to complete the suggested revisions:

Estimated time to Complete Revisions (Required)

(Decision Recommendation)

Less than 1 month

4. Review Commons values the work of reviewers and encourages them to get credit for their work. Select 'Yes' below to register your reviewing activity at Publons; note that the content of your review will not be visible on Publons.

Reviewer Publons

Yes

Review #2

1. Evidence, reproducibility and clarity:

Evidence, reproducibility and clarity (Required)

In this paper, the authors show that the turnover of centriole components is necessary for proper centriole maintenance within *Drosophila* cultured cells (during prologued cell cycle arrest) and within *Drosophila* oocytes, where centrioles are normally degraded prior to fertilisation. They highlight Ana1 as an important player in centriole maintenance. The authors begin with a candidate screen to identify core centriole proteins that are required to properly maintain centrioles. They then focus on Ana1, given that its depletion had the strongest effect, and show that its depletion leads to a reduction in the levels of centriole components in *Drosophila* oocytes. They show that the previously observed ability of centriole-targeted Polo to counteract centriole loss depends at least in part on Ana1 and that targeting Ana1 to centrioles also counteracts centriole loss. The authors conclude that Ana1 is a component of the PCM-promoted centriole integrity pathway.

****Major comments****

1. The authors say that Plk4 depletion does not lead to centriole loss, but there are significant differences in centriole number between the control and Plk4 depletion cells in Fig 1F and S1D. Please comment.

2. One of the main results is that depletion of centriole components leads to a reduction in centrosome numbers when measured 8 days after S-phase arrest. I wonder whether a restriction of centriole duplication could add to this effect? Any cells that were in G2 or M phase when the drugs were added would presumably progress into the following S-phase and duplicate their inherited centrioles, but not if centriole duplication proteins had been depleted. It's true that Plk4 depletion leads to a relatively mild centriole loss phenotype, but can the authors be sure that this is not due to variations in the efficiency of different RNAi constructs? Perhaps the authors can show that Plk4 depletion efficiently prevents centriole duplication under otherwise normal conditions.

3. The authors show that Ana1 depletion has the strongest effect, but this could in theory be due to differences in RNAi efficiency. I don't expect the authors to show the efficiency of all RNAi constructs, but they could state in the text that this is a caveat e.g. "...although we cannot rule out the possibility that differences in RNAi efficiency lead to the observed differences in severity of phenotype..."

4. A key conclusion is that core centriole components turnover to some extent and that the incorporation of new molecules is necessary for centriole maintenance. This is a very interesting and important point and so it would be nice to have more direct data to support it. This could be done in different ways, including transfecting fluorescently tagged centriole components after S-phase arrest and showing that some molecules become incorporated into the centrioles, or by performing FRAP experiments. Of course, it is possible that the turnover is so low that the incorporated fluorescent molecules cannot be detected...

5. The authors show that depletion of Ana1 from oocytes leads to a reduction in the intensity of centriole markers. They do not measure centrosome numbers, as the centrosomes cluster too tightly. The authors therefore can't be certain that Ana1 depletion leads to a reduction in centrosome numbers. The authors could show this by inhibiting centrosome clustering while depleting Ana1. There is a recent BioRxiv paper showing that centrosome clustering can be inhibited by depletion of Kinesin-1.

6. In Figure 3B the authors show that expression of GFP-Polo-PACT partially rescues the effect of "all PCM" depletion, but this seems strange given that Polo's role is presumably to recruit PCM (which has been depleted). Can the authors comment? Also, it would make sense to test whether GFP-Polo-PACT can rescue centriole loss after the depletion of Ana1 alone (not Ana1 and all PCM). If Ana1 has a role in recruiting Polo (either directly or indirectly), which has been shown previously in mitotic cells, then there should be a rescue to some extent.

7. In Fig4A,C, the authors say that γ -tubulin levels at centrosomes increase when GFP-Polo is forced onto the centrosomes - the graph seems to show a big increase, but the pictures do not...? Are the authors measuring total levels at all centrosomes? If so, I think they should be measuring the average at individual centrosomes. Also, why is the level of GFP alone not much higher when expressed with GFPnanoPACT (Fig

1B)? Presumably GFP should be recruited to the centrosomes by GFPnanoPACT.

8. The authors show that tethering Ana1-GFP to the centrioles counteracts centriole loss in oocytes (Fig4G). They say that the centrosomes are most likely inactive because they don't recruit PCM, but they have only looked at γ -tubulin, which is a downstream component of the PCM. I think it is important to check whether Polo is recruited, given that tethering Polo to centrioles also counteracts centriole loss and that a recent paper showed that Ana1 has a role in recruiting Polo to centrosomes (Alvarez-Rodigo et al., 2021). The authors also say that these centrosomes do not organise microtubules but do not show the data.

9. The authors propose that Ana1 is downstream of the PCM, and so over-expressing Ana1 should at least partially rescue centriole loss after PCM depletion. But I don't really agree with this. If Ana1 relies on the PCM then how would its overexpression manage to rescue the phenotype in the absence of the PCM? The finding that over-expressing Ana1 partially rescues centriole loss may instead suggest that Ana1 is either upstream of the PCM or part of an independent pathway. Indeed, the authors show that depletion of both the PCM and Ana1 has a stronger effect than either depletions individually - this is indicative of two independent pathways.

****Minor comments****

1. When the authors say that the centriole wall and cartwheel components are "dynamic" I think that they need to make it clear that this "dynamicity" is not very fast. Using the term dynamic tends to suggest rapid turnover (like in the PCM). Perhaps the authors could use the term "slow exchange" or something similar.

2. The authors currently use a 0 or 1 centriole categorisation - it would be nice to see the breakdown of what percentage of cells have 0, 1, 2, or >2 centrioles, perhaps in a supplementary excel file.

2. Significance:

Significance (Required)

How centrioles are eliminated in certain cells is an interesting question and the data presented is also relevant to understanding centriole biology in general, because it seems that some apparently very stable structural proteins actually turnover. It is widely known that PCM proteins turnover relatively quickly, but core centriole proteins are considered to be stably incorporated. The data will therefore raise interest in the centrosome field. I do, however, feel that for the authors to make this point more strongly it would be good to show this more directly. Overall, this is a very interesting paper that is well written. The data is well presented and supports the

conclusions that centriole components turnover and that Ana1 is involved in maintaining centriole integrity.

3. How much time do you estimate the authors will need to complete the suggested revisions:

Estimated time to Complete Revisions (Required)

(Decision Recommendation)

Between 1 and 3 months

4. Review Commons values the work of reviewers and encourages them to get credit for their work. Select 'Yes' below to register your reviewing activity at Publons; note that the content of your review will not be visible on Publons.

Reviewer Publons

Yes

Review #3

1. Evidence, reproducibility and clarity:

Evidence, reproducibility and clarity (Required)

The team explores a previously developed "centriole stability assay" to monitor the disappearance of centrioles after RNAi-dependent depletion of various centrosome components. Important roles in centriole stability are found for the PCM and for cartwheel proteins, in addition to proteins of the centriole wall. The remainder of the study focuses on the centriole wall protein ANA1: induced degradation of ANA1 during *Drosophila* oogenesis strongly reduces the PCM and other centriole markers, and ANA1-dependent defects cannot be prevented by GFP-Polo-PACT, which is otherwise known to protect from the loss of PCM. In complementary experiments,

forced targeting of ANA1 to the PCM, or overexpression of AN1 protects centriole integrity.

2. Significance:

Significance (Required)

The study shows that ANA1 is important for the integrity of centrosomes. Generally, this work is well executed and correctly controlled. The novelty of the results is somewhat limited, since a role of ANA1 in centrosome assembly has already been reported by others. The present work emphasizes aspects of centrosome protein maintenance, but doesn't provide mechanistic details of protein turnover. The manuscript should be of interest to the scientific community working on the centrosome.

****Other comments:****

I wonder whether the results from the centrosome maintenance experiment with GFP-Polo-PACT (Figure 3) are really very telling: since PCM and other centriole markers are lost upon ANA1-depletion, GFP-Polo-PACT cannot target to the PCM, and it is therefore unsurprising that GFP-Polo-PACT fails to provide its protective effect. Would expression of GFP-Polo-PACT prior to addition of ANA1-RNAi have a protective effect?

****Minor point:****

Figure 1H: it is unclear to me how centrioles are identified with the BLD10 marker in samples that have been treated with BLD10 RNAi.

****Referees cross-commenting****

I agree very much with reviewers 1 and 2 that a role of ANA1 "downstream" of the PCM is not really supported by the data.

I also think that all other points raised in the reviews merit attention.

3. How much time do you estimate the authors will need to complete the suggested revisions:

Estimated time to Complete Revisions (Required)

(Decision Recommendation)

Between 1 and 3 months

4. Review Commons values the work of reviewers and encourages them to get credit for their work. Select 'Yes' below to register your reviewing activity at Publons; note that the content of your review will not be visible on Publons.

Reviewer Publons

Yes

Revision Plan

Manuscript number: #RC-2022-01412

Corresponding author(s): Ana, Pimenta-Marques; Mónica, Bettencourt-Dias

[The “revision plan” should delineate the revisions that authors intend to carry out in response to the points raised by the referees. It also provides the authors with the opportunity to explain their view of the paper and of the referee reports.]

The document is important for the editors of affiliate journals when they make a first decision on the transferred manuscript. It will also be useful to readers of the reprint and help them to obtain a balanced view of the paper.

*If you wish to submit a full revision, please use our "Full Revision" template. **It is important to use the appropriate template to clearly inform the editors of your intentions.**]*

1. General Statements [optional]

This section is optional. Insert here any general statements you wish to make about the goal of the study or about the reviews.

Dear editors of Review Commons, we present here a revision plan of our manuscript, which we would like to be considered as a publication of EMBO reports. We presented this manuscript recently at the EMBO microtubule meeting and discussed it and its reviews with editor Ieva Gaillite. She discussed this with colleagues from EMBO reports, and suggested we submit it there.

We found that contrary to the wide perception that the centrosome is a “static” structure, the centriole actually requires the replenishment of essential components for the maintenance of its integrity. Moreover, our data suggests that the Pericentriolar Material, a multiprotein cloud surrounding the centriole that is known for its role in microtubule nucleation, plays a new key role in the replenishment of such centriole components. Our work sheds new light into the function of the pericentriolar matrix and more generally on how cellular structures are maintained in living cells.

We discuss here how we intend to reply to the referee reports. We look forward to hearing from you whether we should submit a full revised version to your journal.

2. Description of the planned revisions

Insert here a point-by-point reply that explains what revisions, additional experimentations and analyses are planned to address the points raised by the referees.

RESPONSES to REVIEWERS:

(the responses from the authors are shown in blue)

Reviewer #1 (Evidence, reproducibility and clarity (Required)):

Revision Plan

Summary:

In this manuscript Pimenta-Marques build on their previous work addressing how centrioles are stabilized and maintained or destabilized and disassembled, depending on the cell type and developmental context. Using *Drosophila* cell culture and oogenesis as an in vivo model for centriole destabilization, they identify the centriole wall protein Ana1 as a central player in centriole stability. Its presence is required for the maintenance even of mature centrioles, suggesting that there is continued turnover of centriole structural components.

Major comments:

1) The experiments and results are very well described and most of the conclusions are supported by the data. One aspect needs clarification though. It is not clear to this reviewer how the authors envision the regulation and mechanism by which Ana1 functions in centriole stability. The data suggest that it can stabilize centrioles independent of PCM (Fig. 3B, 5B), yet the authors claim in the results and discussion that it functions downstream of PCM. As presented, this does not make sense. I would argue the opposite, it may function upstream or in parallel to the PCM.

We thank the reviewer for his/her comments. During centriole assembly, it is indeed the case that Ana1 helps recruiting the PCM in a process called centriole to centrosome conversion. Here, in studying centriole stability we see that:

a) Targeting ANA1 to centrosomes in oogenesis by use of the GFPnanoPACT (Figure 4), did not lead to any significant change in the levels of gamma-tubulin on centrosomes (figure 4,C). b) In figure 5 overexpression of Ana1 can rescue centriole loss caused by the depletion of PCM ("ALL PCM").

Overall our results suggest that the PCM is important to maintain ANA1 on the centrosome, providing structural integrity. We will revise our manuscript in order to clearly state this hypothesis and to ensure it is clear how we envision the regulation mechanism by which ANA1 functions in centriole stability. If the reviewer considers it useful we can add in figure 5 a working model summarizing the main conclusions of this work.

Related to the above, the last sentence of the intro states: "Finally, we found that both Polo and the PCM require ANA1 to promote centriole structural integrity." This is shown for Polo, but where is the data showing that PCM requires ANA1 for promoting centriole stability?

As previously mentioned, overexpressing Ana1 in S-phase arrested cells depleted of the PCM rescues the loss of centrioles observed upon PCM depletion (figure 5). Therefore these data suggest that the PCM requires ANA1 to perform its role in maintaining centrosome stability.

2) I have a concern regarding the number n used for statistics in the quantifications. In many cases it seems that the number n of cells etc. was used (e.g. n>100 cells) rather than the number of experiments (e.g. n=3). The statistics should measure variability between experimental repetitions, not between cells etc. If statistics were indeed not done on experiments and would have to be changed, some of the observed effects may not be statistically significant and would require additional experimental replicates, which would increase the time needed for revision.

We will provide more details in the revised version of the manuscript in order to make it clear how many experiments were performed and the "n" used. In any case, we would like to clarify that for

Revision Plan

all the experiments we always performed 3 independent replicates for each condition. For the candidate screen in s-phase arrested cells (figure 1 and supplementary figure 1) we analyzed the fraction of cells arrested by a logistic regression in which one of the factors was the replicate, in addition to the treatment. This provides an estimate of the effect of the treatment, controlling for variability between replicates. In other words, we separate the variation that can be attributed to replicate variation from the variation that can be attributed to the treatment.

In Figures 3 and 5, statistics were performed between the 3 independent experiments. The “N” refers to the number of cells within each independent experiment. We will change the text in order to make it clear.

Because in oogenesis we do not have such large datasets, it might be misleading to perform statistics between different experiments, as the different “ns” within the same condition might vary. Therefore for these experiments we used a Fisher test, to test if the different replicates for each condition were different. Since they were not, and consistency in data was observed between the 3 independent experiments, we pulled the data together. Therefore, the “ns” mentioned in the oogenesis experiments refer to the total “n” after polling together the replicates of each experimental condition. We will explain better how we treated all datasets in the revised version of the manuscript.

Minor comments:

1) I would advise the authors to improve the presentation of the figures. In particular the labels are in many cases very small and difficult to read. Readability is also reduced by the use of bold font in the labels and a mix of various font sizes within single figure panels.

We thank the reviewer for that comment and we will improve the figures in order to make them easier to follow and read.

2) The result section could be shortened/become more readable by moving several paragraphs to the intro or discussion.

We thank the reviewer for that comment. We will revise the text and shorten it to adapt the text to the format of EMBO reports to which we propose to submit this work.

3) The introduction is quite long and some parts read more like an introduction of a review on the topic.

We thank the reviewer for that comment. We will revise the text and shorten it, which will also include cutting the introduction and making it more focused on the topic under study.

Reviewer #1 (Significance (Required)):

This is a nice, focused study on the requirements underlying centriole stability and maintenance. The first part identifies the cartwheel, the centriole wall, and the PCM as important for centriole maintenance. The remaining parts identify and focus on the essential role of ANA1 in this process. This is an important finding, since the mechanisms underlying centriole stability and maintenance are poorly understood, yet highly relevant. Some cell types inactivate and/or disassemble

Revision Plan

centrioles during differentiation and this is likely important to their function. Providing more mechanistic insight, for example, regarding the relationship between ANA1 and PCM recruitment or the regulation of ANA1's centriole function by Polo, would have further strengthened the study. The audience interested in this work will be cell and developmental biologists.

My expertise is in centrosome biology and microtubule organization.

We are glad this reviewer thinks this is an important study and that it furthers our understanding on the mechanisms underlying centriole stability.

Referees cross-commenting

I agree with the additional points raised by the other reviewers. I still think that overall the paper is fine and most things could be addressed in a reasonable time frame. The work does not provide much mechanism though. In this regard, the confusing placement of ANA1 downstream of PCM, would be the only mechanistic aspect, and it seems the authors got it wrong, at least based on the provided data. Here, additional experiments could elucidate these relationships further, but if this is not the goal, text changes could also address this and it would remain a smaller, more focused study.

As we discussed in our first answer to this reviewer, we believe that adding a figure with a working model will help to interpret our data.

Reviewer #2 (Evidence, reproducibility and clarity (Required)):

In this paper, the authors show that the turnover of centriole components is necessary for proper centriole maintenance within *Drosophila* cultured cells (during prologued cell cycle arrest) and within *Drosophila* oocytes, where centrioles are normally degraded prior to fertilisation. They highlight Ana1 as an important player in centriole maintenance. The authors begin with a candidate screen to identify core centriole proteins that are required to properly maintain centrioles. They then focus on Ana1, given that its depletion had the strongest effect, and show that its depletion leads to a reduction in the levels of centriole components in *Drosophila* oocytes. They show that the previously observed ability of centriole-targeted Polo to counteract centriole loss depends at least in part on Ana1 and that targeting Ana1 to centrioles also counteracts centriole loss. The authors conclude that Ana1 is a component of the PCM-promoted centriole integrity pathway.

Major comments

1) The authors say that Plk4 depletion does not lead to centriole loss, but there are significant differences in centriole number between the control and Plk4 depletion cells in Fig 1F and S1D. Please comment.

We thank the reviewer for the comment. As the reviewer noticed, indeed PLK4 does lead to some degree of centriole loss. However, by comparing the effect of PLK4 depletion with the one of the cartwheel and centriolar components, it does not lead to a pronounced increase in the percentage of cells with 0-1 centrioles, judged by the different markers. The modest effect of PLK4 depletion on centrosome numbers may result from an indirect effect on centrosome stability, resulting from its ability to interact with some of the proteins that have stronger phenotypes in our assay, such

Revision Plan

as CPAP/SAS4 (Moyer & Holland, 2019), or components of the PCM, such as CEP152/Asl (Bahtz et al., 2012; Dzhindzhev et al., 2010). This will be further explained in the modified text.

2) One of the main results is that depletion of centriole components leads to a reduction in centrosome numbers when measured 8 days after S-phase arrest. I wonder whether a restriction of centriole duplication could add to this effect? Any cells that were in G2 or M phase when the drugs were added would presumably progress into the following S-phase and duplicate their inherited centrioles, but not if centriole duplication proteins had been depleted. It's true that Plk4 depletion leads to a relatively mild centriole loss phenotype, but can the authors be sure that this is not due to variations in the efficiency of different RNAi constructs? Perhaps the authors can show that Plk4 depletion efficiently prevents centriole duplication under otherwise normal conditions.

We thank the reviewer for his comment and understand his concern. Nonetheless, in our experimental assay, which was used already in Pimenta-Marques et al. (2016), both dsRNA for protein depletion as well as the drugs used to arrest cells in S phase, were administered simultaneously. Given that the dsRNA takes at least 24 hrs to have an effect, and the cell cycle of S2 cells is approximately 24 hours, we don't expect that the targeted proteins would be depleted before the drugs arrest the cells. Moreover, it is important to point out that in contrast to some cultured Human cell lines (Balczon et al., 1995; Stucke, Silljé, Arnaud, & Nigg, 2002), *Drosophila D.mel* cells do not reduplicate their centrioles upon induced S phase arrest (Dzhindzhev et al., 2014). Therefore, upon treatment with HU and APH, the number of centrioles remains unchanged which allows us to address the effect of the different depletions regarding centriole stability. Also, it is well established, depletion or inhibition of PLK4's kinase activity in proliferating cultured cells prevents centriole duplication. Therefore, as the reviewer pointed, the fact that we do not have a strong phenotype regarding centriole loss upon PLK4 depletion, does indeed reinforce that centriole duplication is likely to not be strongly impaired in our assay.

3) The authors show that Ana1 depletion has the strongest effect, but this could in theory be due to differences in RNAi efficiency. I don't expect the authors to show the efficiency of all RNAi constructs, but they could state in the text that this is a caveat e.g. "...although we cannot rule out the possibility that differences in RNAi efficiency lead to the observed differences in severity of phenotype..."

We thank the reviewer for the comment and will include that suggestion.

4) A key conclusion is that core centriole components turnover to some extent and that the incorporation of new molecules is necessary for centriole maintenance. This is a very interesting and important point and so it would be nice to have more direct data to support it. This could be done in different ways, including transfecting fluorescently tagged centriole components after S-phase arrest and showing that some molecules become incorporated into the centrioles, or by performing FRAP experiments. Of course, it is possible that the turnover is so low that the incorporated fluorescent molecules cannot be detected...

We are glad that the reviewer agrees this is an interesting point. We thank the reviewer for this suggestion. However, performing FRAP in our experimental setup, as discussed with EMBO editor, is very difficult. In *Drosophila* cultured cells, such experiments would require arresting cells in S-phase for 8 days, while transfecting with two constructs (to identify centrosome and to FRAP candidate protein), which is complicated. We have attempted to perform such experiments in

Revision Plan

oogenesis, however, centrioles are very dynamic and move, and also, oocytes are thick, which makes this experiment technically very difficult.

5) The authors show that depletion of Ana1 from oocytes leads to a reduction in the intensity of centriole markers. They do not measure centrosome numbers, as the centrosomes cluster too tightly. The authors therefore can't be certain that Ana1 depletion leads to a reduction in centrosome numbers. The authors could show this by inhibiting centrosome clustering while depleting Ana1. There is a recent BioRxiv paper showing that centrosome clustering can be inhibited by depletion of Kinesin-1.

We thank the reviewer for this comment, however, although we cannot individualize centrosomes, our data strongly suggests that ANA1 depletion does indeed lead to a loss of the whole centrosomal structure. Our results show that upon targeting ANA1 for degradation by using the GFPdegrad system both PCM components (D-PLP and gamma-tubulin) as well as the centriolar protein CP110 are reduced from the centrosomal foci. This suggests that the centrosome as a hole is being prematurely lost rather than the specific loss of components whose maintenance could directly depend on ANA1. Moreover these results are consistent with what we observe in cultured cells, where the percentage of cells with 0-1 centrioles increases dramatically upon ANA1 RNAi treatment. Therefore, the data strongly suggests that the degradation of ANA1 in oogenesis is most likely leading to centriole loss. The suggested experiment would be very time consuming, with no guarantees of generating healthy flies suitable for such experiments, given the large number of insertions needed in the genome.

6) In Figure 3B the authors show that expression of GFP-Polo-PACT partially rescues the effect of "all PCM" depletion, but this seems strange given that Polo's role is presumably to recruit PCM (which has been depleted). Can the authors comment?

As previously pointed out by the reviewer, the dsRNAi constructs are likely to show variability in their efficiencies to deplete a given protein, and they also do not allow the total depletion of the targeted protein. Taking this in account, we expect that targeting Polo to the centrosome (expression of GFP-Polo-PACT) leads to the recruitment/maintenance of the remaining PCM which was not depleted by the dsRNAi, explaining the observed rescue.

Also, it would make sense to test whether GFP-Polo-PACT can rescue centriole loss after the depletion of Ana1 alone (not Ana1 and all PCM). If Ana1 has a role in recruiting Polo (either directly or indirectly), which has been shown previously in mitotic cells, then there should be a rescue to some extent.

We have not proposed that Ana1 recruits Polo in this maintenance situation. In fact, while ANA1 is involved in the recruitment of Polo during centrosome maturation (Alvarez-Rodrigo, Wainman, Saurya, & Raff, 2021), the expression of the ANA1 mutant transgene which abolishes the recruitment of Polo did not affect centrosome duplication nor their maintenance (Alvarez-Rodrigo et al., 2021). Therefore, taking these data into account, we do not favor a scenario where the recruitment of Polo by ANA1 is critical for the maintenance of the integrity of the centrosomal structure.

7) In Fig4A,C, the authors say that γ -tubulin levels at centrosomes increase when GFP-Polo is

Revision Plan

forced onto the centrosomes - the graph seems to show a big increase, but the pictures do not...? Are the authors measuring total levels at all centrosomes?

We thank the reviewer for his comments. We will revise the manuscript to clearly explain how gamma-tubulin levels were quantified. We will also add a figure which better demonstrates our observations. In the different conditions tested, which includes the forced targeting of GFP-Polo to the centrosomes, the sum intensity levels of gamma-tubulin on centrosomes were measured. Since we want to evaluate the quantity of protein still present at centrosomes, we believe that measuring sum intensity is closer to the amount of protein present than average intensity. In addition, as previously explained, it is not feasible to perform measurements in individual centrosomes (Pimenta-Marques et al., 2016). Thus, centrosomes were identified on the basis of the colocalization between the BLD10 antibody staining and the GFP signal of each condition. We have previously found that in these stages of oogenesis (stages 10) centriolar components are still present (Pimenta-Marques et al., 2016) with centriolar loss being more pronounced at stages 12.

If so, I think they should be measuring the average at individual centrosomes. Also, why is the level of GFP alone not much higher when expressed with GFPnanoPACT (Fig 1B)? Presumably GFP should be recruited to the centrosomes by GFPnanoPACT.

Regarding the GFP levels, such differences are likely to result from different molecules being targeted to the centrioles. Therefore, it is very likely that the amount of these proteins at the centriole might be influenced by other factors, such as protein folding, degradation, as well as the efficiency of how the nanobody binds to the GFP. The most important conclusion from this analysis is that the levels of ANA1-GFP at the centrioles are much higher when ANA1 is targeted by GFPnanoPACT, in comparison with no targeting (figure 4B). This is an important result as targeting ANA1 to centrioles prevents centriole loss, which supports our findings that higher levels of ANA1 provide stability to centrioles.

8) The authors show that tethering Ana1-GFP to the centrioles counteracts centriole loss in oocytes (Fig4G). They say that the centrosomes are most likely inactive because they don't recruit PCM, but they have only looked at γ -tubulin, which is a downstream component of the PCM. I think it is important to check whether Polo is recruited, given that tethering Polo to centrioles also counteracts centriole loss and that a recent paper showed that Ana1 has a role in recruiting Polo to centrosomes (Alvarez-Rodigo et al., 2021). The authors also say that these centrosomes do not organise microtubules but do not show the data.

As discussed in our manuscript, we expect centrioles maintained by tethering of ANA1-GFP to be inactive as we did not observe any obvious defects in meiosis, and those eggs lead to viable zygotes after fertilization. This is in sharp contrast to the targeting of Polo, where centrioles contain gamma-tubulin (figure 4C and (Pimenta-Marques et al., 2016)) and nucleate microtubules which interfere with meiosis, mitosis and embryo development (Pimenta-Marques et al., 2016). In the revised version of our manuscript we will generate flies expressing ANA1-GFP targeted to centrioles by the co-expression with the GFPnanoPACT construct, and we will additionally express Jupiter-mCherry (Lowe et al., 2014) as a reporter for centrosomal microtubule nucleation.

Regarding the hypothesis raised by this reviewer, that ANA1 is involved in the recruitment of Polo, we have addressed this point in the reviewer's comment 6, where we explained why we don't favor a scenario where Polo is recruited by ANA1 for a role in centriole integrity.

Revision Plan

9) The authors propose that Ana1 is downstream of the PCM, and so over-expressing Ana1 should at least partially rescue centriole loss after PCM depletion. But I don't really agree with this. If Ana1 relies on the PCM then how would its overexpression manage to rescue the phenotype in the absence of the PCM? The finding that over-expressing Ana1 partially rescues centriole loss may instead suggest that Ana1 is either upstream of the PCM or part of an independent pathway. Indeed, the authors show that depletion of both the PCM and Ana1 has a stronger effect than either depletions individually - this is indicative of two independent pathways.

We thank the reviewer for his/her comment. Please see answer 1 to reviewer 1.

Minor

comments

1) When the authors say that the centriole wall and cartwheel components are "dynamic" I think that they need to make it clear that this "dynamicity" is not very fast. Using the term dynamic tends to suggest rapid turnover (like in the PCM). Perhaps the authors could use the term "slow exchange" or something similar.

We thank the reviewer for this comment and will change the wording accordingly.

2) The authors currently use a 0 or 1 centriole categorisation - it would be nice to see the breakdown of what percentage of cells have 0, 1, 2, or >2 centrioles, perhaps in a supplementary excel file.

We thank the reviewer for his/her comment and suggestion. We will complement the existing data with more information.

Reviewer #2 (Significance (Required)):

How centrioles are eliminated in certain cells is an interesting question and the data presented is also relevant to understanding centriole biology in general, because it seems that some apparently very stable structural proteins actually turnover. It is widely known that PCM proteins turnover relatively quickly, but core centriole proteins are considered to be stably incorporated. The data will therefore raise interest in the centrosome field. I do, however, feel that for the authors to make this point more strongly it would be good to show this more directly. Overall, this is a very interesting paper that is well written. The data is well presented and supports the conclusions that centriole components turnover and that Ana1 is involved in maintaining centriole integrity.

We are glad this Reviewer thinks this is an important paper in centriole and centrosome biology.

Reviewer #3 (Evidence, reproducibility and clarity (Required)):

The team explores a previously developed "centriole stability assay" to monitor the disappearance of centrioles after RNAi-dependent depletion of various centrosome components. Important roles in centriole stability are found for the PCM and for cartwheel proteins, in addition to proteins of the centriole wall. The remainder of the study focuses on the centriole wall protein ANA1: induced degradation of ANA1 during *Drosophila* oogenesis strongly reduces the PCM and other centriole markers, and ANA1-dependent defects cannot be prevented by GFP-Polo-PACT, which is

Revision Plan

otherwise known to protect from the loss of PCM. In complementary experiments, forced targeting of ANA1 to the PCM, or overexpression of AN1 protects centriole integrity.

Reviewer #3 (Significance (Required)):

The study shows that ANA1 is important for the integrity of centrosomes. Generally, this work is well executed and correctly controlled. The novelty of the results is somewhat limited, since a role of ANA1 in centrosome assembly has already been reported by others. The present work emphasizes aspects of centrosome protein maintenance, but doesn't provide mechanistic details of protein turnover. The manuscript should be of interest to the scientific community working on the centrosome.

We thank the reviewer for the comments and are glad that the reviewer considers that our work was well executed.

Other comments:

I wonder whether the results from the centrosome maintenance experiment with GFP-Polo-PACT (Figure 3) are really very telling: since PCM and other centriole markers are lost upon ANA1-depletion, GFP-Polo-PACT cannot target to the PCM, and it is therefore unsurprising that GFP-Polo-PACT fails to provide its protective effect. Would expression of GFP-Polo-PACT prior to addition of ANA1-RNAi have a protective effect?

We thank the reviewer for the comment. We understand the point raised, and this might underlie why GFP-Polo-PACT only partially rescues "All PCM" depleted cells. Importantly, upon depletion of ANA1, GFP-Polo-PACT loses its ability to partially rescue centriole numbers, supporting that ANA1 is a player in this pathway. It is actually likely that GFP-Polo-PACT expression occurs before ANA1 RNAi has an effect, as ANA1 RNAi was performed only 16H before GFP-Polo-PACT transfection. We will discuss this experiment design in the text.

Minor point:

Figure 1H: it is unclear to me how centrioles are identified with the BLD10 marker in samples that have been treated with BLD10 RNAi.

We thank the reviewer for this comment. In our screen centrosomes were identified by using different markers, besides BLD10. Such markers include the PCM protein D-PLP, the centriolar wall proteins SAS4 and ANA1, as well as CP110, which is a protein present at the distal end of the centriole (figure 1 and supplementary figure 1). Accordingly, in the case of BLD10 RNAi, we analyzed this condition by taking in account not only the staining provided by the BLD10 antibody, but also the referred markers.

****Referees cross-commenting****

I agree very much with reviewers 1 and 2 that a role of ANA1 "downstream" of the PCM is not really supported by the data.

We thank the reviewer for this comment. This was also an issue raised by the other reviewers to which we have already replied.

Revision Plan

I also think that all other points raised in the reviews merit attention.

REFERENCES USED IN REPLIES TO REVIEWERS:

- Alvarez-Rodrigo, I., Wainman, A., Saurya, S., & Raff, J. W. (2021). Ana1 helps recruit Polo to centrioles to promote mitotic PCM assembly and centriole elongation. *Journal of Cell Science*, 134(14). <https://doi.org/10.1242/jcs.258987>
- Bahtz, R., Seidler, J., Arnold, M., Haselmann-Weiss, U., Antony, C., Lehmann, W. D., & Hoffmann, I. (2012). GCP6 is a substrate of Plk4 and required for centriole duplication. *Journal of Cell Science*, 125(2), 486–496. <https://doi.org/10.1242/jcs.093930>
- Balczon, R., Bao, L., Zimmer, W. E., Brown, K., Zinkowski, R. P., & Brinkley, B. R. (1995). Dissociation of centrosome replication events from cycles of DNA synthesis and mitotic division in hydroxyurea-arrested Chinese hamster ovary cells. *Journal of Cell Biology*, 130(1), 105–115. <https://doi.org/10.1083/jcb.130.1.105>
- Dzhindzhev, N. S., Tzolovsky, G., Lipinski, Z., Schneider, S., Lattao, R., Fu, J., ... Glover, D. M. (2014). Plk4 phosphorylates ana2 to trigger SAS6 recruitment and procentriole formation. *Current Biology*, 24(21), 2526–2532. <https://doi.org/10.1016/j.cub.2014.08.061>
- Dzhindzhev, N. S., Yu, Q. D., Weiskopf, K., Tzolovsky, G., Cunha-Ferreira, I., Riparbelli, M., ... Glover, D. M. (2010). Asterless is a scaffold for the onset of centriole assembly. *Nature*, 467(7316), 714–718. <https://doi.org/10.1038/nature09445>
- Lowe, N., Rees, J. S., Roote, J., Ryder, E., Armean, I. M., Johnson, G., ... Johnston, D. S. (2014). Analysis of the expression patterns, Subcellular localisations and interaction partners of drosophila proteins using a pigp protein trap library. *Development (Cambridge)*, 141(20), 3994–4005. <https://doi.org/10.1242/dev.111054>
- Moyer, T. C., & Holland, A. J. (2019). Plk4 promotes centriole duplication by phosphorylating stil to link the procentriole cartwheel to the microtubule wall. *ELife*, 8. <https://doi.org/10.7554/eLife.46054>
- Pimenta-Marques, A., Bento, I., Lopes, C. A. M., Duarte, P., Jana, S. C., & Bettencourt-Dias, M. (2016). A mechanism for the elimination of the female gamete centrosome in *Drosophila melanogaster*. *Science*, 353(6294), aaf4866–aaf4866. <https://doi.org/10.1126/science.aaf4866>
- Stucke, V. M., Silljé, H. H. W., Arnaud, L., & Nigg, E. A. (2002). Human Mps1 kinase is required for the spindle assembly checkpoint but not for centrosome duplication. *EMBO Journal*, 21(7), 1723–1732. <https://doi.org/10.1093/emboj/21.7.1723>

3. Description of the revisions that have already been incorporated in the transferred manuscript

Please insert a point-by-point reply describing the revisions that were already carried out and included in the transferred manuscript. If no revisions have been carried out yet, please leave this section empty.

4. Description of analyses that authors prefer not to carry out

Please include a point-by-point response explaining why some of the requested data or additional analyses might not be necessary or cannot be provided within the scope of a revision. This can be due to time or resource limitations or in case of disagreement about the necessity of such additional data given the scope of the study. Please leave empty if not applicable.

Here we highlight the replies made to the reviewers where we discuss the analysis we cannot carry out (responses are shown in blue):

Revision Plan

Reviewer #2

4) A key conclusion is that core centriole components turnover to some extent and that the incorporation of new molecules is necessary for centriole maintenance. This is a very interesting and important point and so it would be nice to have more direct data to support it. This could be done in different ways, including transfecting fluorescently tagged centriole components after S-phase arrest and showing that some molecules become incorporated into the centrioles, or by performing FRAP experiments. Of course, it is possible that the turnover is so low that the incorporated fluorescent molecules cannot be detected...

We are glad that the reviewer agrees this is an interesting point. We thank the reviewer for this suggestion. However, performing FRAP in our experimental setup, as discussed with EMBO editor, is very difficult. In *Drosophila* cultured cells, such experiments would require arresting cells in S-phase for 8 days, while transfecting with two constructs (to identify centrosome and to FRAP candidate protein), which is complicated. We have attempted to perform such experiments in oogenesis, however, centrioles are very dynamic and move, and also, oocytes are thick, which makes this experiment technically very difficult.

5) The authors show that depletion of Ana1 from oocytes leads to a reduction in the intensity of centriole markers. They do not measure centrosome numbers, as the centrosomes cluster too tightly. The authors therefore can't be certain that Ana1 depletion leads to a reduction in centrosome numbers. The authors could show this by inhibiting centrosome clustering while depleting Ana1. There is a recent BioRxiv paper showing that centrosome clustering can be inhibited by depletion of Kinesin-1.

We thank the reviewer for this comment, however, although we cannot individualize centrosomes, our data strongly suggests that ANA1 depletion does indeed lead to a loss of the whole centrosomal structure. Our results show that upon targeting ANA1 for degradation by using the GFPdegrad system both PCM components (D-PLP and gamma-tubulin) as well as the centriolar protein CP110 are reduced from the centrosomal foci. This suggests that the centrosome as a hole is being prematurely lost rather than the specific loss of components whose maintenance could directly depend on ANA1. Moreover these results are consistent with what we observe in cultured cells, where the percentage of cells with 0-1 centrioles increases dramatically upon ANA1 RNAi treatment. Therefore, the data strongly suggests that the degradation of ANA1 in oogenesis is most likely leading to centriole loss. The suggested experiment would be very time consuming, with no guarantees of generating healthy flies suitable for such experiments, given the large number of insertions needed in the genome.

4th Jul 2022

RE: Manuscript EMBOR-2022-55667V1, Ana1/CEP295 is an essential player in the centrosome maintenance program regulated by Polo kinase

Dear Dr. Pimenta-Marques,

Thank you for the submission of your manuscript to EMBO reports. I have now had the opportunity to read and to discuss it with my colleagues here, including our chief editor Bernd Pulverer, and I regret to say that we all agree that the manuscript is not well suited for our journal.

We note that your study investigates the effects of RNAi-mediated depletion of centrosome components on the maintenance of centriole stability in *Drosophila*. It identifies and implicates the centriole wall protein ANA1 in centriole maintenance. We appreciate that your findings contribute to a better understanding of the components that are involved in centriole stability and maintenance, which will certainly be of interest to the centrosome research community.

However, we also note that your study does not provide insight into the dynamics of centriolar maintenance, or how proteins are turned over in centriole maintenance. In particular, it does not address the relationships between ANA1 and PCM components or the Polo kinase, which would have strengthened your report and disambiguated your interpretation of ANA1 functioning downstream of the PCM, which raised concerns with the referees. Also, given that ANA1 is a known component of the centriolar wall, we think that a role in centriole maintenance is not unexpected. Taken together, we have therefore decided not to proceed with the further handling of your manuscript at EMBO reports.

While we cannot pursue this manuscript further, we encourage you to transfer your study to our not-for-profit open-access sister journal, Life Science Alliance (LSA). We shared your manuscript and the accompanying proposed revision plan with LSA Scientific Editor, Novella Guidi, who is interested in these findings, and would like to invite further consideration of this manuscript at LSA provided you revise the manuscript in accordance to what you have laid out in your revision plan. You do not need to revise the manuscript before transferring it to LSA. Once you transfer, Dr. Guidi will email you an invitation to revise and resubmit, listing the same request as mentioned above. Please feel free to reach out at n.guidi@life-science-alliance.org if you have any questions about the LSA journal, the transfer process or the revisions requested.

As a service to authors, EMBO provides authors with the possibility to transfer a manuscript that one journal cannot offer to publish to another EMBO publication. The full manuscript and if applicable, reviewers reports are automatically sent to the receiving journal to allow for fast handling and a prompt decision on your manuscript. For more details of this service, and to transfer your manuscript to another EMBO title please click on Link Not Available

Kind regards,
Esther

=====
Rev_Com_number: RC-2022-01412

New_manu_number: EMBOR-2022-55667V1

Corr_author: Pimenta-Marques

Title: Ana1/CEP295 is an essential player in the centrosome maintenance program regulated by Polo kinase

Dear editors,

Thank you for your reply. After our discussion with editor Ieva Gailite we were surprised by your comments and realized we might have not properly explained the importance of the findings of our work.

In your decision letter you use two main arguments: 1. that we did not address the relationship between ANA1 and Polo kinase and ANA1 and the PCM, and 2. that given that ANA1 is a wall component you do not find it surprising that it is involved in centriole maintenance.

In relation to the first one we show for the first time that ANA1 is needed for polo-induced centriole maintenance in the germline. Moreover, we show that in tissue culture cells overexpression of ANA1 is sufficient to overcome the centriole loss induced by depletion of the PCM. Finally, and this also addresses the second point, ANA1, contrary to other wall components is sufficient to maintain centrioles in the germline.

The concept of maintenance is one that is yet very little explored in biology. Moreover, the fact that structures that are as stable as centrioles need to be actively maintained by expression of some critical components is very novel. Moreover, we take advantage in our studies of the beautiful phenomena of centriole elimination in the female germline as a way of studying a mechanism that may be universal, but is at the same time very relevant for developmental processes and fertility.

We believe that our findings have different critical implications, different from the ones previously shown in the context of cell proliferation and therefore centriole duplication. Through development, many cells halt the cell cycle and/or differentiate. In contrast to proliferating cells, we know very little about the role and maintenance of centrioles in such cells. Therefore, understanding how centrioles can be maintained or destabilized in such cells, and which are the critical components for that, might have critical implications. Our study opens the door for future studies in normal development, but also in diseases where centrosomes are altered, such as cancer. We hope you can reconsider your decision.

Best regards

Ana Marques and Mónica Bettencourt-Dias

Dear Ana,

Thank you for your email asking us to reconsider our decision on your manuscript. I have now received the comments from an advisor/arbitrator, which are included below.

Basically, the advisor agrees that a revised study could be a good fit for EMBO reports. However, s/he points out that the following 2 points raised by the reviewers need to be better addressed experimentally:

1. Clearly there is a lot of confusion by some of the conclusions, for example why the authors claim that Ana1 functions downstream of PCM, although this is not well supported by their data. Considering this is one of the few mechanistic insight, and based on the reviewers comments this should be better addressed with experiments.
2. The authors suggest that GFP-Polo-PACT partial rescue after all PCM depletion is due to dsRNA not being 100% efficient. I understand that but this could be then shown here. Could the authors look at pcm recruitment in these conditions? Can they detect pcm recruitment to the centrosomes in 'all pcm depleted' conditions? And the same for Ana1 overexpression rescue of centriole loss in 'all pcm depletion'. Otherwise it is very confusing.

If you think that you can experimentally address and strengthen these points, we will be happy to consider a successfully revised manuscript for publication here.

I would thus like to invite you to revise your manuscript along the lines you suggest and as outlined above, with the understanding that the referee concerns must be fully addressed and their suggestions taken on board. Please address all referee concerns in a complete point-by-point response. Acceptance of the manuscript will depend on a positive outcome of a second round of review. It is EMBO reports policy to allow a single round of major revision only and acceptance or rejection of the manuscript will therefore depend on the completeness of your responses included in the next, final version of the manuscript.

We realize that it is difficult to revise to a specific deadline. In the interest of protecting the conceptual advance provided by the work, we recommend a revision within 3 months (19th Oct 2022). Please discuss the revision progress ahead of this time with the editor if you require more time to complete the revisions.

- 1) A data availability section providing access to data deposited in public databases is missing. If you have not deposited any data, please add a sentence to the data availability section that explains that.
- 2) Your manuscript contains statistics and error bars based on $n=2$. Please use scatter blots in these cases. No statistics should be calculated if $n=2$.

3) We replaced Supplementary Information with Expanded View (EV) Figures and Tables that are collapsible/expandable online. A maximum of 5 EV Figures can be typeset. EV Figures should be cited as 'Figure EV1, Figure EV2' etc... in the text and their respective legends should be included in the main text after the legends of regular figures.

5) a complete author checklist, which you can download from our author guidelines <<https://www.embopress.org/page/journal/14693178/authorguide>>. Please insert information in the checklist that is also reflected in the manuscript. The completed author checklist will also be part of the RPF.

6) Please note that all corresponding authors are required to supply an ORCID ID for their name upon submission of a revised manuscript (<<https://orcid.org/>>). Please find instructions on how to link your ORCID ID to your account in our manuscript tracking system in our Author guidelines <<https://www.embopress.org/page/journal/14693178/authorguide#authorshipguidelines>>

8) We would also encourage you to include the source data for figure panels that show essential data. Numerical data should be provided as individual .xls or .csv files (including a tab describing the data). For blots or microscopy, uncropped images should be submitted (using a zip archive if multiple images need to be supplied for one panel). Additional information on source data and instruction on how to label the files are available at <<https://www.embopress.org/page/journal/14693178/authorguide#sourcedata>>.

- the name of the statistical test used to generate error bars and P values,
- the number (n) of independent experiments (please specify technical or biological replicates) underlying each data point,
- the nature of the bars and error bars (s.d., s.e.m.),
- If the data are obtained from n {less than or equal to} 2, use scatter blots showing the individual data points.

We would also welcome the submission of cover suggestions, or motifs to be used by our Graphics Illustrator in designing a

cover.

I look forward to seeing a revised form of your manuscript when it is ready.

Manuscript number: #RC-2022-01412

Corresponding author(s): Ana, Pimenta-Marques; Mónica, Bettencourt-Dias

Rebuttal

Dear Esther,

We thank you, the reviewers and advisor for your reviews and comments in our revision plans. It took us longer than expected to resubmit as the two first authors could not dedicate as much time to it (starting a lab/applying for grants; leaving the lab). We have now been able to finish those plans, substantially improving the manuscript. You can find our answers below in blue and changes in the manuscript also in blue.

1. Clearly there is a lot of confusion by some of the conclusions, for example why the authors claim that Ana1 functions downstream of PCM, although this is not well supported by their data. Considering this is one of the few mechanistic insights, and based on the reviewer comments this should be better addressed with experiments.

We have now performed more experiments and have changed the text (abstract, results and discussion) to be clearer about the role of ANA1 in centriole stability.

2. The authors suggest that GFP-Polo-PACT partial rescue after all PCM depletion is due to dsRNA not being 100% efficient. I understand that but this could be then shown here. Could the authors look at pcm recruitment in these conditions? Can they detect pcm recruitment to the centrosomes in 'all pcm depleted' conditions? And the same for Ana1 overexpression rescue of centriole loss in 'all pcm depletion'. Otherwise it is very confusing.

We thank the advisor for this suggestion as this experiment substantially improved the interpretation of results. Contrary to our expectations, there was no rescue of the PCM markers (D-PLP, SPD-2 and ASL; new supplementary figure 3, B-E) in Polo-PACT partial rescue. Importantly, ANA1 was present in the "rescued" centrioles (new supplementary figure 3, F). Furthermore, we did not observe rescue if ANA1 was depleted (condition of ALLPCM + ANA1 RNAi, figure 3B)). Therefore, we conclude that Polo-PACT needs ANA1 to be able to promote the rescue. We have changed the text accordingly and we think the manuscript has substantially improved with these new experiments.

Reviewer #1 (Evidence, reproducibility and clarity (Required)):

Summary:

In this manuscript Pimenta-Marques build on their previous work addressing how centrioles are stabilized and maintained or destabilized and disassembled, depending on the cell type and developmental context. Using *Drosophila* cell culture and oogenesis as an *in vivo* model for centriole destabilization, they identify the centriole wall protein Ana1 as a central player in centriole stability. Its presence is required for the maintenance even of mature centrioles, suggesting that there is continued turnover of centriole structural components.

Major comments:

1) The experiments and results are very well described and most of the conclusions are supported by the data. One aspect needs clarification though. It is not clear to this reviewer how the authors envision the regulation and mechanism by which Ana1 functions in centriole stability. The data suggest that it can stabilize centrioles independent of PCM (Fig. 3B, 5B), yet the authors claim in the results and discussion that it functions downstream of PCM. As presented, this does not make sense. I would argue the opposite, it may function upstream or in parallel to the PCM.

We thank the reviewer for his/her comments. We realize the writing was not clear and have now revised the text and added additional experiments that show that Ana1 can stabilise centrioles independently of the PCM as suggested by the reviewer (see point 2 above, new supplementary figure 3 and new supplementary figure 5). We found that in cells depleted of PCM ("All PCM" RNAi), such centrioles when rescued by expressing ANA1-GFP, do not contain the PCM marker SPD-2 (new supplementary figure 5 B). Consistent with this, in cells depleted of PCM, when centriole loss is rescued by expressing GFP-Polo-PACT, rescued centrioles do not contain PCM (D-PLP, SPD-2 and ASL), but contain ANA1 (new supplementary figure 3 B-F).

Related to the above, the last sentence of the intro states: "Finally, we found that both Polo and the PCM require ANA1 to promote centriole structural integrity." This is shown for Polo, but where is the data showing that PCM requires ANA1 for promoting centriole stability?

This is a difficult question to address in oogenesis. During oogenesis, several PCM components are downregulated before centrioles are eliminated (1). In this context expressing single PCM components was not sufficient to prevent centriole loss (1). Therefore, the depletion of ANA1 in a scenario where we ectopically "rescue" PCM downregulation in oogenesis, would require the simultaneous expression of the 4 major PCM proteins. Such experiment is technically difficult to perform. Alternatively, we expressed Polo (tethered to the centriole), which prevented PCM loss and centriole loss (1). Importantly, removal of ANA1, in a context where the PCM is present (for example in cultured cells) is sufficient to lead to centriole instability, suggesting that the stability promoted by the PCM relies (directly or indirectly) on having ANA1. We understand that the way we had written the manuscript was leading to ambiguity, and have now changed and substantially simplified the abstract, intro, results, and discussion to prevent overstated and erroneous interpretations and conclusions (please see modified text in blue).

2) I have a concern regarding the number *n* used for statistics in the quantifications. In many cases it seems that the number *n* of cells etc. was used (e.g. $n > 100$ cells) rather than the number of experiments (e.g. $n = 3$). The statistics should measure variability between experimental repetitions, not between cells etc. If statistics were indeed not done on experiments and would have to be changed, some of the observed effects may not be statistically significant and would require additional experimental replicates, which would increase the time needed for revision.

We agree with the reviewer that the text was not clear. We added more details in the figure legends and added a new section in Material and Methods on statistics and number of replicates and samples. For all the experiments in cells and eggs we always performed 3 independent biological replicates for each condition.

We have additionally reanalysed our data with the statistics facility from the IGC, ensuring the selection of the most appropriate tests for our datasets. Consequently, we have revised the statistical tests applied to all experiments related to oogenesis. It's noteworthy that this adjustment has not significantly altered the significance of our findings. We express gratitude to the reviewer for their input, as their comment has facilitated enhancements in our statistical methodologies.

Minor comments:

1) I would advise the authors to improve the presentation of the figures. In particular the labels are in many cases very small and difficult to read. Readability is also reduced by the use of bold font in the labels and a mix of various font sizes within single figure panels.

We thank the reviewer for this comment. We have improved the figures to make them easier to read.

2) The result section could be shortened/become more readable by moving several paragraphs to the intro or discussion. 3) The introduction is quite long and some parts read more like an introduction of a review on the topic.

We have revised and shortened the text.

Reviewer #1 (Significance (Required)):

This is a nice, focused study on the requirements underlying centriole stability and maintenance. The first part identifies the cartwheel, the centriole wall, and the PCM as important for centriole maintenance. The remaining parts identify and focus on the essential role of ANA1 in this process. This is an important finding, since the mechanisms underlying centriole stability and maintenance are poorly understood, yet highly relevant. Some cell types inactivate and/or disassemble centrioles during differentiation and this is likely important to their function. Providing more mechanistic insight, for example, regarding the relationship between ANA1 and PCM recruitment or the regulation of ANA1's centriole function by Polo, would have further strengthened the study. The audience interested in this work will be cell and developmental biologists.

My expertise is in centrosome biology and microtubule organization.

We are glad this reviewer thinks this is an important study and that it furthers our understanding on the mechanisms underlying centriole stability.

****Referees cross-commenting****

I agree with the additional points raised by the other reviewers. I still think that overall the paper is fine and most things could be addressed in a reasonable time frame. The work does not provide much mechanism though. In this regard, the confusing placement of ANA1 downstream of PCM, would be the only mechanistic aspect, and it seems the authors got it wrong, at least based on the provided data. Here, additional experiments could elucidate these relationships further, but if

this is not the goal, text changes could also address this and it would remain a smaller, more focused study.

We have now revised our model and text to avoid ambiguity.

Reviewer #2 (Evidence, reproducibility, and clarity (Required)):

In this paper, the authors show that the turnover of centriole components is necessary for proper centriole maintenance within *Drosophila* cultured cells (during prologued cell cycle arrest) and within *Drosophila* oocytes, where centrioles are normally degraded prior to fertilisation. They highlight Ana1 as an important player in centriole maintenance. The authors begin with a candidate screen to identify core centriole proteins that are required to properly maintain centrioles. They then focus on Ana1, given that its depletion had the strongest effect, and show that its depletion leads to a reduction in the levels of centriole components in *Drosophila* oocytes. They show that the previously observed ability of centriole-targeted Polo to counteract centriole loss depends at least in part on Ana1 and that targeting Ana1 to centrioles also counteracts centriole loss. The authors conclude that Ana1 is a component of the PCM-promoted centriole integrity pathway.

Major comments

1) The authors say that Plk4 depletion does not lead to centriole loss, but there are significant differences in centriole number between the control and Plk4 depletion cells in Fig 1F and S1D. Please comment.

As the reviewer noticed, PLK4 depletion leads to a significant difference in centriole number. However, this is a very modest effect, as compared to the one observed after depletion of cartwheel or wall components. This may be an indirect effect, resulting from PLK4's known ability to interact, phosphorylate and recruit components that we show here to be important for centriole maintenance, such as CPAP/SAS4 (2,3). The observed loss of SAS4 foci upon Plk4 depletion suggests that PLK4 might be involved in SAS4 maintenance, additionally to its known role in recruitment to centrioles (2). This result is in line with other experiments where cells were subjected to PLK4 inhibition for a very long time and loss of centriole integrity was not observed (4,5). This is now explained in the modified text at the end of page 4.

2) One of the main results is that depletion of centriole components leads to a reduction in centrosome numbers when measured 8 days after S-phase arrest. I wonder whether a restriction of centriole duplication could add to this effect? Any cells that were in G2 or M phase when the drugs were added would presumably progress into the following S-phase and duplicate their inherited centrioles, but not if centriole duplication proteins had been depleted. It's true that Plk4 depletion leads to a relatively mild centriole loss phenotype, but can the authors be sure that this is not due to variations in the efficiency of different RNAi constructs? Perhaps the authors can show that Plk4 depletion efficiently prevents centriole duplication under otherwise normal conditions.

In contrast to some cultured Human cell lines (6,7), *Drosophila D.mel* cells do not reduplicate their centrioles upon induced S phase arrest (6,7). Therefore, upon treatment with HU and APH, as seen in controls, the number of centrioles remains unchanged, which allows us to address the effect of the different depletions in centriole stability. In our experimental assay, developed in Pimenta-Marques et al. (2016), both dsRNA and drugs used to arrest cells in S phase, were administered simultaneously. Given that the dsRNA takes at least 24 hrs to have an effect and

the cell cycle of DMEL cells is approximately 24 hours, the targeted proteins should not be depleted before the drugs arrest the cells. Indeed, we only normally score duplication phenotypes at 3 to 4 days post PLK4 dsRNA treatment of cycling cells (8). We have now better explained the experimental setting and interpretation of this experiment in the text (Page 3) and figure legends.

3) The authors show that Ana1 depletion has the strongest effect, but this could in theory be due to differences in RNAi efficiency. I don't expect the authors to show the efficiency of all RNAi constructs, but they could state in the text that this is a caveat e.g. "...although we cannot rule out the possibility that differences in RNAi efficiency lead to the observed differences in severity of phenotype..."

We thank the reviewer for the comment and have included that suggestion "From all the candidates tested, although we cannot rule out the possibility that differences in RNAi efficiency led to the observed differences in severity of phenotype, depletion of ANA1 led to the strongest effect on different centrosomal markers" (beginning page 5).

4) A key conclusion is that core centriole components turnover to some extent and that the incorporation of new molecules is necessary for centriole maintenance. This is a very interesting and important point and so it would be nice to have more direct data to support it. This could be done in different ways, including transfecting fluorescently tagged centriole components after S-phase arrest and showing that some molecules become incorporated into the centrioles, or by performing FRAP experiments. Of course, it is possible that the turnover is so low that the incorporated fluorescent molecules cannot be detected...

We are glad that the reviewer agrees this is an interesting point. However, performing FRAP in our experimental setup is very difficult. In *Drosophila* cultured cells, such experiments require arresting cells in S-phase for 8 days, while transfecting with two constructs (to identify centrosome and to FRAP candidate protein), which proved to be complicated. We have also attempted to perform such experiments in oogenesis; however, oocytes are very thick, and centrioles are very dynamic and move. These two conditions made this experiment technically very difficult, despite the multiple trials. We discussed this limitation in the rebuttal plan that was accepted by the editor.

5) The authors show that depletion of Ana1 from oocytes leads to a reduction in the intensity of centriole markers. They do not measure centrosome numbers, as the centrosomes cluster too tightly. The authors therefore can't be certain that Ana1 depletion leads to a reduction in centrosome numbers. The authors could show this by inhibiting centrosome clustering while depleting Ana1. There is a recent BioRxiv paper showing that centrosome clustering can be inhibited by depletion of Kinesin-1.

In this work, we used the strategy validated in Marques et al (2016) (1) where we measured the intensity of the cluster and of individual scattered centrioles and could account for the 64 centrioles existing in the oocyte. The suggested experiment would be very time consuming, given the large number of insertions needed in the genome, with no guarantees of generating healthy flies suitable for such experiments.

Although we cannot individualize centrosomes, our data strongly suggests that ANA1 depletion leads to centrosome loss. Our results show that upon targeting ANA1 for degradation with the GFPdegrad system, both PCM and centriole components (D-PLP, gamma-tubulin and CP110) are reduced from the centrosomal foci (figure 2). This suggests that the whole centrosome is being prematurely lost rather than the specific loss of components whose maintenance depends on ANA1. These results are consistent with what we observed in cultured cells, where the

percentage of cells with 0-1 centrioles strongly increases upon ANA1 RNAi treatment (figure 1). Therefore, the data strongly suggests that the degradation of ANA1 in oogenesis leads to centriole loss. We thank the reviewer for pointing this issue and have added a sentence in the text to say that the methodology used was validated before (Page 6).

6) In Figure 3B the authors show that expression of GFP-Polo-PACT partially rescues the effect of "all PCM" depletion, but this seems strange given that Polo's role is presumably to recruit PCM (which has been depleted). Can the authors comment?

We have investigated what the reviewer suggested. Contrary to our initial expectations, in Polo-PACT partial rescue, there is no partial rescue of the PCM markers D-PLP, SPD-2 and ASL (new Supplementary Figure 3, B-E). However, and importantly, ANA1 is partially rescued (new Supplementary Figure 3 F). Given that we do not observe Polo-PACT partial rescue of centrioles if ANA1 is also depleted (condition of ALLPCM + ANA1 RNAi, figure 3 B), we conclude that Polo-PACT promotes the rescue at least in part through ANA1's function in centriole maintenance. We have changed the text accordingly.

Also, it would make sense to test whether GFP-Polo-PACT can rescue centriole loss after the depletion of Ana1 alone (not Ana1 and all PCM). If Ana1 has a role in recruiting Polo (either directly or indirectly), which has been shown previously in mitotic cells, then there should be a rescue to some extent.

We thank the reviewer for this comment and have now performed the suggested experiment. We depleted ANA1 and expressed GFP-Polo-PACT in our centriole stability assay, where cells were arrested in S-phase for 4 days. We found that GFP-Polo-PACT cannot rescue centriole loss induced by depletion of ANA1 (data now presented in new supplementary figure 3 H). This is consistent with the new experiments we have performed, where we show that centrioles which are rescued by GFP-Polo-PACT in cells depleted of PCM, contain ANA1 (new supplementary figure 3 F).

Alvarez-Rodrigo et al., 2021 showed that phosphorylation of ANA1 by Polo at predicted Polo-box binding domains is important for Polo recruitment to centrosomes in mitotic embryos but is not required for centriole structural integrity. Therefore, we do not favor a scenario where ANA1 recruits Polo for a role in the maintenance of centriole integrity in interphase cells.

7) In Fig4A,C, the authors say that γ -tubulin levels at centrosomes increase when GFP-Polo is forced onto the centrosomes - the graph seems to show a big increase, but the pictures do not...? Are the authors measuring total levels at all centrosomes?

We thank the reviewer for this comment. We have now substituted the image for an image which best represents our observations. The increase of γ -tubulin levels upon tethering of Polo to centrioles is an observation which we have previously made by expressing GFP-Polo-PACT (1). Here these observations were confirmed by tethering Polo to centrioles with the GFPnanobodyPACT construct.

In this experiment, different proteins were tethered to centrioles by the GFPnanoPACT construct, and the intensity of the different markers was analyzed by measuring the sum intensity levels of GFP, gamma-tubulin, CNN and Jupiter-mCherry on centrosomes. This is now explained in the methods section.

If so, I think they should be measuring the average at individual centrosomes. Also, why is the level of GFP alone not much higher when expressed with GFPnanoPACT (Fig 1B)? Presumably GFP should be recruited to the centrosomes by GFPnanoPACT.

As previously explained (point 5), we cannot identify individual centrosomes (1). Centrioles were identified based on the colocalization between the BLD10 antibody staining (centriole marker) and the GFP signal of each condition. We have previously found that in these stages of oogenesis (stages 10) core centriolar components, such as BLD10, ANA1 and SAS6 are still present (1) with centriolar loss being more pronounced at stages 12.

Regarding the total GFP levels observed after targeting different components, it is likely that the amount of these proteins at the centriole might be influenced by several factors, including expression, post translation regulation, as well as the efficiency of how the nanobody binds to the GFP. The important conclusion from this analysis is that the levels of ANA1-GFP at the centrioles are much higher when ANA1 is targeted by GFPnanoPACT, in comparison with no targeting (figure 4 B), and that only that condition provides stability to centrioles. We have now made a comment about this point in the figure legend.

8) The authors show that tethering Ana1-GFP to the centrioles counteracts centriole loss in oocytes (Fig4G). They say that the centrosomes are most likely inactive because they don't recruit PCM, but they have only looked at γ -tubulin, which is a downstream component of the PCM. I think it is important to check whether Polo is recruited, given that tethering Polo to centrioles also counteracts centriole loss and that a recent paper showed that Ana1 has a role in recruiting Polo to centrosomes (Alvarez-Rodigo et al., 2021). The authors also say that these centrosomes do not organise microtubules but do not show the data.

To complement the γ -tubulin data, as suggested by the reviewer, we have now quantified the levels of CNN, a major PCM component which is phosphorylated by Polo to promote PCM assembly (9). Consistent with our observations for gamma-tubulin, we observed that ANA1 tethering to centrioles did not promote additional recruitment/maintenance of the PCM component CNN (new supplementary figure 4 C).

Furthermore, to investigate microtubule nucleation, we have generated flies in which the GFPnanoPACT construct is co-expressed with Jupiter-mCherry (10), a reporter for centrosomal microtubule nucleation. While tethering of Polo with co-expression of GFPnanoPACT led to a mild but significant increase in the levels of Jupiter-mCherry, the same was not observed for centrioles that were maintained by expressing ANA1 (new supplementary figure 4 A, D), thus validating our claims.

We did not investigate Polo levels as Polo is transcriptionally downregulated in late oogenesis and it is difficult to detect at late stages (11,12). Moreover, if Polo would be present at the centrioles maintained by ectopic ANA1 expression we would have expected those structures to contain PCM and to nucleate microtubules, which they did not.

Regarding the hypothesis raised by this reviewer that ANA1 is involved in the recruitment of Polo this was also addressed in comment 6 above.

9) The authors propose that Ana1 is downstream of the PCM, and so over-expressing Ana1 should at least partially rescue centriole loss after PCM depletion. But I don't really agree with this. If Ana1 relies on the PCM then how would its overexpression manage to rescue the phenotype in the absence of the PCM? The finding that over-expressing Ana1 partially rescues

centriole loss may instead suggest that Ana1 is either upstream of the PCM or part of an independent pathway. Indeed, the authors show that depletion of both the PCM and Ana1 has a stronger effect than either depletions individually - this is indicative of two independent pathways.

We thank the reviewer for this comment. We have re-written the whole manuscript to avoid ambiguity. We suggest that the PCM helps ensuring adequate ANA1 presence at the centriole to ensure its maintenance- this would explain why overexpressing of ANA1 partially rescues PCM depletion; and why double depletion leads to a stronger phenotype.

“Altogether, our data suggests that ANA1 is important for maintaining the integrity of the centriole structure and that the PCM reinforces that role, perhaps through facilitating the incorporation and/or maintenance of ANA1 at the centriole.”

Minor comments

1) When the authors say that the centriole wall and cartwheel components are "dynamic" I think that they need to make it clear that this "dynamicity" is not very fast. Using the term dynamic tends to suggest rapid turnover (like in the PCM). Perhaps the authors could use the term "slow exchange" or something similar.

We thank the reviewer for this comment and have changed the wording in the discussion.

2) The authors currently use a 0 or 1 centriole categorisation - it would be nice to see the breakdown of what percentage of cells have 0, 1, 2, or >2 centrioles, perhaps in a supplementary excel file.

We thank the reviewer for the comment. We have done this analysis (see below) but do not think the data brings additional information and therefore would rather not increase the amount of supplementary info.

Figure 1: Candidate screen in *Drosophila* cultured cells for centrosome maintenance. Breakdown of the categories seen in Fig. S 1.

Reviewer #2 (Significance (Required)):

How centrioles are eliminated in certain cells is an interesting question and the data presented is also relevant to understanding centriole biology in general, because it seems that some apparently very stable structural proteins actually turnover. It is widely known that PCM proteins turnover relatively quickly, but core centriole proteins are considered to be stably incorporated. The data will therefore raise interest in the centrosome field. I do, however, feel that for the authors to make this point more strongly it would be good to show this more directly. Overall, this is a very interesting paper that is well written. The data is well presented and supports the conclusions that centriole components turnover and that Ana1 is involved in maintaining centriole integrity.

We are glad this Reviewer thinks this is an important paper in centriole and centrosome biology.

Reviewer #3 (Evidence, reproducibility and clarity (Required)):

The team explores a previously developed "centriole stability assay" to monitor the disappearance of centrioles after RNAi-dependent depletion of various centrosome components. Important roles in centriole stability are found for the PCM and for cartwheel proteins, in addition to proteins of the centriole wall. The remainder of the study focuses on the centriole wall protein ANA1: induced degradation of ANA1 during *Drosophila* oogenesis strongly reduces the PCM and other centriole markers, and ANA1-dependent defects cannot be prevented by GFP-Polo-PACT, which is otherwise known to protect from the loss of PCM. In complementary experiments, forced targeting of ANA1 to the PCM, or overexpression of AN1 protects centriole integrity.

Reviewer #3 (Significance (Required)):

The study shows that ANA1 is important for the integrity of centrosomes. Generally, this work is well executed and correctly controlled. The novelty of the results is somewhat limited, since a role of ANA1 in centrosome assembly has already been reported by others. The present work emphasizes aspects of centrosome protein maintenance, but doesn't provide mechanistic details of protein turnover. The manuscript should be of interest to the scientific community working on the centrosome.

We thank the reviewer for the comments and are glad that the reviewer considers that our work was well executed.

Other comments:

I wonder whether the results from the centrosome maintenance experiment with GFP-Polo-PACT (Figure 3) are really very telling: since PCM and other centriole markers are lost upon ANA1-depletion, GFP-Polo-PACT cannot target to the PCM, and it is therefore unsurprising that GFP-Polo-PACT fails to provide its protective effect. Would expression of GFP-Polo-PACT prior to addition of ANA1-RNAi have a protective effect?

We thank the reviewer for the comment. It is actually very likely that GFP-Polo-PACT expression occurs before ANA1 or PCM RNAi has an effect, as the RNAi was performed only 16hr before GFP-Polo-PACT transfection (figure 3 A) and it takes 3-4 days to have an effect in these cells(13).

Moreover, upon depletion of ANA1, GFP-Polo-PACT loses its ability to partially rescue centriole numbers (Figure 3 B), supporting that ANA1 is a player in this pathway (as seen also in oocytes). We now discuss this experiment design in the figure legend.

Minor point:

Figure 1H: it is unclear to me how centrioles are identified with the BLD10 marker in samples that have been treated with BLD10 RNAi.

We thank the reviewer for this comment. In our screen, centrosomes were identified by using different markers, besides BLD10. Such markers include the PCM protein D-PLP, the centriolar wall proteins SAS4 and ANA1, as well as the cap protein, CP110 (figure 1 and supplementary figure 1). In the case of BLD10 RNAi, we analyzed its phenotype, taking in account the staining provided by the other markers and observed a significant loss of SAS4, ANA1, CP110 and D-PLP (figure 1 H and supplementary figure 1 F).

****Referees cross-commenting****

I agree very much with reviewers 1 and 2 that a role of ANA1 "downstream" of the PCM is not really supported by the data.

We thank the reviewer for this comment. We have now performed other experiments and substantially changed the text as answered to the other reviewers.

I also think that all other points raised in the reviews merit attention.

REFERENCES USED IN REPLIES TO REVIEWERS:

1. Pimenta-Marques A, Bento I, Lopes CAM, Duarte P, Jana SC, Bettencourt-Dias M. A mechanism for the elimination of the female gamete centrosome in *Drosophila melanogaster*. *Science* (80-) [Internet]. 2016 Jul 1 [cited 2019 Jan 29];353(6294):aaf4866–aaf4866. Available from: <http://www.ncbi.nlm.nih.gov/pubmed/27229142>
2. Moyer TC, Holland AJ. Plk4 promotes centriole duplication by phosphorylating stil to link the procentriole cartwheel to the microtubule wall. *Elife*. 2019 May 1;8.
3. Chang J, Cizmecioglu O, Hoffmann I, Rhee K. PLK2 phosphorylation is critical for CPAP function in procentriole formation during the centrosome cycle. *EMBO J* [Internet]. 2010 Jul 21 [cited 2023 Jul 13];29(14):2395–406. Available from: <http://emboj.embopress.org/cgi/doi/10.1038/emboj.2010.118>
4. Wong YL, Anzola J V., Davis RL, Yoon M, Motamedi A, Kroll A, et al. Reversible centriole depletion with an inhibitor of Polo-like kinase 4. *Science* (80-) [Internet]. 2015 Jun 5 [cited 2022 Mar 28];348(6239):1155–60. Available from: <https://www.science.org/doi/10.1126/science.aaa5111>
5. Nabais C, Pessoa D, de-Carvalho J, van Zanten T, Duarte P, Mayor S, et al. Plk4 triggers autonomous de novo centriole biogenesis and maturation. *J Cell Biol* [Internet]. 2021 May 3 [cited 2021 Mar 25];220(5). Available from: <https://rupress.org/jcb/article/doi/10.1083/jcb.202008090/211915/Plk4-triggers-autonomous-de-novo-centriole>
6. Balczon R, Bao L, Zimmer WE, Brown K, Zinkowski RP, Brinkley BR. Dissociation of centrosome replication events from cycles of DNA synthesis and mitotic division in hydroxyurea-arrested Chinese hamster ovary cells. *J Cell Biol*. 1995 Jul 1;130(1):105–15.
7. Stucke VM, Silljé HHW, Arnaud L, Nigg EA. Human Mps1 kinase is required for the spindle assembly checkpoint but not for centrosome duplication. *EMBO J*. 2002 Apr 2;21(7):1723–32.
8. Bettencourt-Dias M, Rodrigues-Martins A, Carpenter L, Riparbelli M, Lehmann L, Gatt MK, et al. SAK/PLK4 is required for centriole duplication and flagella development. *Curr Biol* [Internet]. 2005 Dec 20 [cited 2019 Jan 29];15(24):2199–207. Available from: <http://www.ncbi.nlm.nih.gov/pubmed/16326102>
9. Conduit PT, Feng Z, Richens JH, Baumbach J, Wainman A, Bakshi SD, et al. The centrosome-specific phosphorylation of Cnn by Polo/Plk1 drives Cnn scaffold assembly and centrosome maturation. *Dev Cell*. 2014 Mar 31;28(6):659–69.
10. Lowe N, Rees JS, Roote J, Ryder E, Armean IM, Johnson G, et al. Analysis of the expression patterns, Subcellular localisations and interaction partners of drosophila proteins using a pigp protein trap library. *Dev*. 2014 Oct 1;141(20):3994–4005.
11. Jambor H, Surendranath V, Kalinka AT, Mejstrik P, Saalfeld S, Tomancak P. Systematic imaging reveals features and changing localization of mRNAs in *Drosophila* development. *Elife*. 2015 Apr 2;2015(4).
12. Xiang Y, Takeo S, Florens L, Hughes SE, Huo L-J, Gilliland WD, et al. The Inhibition of Polo Kinase by Matrimony Maintains G2 Arrest in the Meiotic Cell Cycle. *PLoS Biol* [Internet]. 2007 [cited 2014 Jan 16];5(12):e323. Available from: <http://biology.plosjournals.org/perlserv/?request=get-document&doi=10.1371%2Fjournal.pbio.0050323>
13. Bettencourt-Dias M, Goshima G. RNAi in drosophila S2 cells as a tool for studying cell cycle progression. *Methods Mol Biol*. 2009;545:39–62.

Dear Ana,

Thank you for the submission of your revised manuscript to EMBO reports. We have now received the comments from the referees. As you will see, both referees ask for some more clarifications and analyses that I would like you to address and incorporate.

A few editorial requests will also need to be addressed:

- The manuscript has 5 main figures but the results and discussion sections are not combined. Please either combine the sections or add one more main figure, to publish this study as a full article. "Conclusions" needs to be corrected to "Discussion".
- Please add up to 5 keywords to the ms file.
- Please remove the statement "data not shown" on page 8 per journal policy.
- Please complete the section on statistics in the author checklist and send us a new checklist.
- Please add all funding information also to our online ms handling system.
- FIGURE CALLOUTS for Fig. 4F and Tables EV2-EV3 are missing, please add.
- Please upload all source data as one (zipped) folder per figure.
- Please address these comments from our data editors:
 1. Please note that the figure legend style does not comply with the journal guidelines; i.e. the legends are in ""run-on"" style. Each panel in the figure legend needs to start in a new line.
 2. Please note that a separate 'Data Information' section is required in the legends of figures 1; 3; EV1; EV3; EV5.
 3. Please note that in figure EV5, legend for panel c is provided; however, the corresponding figure panel is absent in the figure.
 4. Please indicate the statistical test used for data analysis in the legends of figures 1d-h; 4b-d, g; EV1b-f; EV4b-d
 5. Please note that in figures 2b, 5b, EV3h, EV4b-d, EV5a-b there is a mismatch between the annotated p values in the figure legend and the annotated p values in the figure file that should be corrected.
 6. Please define the annotated p values ** in the legend of figure EV2c
 7. Please note that the box plots need to be defined in terms of minima, maxima, centre, bounds of box and whiskers in the legend of figures 2b; 3d; 4b-d; EV4b-d
 8. Please note that information related to n is missing in the legend of figures 2b; 4b-d, g; EV2c.
 9. Please define what arrows indicate in legend of figure 2a; 5a; EV2a

I would like to suggest some minor changes to the abstract that needs to be written in present tense. Please let me know whether you agree with this:

Centrioles are part of centrosomes and cilia, which are microtubule organising centres (MTOC) with diverse functions. Despite their stability, centrioles can disappear during differentiation, such as in oocytes, but little is known about the regulation of their structural integrity. Our previous research revealed that the pericentriolar material (PCM) that surrounds centrioles and its recruiter, Polo kinase, are downregulated in oogenesis and sufficient for maintaining both centrosome structural integrity and MTOC activity. We now show that the expression of specific components of the centriole cartwheel and wall, including ANA1/CEP295, is essential for maintaining centrosome integrity. We find that Polo kinase requires ANA1 to promote centriole stability in cultured cells and eggs. Additionally, ANA1 expression prevents the loss of centrioles observed upon PCM-downregulation. However, the centrioles maintained by overexpressing and tethering ANA1 are inactive, unlike the MTOCs observed upon tethering Polo kinase. These findings demonstrate that several centriole components are needed to maintain centrosome structure. Our study also highlights that centrioles are more dynamic than previously believed, with their structural stability relying on the continuous expression of multiple components.

EMBO press papers are accompanied online by A) a short (1-2 sentences) summary of the findings and their significance, B) 2-3 bullet points highlighting key results and C) a synopsis image that is exactly 550 pixels wide and 200-600 pixels high (the height is variable). You can either show a model or key data in the synopsis image. Please note that text needs to be readable at the final size. Please send us this information along with the final manuscript.

I look forward to seeing a revised form of your manuscript as soon as possible.

Referee #1:

In the revised manuscript the authors have clarified some of the issues that I raised, but some confusion remains.

Specific points:

1) Ana1 overexpression almost completely rescued centriole numbers after "all PCM" depletion, based on two different markers, suggesting that Ana1 is the critical component. Why then does the double depletion (all PCM + Ana1 RNAi) cause additional reduction in cells with less than normal centriole number? The text argues that Ana1 is important for this function and that the PCM facilitates or reinforces Ana1 binding at centrioles. This would still be the same pathway and therefore it is not obvious why there is an additive effect in the double depletion condition. The authors should offer some explanation or interpretation for this. As currently presented, it is still not clear whether Ana1 is downstream of the PCM or if the PCM has another Ana1-independent role in centriole maintenance. If the crucial role of PCM in centriole maintenance would be to promote Ana1 at centrioles then the double depletion should not have additive effects (assuming efficient depletions in both cases).

2) In some of the experiments quantifying centrioles, instead of the y-axis label "% of cells with 0-1 centrioles" the label should indicate "% of cells with 0-1 (protein name)-positive centrioles" or similar. Currently the plots would suggest different centriole numbers for the same treatment, which is not the case. For a given treatment the centriole numbers are always the same, they are just not positive for all markers.

Related to this, other plots have the y-axis label "% cells 0-1 centrioles identified by respective marker". I assume this is the same as "% of cells with 0-1 centrioles"? The authors should come up with a better, uniform labeling of the plots.

2) Fig. 3A ref on p. 9 seems incorrect.

Referee #2:

The reviewers have made great efforts to address most of the original concerns. Overall, the paper is nearly ready for publication, but I think there still needs to be a few data analyses or text modifications.

1) One key conclusion is that Ana-1 can mediate centriole stability independent of the PCM. This is in part based on the finding that expression of GFP-Polo-PACT can partially rescue centriole loss in cells depleted of "all PCM", but not when Ana-1 is also depleted. The authors state that these rescued centrioles "do not have PCM", but the data showing this is not a quantification of PCM signal. It is instead a measure of the % cells with 0-1 centrioles as defined by staining with various PCM markers (Fig EV3B-E). There are 2 problems here: 1) the % of cells scored as having 0-1 centrioles when using PCM markers is not 100%, as it technically should be if the centrioles completely lacked PCM. Asl is the clearest example, with ~20% of cells containing >1 observable centrosome (which must therefore have Asl present); 2) presumably some of the cells scored in this analysis contained 1 centriole, as the category is "0-1 centriole" i.e. these centrioles will have had the PCM marker. Shouldn't the y-axis therefore be "% of cells with 0 centrioles"? Otherwise, the reader has to assume that sometimes the authors could see rescued centrioles with the PCM markers, muddying the conclusion that the PCM was absent. I assume the authors have an excel sheet with the scores for each condition, they could simply plot the data for % cells with 0 centrioles. A better way would be to measure the average PCM signal at centrioles in each condition, but I appreciate that this would be a lot more work. All in all, I suspect that when quantified correctly, the authors will be able to conclude that there are low levels of PCM at these centrioles, but not that the PCM is completely missing. Some representative images would be nice too. Note that the same comment applies to Spd-2 recruitment in Figure EV5B.

2) The authors test whether Ana-1 is required for Polo-induced maintenance of centrioles in St12 oocytes. They show a reduction from ~90% to ~60% of eggs containing centrioles when Ana1 is depleted in the background of expressing GFP-Polo-PACT (Fig EV2C). As stated later in the paper, only ~30% of control eggs contain centrioles at this stage, so the reduction from

90 to 60% is not complete. Nevertheless, the authors conclude that "Polo-induced centrosome maintenance in vivo is dependent on the presence of ANA1". I think this should be re-phrased to "at least partially dependent", as the data suggest that other mechanisms could be involved.

3) When the authors tether Ana-1 to centrioles and show that this prevents centriole loss in oocytes they then state that "Given these centrioles are naked (no PCM), our data also shows that ANA1 is critical for maintaining centriole integrity, independently of its known role in PCM recruitment during centriole-to-centrosome conversion in cycling cells" but I'm not sure they can say that the centrioles are naked. In Figure 4A they 4C they show that γ -tubulin levels are lower than when tethering Polo to the centriole and similar to controls, but there is still some γ -tubulin present (i.e. controls still have a γ -tubulin signal). It's the same for Cnn in Figure EV4AC. It's likely that these low signals of γ -tubulin and Cnn reflect a fraction of the proteins that associate tightly with the PCM that is different from PCM recruited during mitosis, but have the authors ruled out that these tightly associated PCM fractions don't contribute to Ana-1's ability to maintain the centrioles? This is especially relevant given there appears to be some PCM signal left in cultured cells after PCM depletion (see point above). It is clear that the tethering of Ana-1 does not lead to the recruitment of additional PCM, so the authors can claim that Ana-1 functions in centriole stability independent of recruiting any additional PCM to the centrioles. Perhaps it should be re-phased like this.

4) In terms of the conclusion that the PCM may exert its function in centriole stability by recruiting and maintaining Ana-1, I still think the authors need to reword the last paragraph of the results to help the reader. The result is that over-expressing Ana-1 can partially rescue the "all PCM" RNAi phenotype. For this to be true, and for the result to match the conclusion, there MUST be some PCM left for the over-expressed Ana-1 to be stabilised by, unless the authors are suggesting that the addition of more Ana-1 into the cytoplasmic pool somehow overrides the need for the PCM to recruit/stabilise Ana-1. Can the authors explain more clearly their idea in the last paragraph?

Manuscript number: #RC-2022-01412

Corresponding author(s): Ana, Pimenta-Marques; Mónica, Bettencourt-Dias

Dear Esther,

We would like to thank the referees for their comments which have improved our manuscript. We have changed our manuscript accordingly and provide here a point-by-point answer to their questions. Furthermore, we have incorporated all modifications and supplemented the manuscript with the additional information requested by the data editors.

Referee 1:

In the revised manuscript the authors have clarified some of the issues that I raised, but some confusion remains.

Specific points:

1) Ana1 overexpression almost completely rescued centriole numbers after "all PCM" depletion, based on two different markers, suggesting that Ana1 is the critical component. Why then does the double depletion (all PCM + Ana1 RNAi) cause additional reduction in cells with less than normal centriole number? The text argues that Ana1 is important for this function and that the PCM facilitates or reinforces Ana1 binding at centrioles. This would still be the same pathway and therefore it is not obvious why there is an additive effect in the double depletion condition. The authors should offer some explanation or interpretation for this. As currently presented, it is still not clear whether Ana1 is downstream of the PCM or if the PCM has another Ana1-independent role in centriole maintenance. If the crucial role of PCM in centriole maintenance would be to promote Ana1 at centrioles then the double depletion should not have additive effects (assuming efficient depletions in both cases).

We thank the reviewer for this comment and have changed the text to accommodate that concern (see page 9 in green). While ANA1 is a critical component in stability, we have identified other components in our screen that also contribute for stability (ANA2 and SAS6; Fig.1 and EV1). Therefore, one possibility is that the PCM not only facilitates or reinforces ANA1 binding to centrioles but enforces the presence of the other centriole components. This could explain why the combined effect of depleting the PCM (ALL PCM) and ANA1 leads to an increase in loss of centrioles in comparison with individual treatments. The fact that overexpression of ANA1 on its own is sufficient to rescue centriole loss after PCM depletion (Fig 6), suggests that higher levels of ANA1 override the requirements for other centriolar components, which does not occur in control conditions.

2) In some of the experiments quantifying centrioles, instead of the y-axis label "% of cells with 0-1 centrioles" the label should indicate "% of cells with 0-1 (protein name)-positive centrioles" or similar. Currently the plots would suggest different centriole numbers for the

same treatment, which is not the case. For a given treatment the centriole numbers are always the same, they are just not positive for all markers.

Related to this, other plots have the y-axis label "% cells 0-1 centrioles identified by respective marker". I assume this is the same as "% of cells with 0-1 centrioles"? The authors should come up with a better, uniform labeling of the plots.

We thank the reviewer for this comment, our plots do indeed refer to the number of cells which are positive for the different markers analyzed. We have changed the y axis label and use an uniform labeling as: % cells with 0-1 (respective protein name) positive centrioles.

2) Fig. 3A ref on p. 9 seems incorrect.

We thank the reviewer for that comment. We have corrected the reference.

Referee 2:

The authors have made great efforts to address most of the original concerns. Overall, the paper is nearly ready for publication, but I think there still needs to be a few data analyses or text modifications.

1) One key conclusion is that Ana-1 can mediate centriole stability independent of the PCM. This is in part based on the finding that expression of GFP-Polo-PACT can partially rescue centriole loss in cells depleted of "all PCM", but not when Ana-1 is also depleted. The authors state that these rescued centrioles "do not have PCM", but the data showing this is not a quantification of PCM signal. It is instead a measure of the % cells with 0-1 centrioles as defined by staining with various PCM markers (Fig EV3B-E). There are 2 problems here: 1) the % of cells scored as having 0-1 centrioles when using PCM markers is not 100%, as it technically should be if the centrioles completely lacked PCM. Asl is the clearest example, with ~20% of cells containing >1 observable centrosome (which must therefore have Asl present); 2) presumably some of the cells scored in this analysis contained 1 centriole, as the category is "0-1 centriole" i.e. these centrioles will have had the PCM marker. Shouldn't the y-axis therefore be "% of cells with 0 centrioles"? Otherwise, the reader has to assume that sometimes the authors could see rescued centrioles with the PCM markers, muddying the conclusion that the PCM was absent. I assume the authors have an excel sheet with the scores for each condition, they could simply plot the data for % cells with 0 centrioles. A better way would be to measure the average PCM signal at centrioles in each condition, but I appreciate that this would be a lot more work. All in all, I suspect that when quantified correctly, the authors will be able to conclude that there are low levels of PCM at these centrioles, but not that the PCM is completely missing. Some representative images would be nice too. Note that the same comment applies to Spd-2 recruitment in Figure EV5B.

We thank the reviewer for this comment and understand that we were not explaining the data well enough. Given the doubt is about presence/absence of PCM in the rescued

centrioles, we have added a new figure EV3 (graphs and images) and new graphs and images in Fig EV5, where we analyze the percentage of cells with normal number of centrioles (more than 1) that have (or not) a PCM marker. In these graphs it is clear that there is no increase in the proportion of centrioles that have a PCM marker, upon expression of GFP-Polo-PACT (Fig. EV3) or ANA1 (Fig. EV5). We stress that result also in the text in pages 6 and 9. We think this new data representation makes the results easier to interpret.

Additionally, in the legend of Fig EV2 we state that “Note that upon “All PCM” RNAi there is a large increase in the percentage of cells with abnormally low PCM foci per cell (zero or one) in the case of D-PLP, Asl or SPD2, which is not reduced upon GFP-Polo-PACT expression”.

2) The authors test whether Ana-1 is required for Polo-induced maintenance of centrioles in St12 oocytes. They show a reduction from ~90% to ~60% of eggs containing centrioles when Ana1 is depleted in the background of expressing GFP-Polo-PACT (Fig EV2C). As stated later in the paper, only ~30% of control eggs contain centrioles at this stage, so the reduction from 90 to 60% is not complete. Nevertheless, the authors conclude that "Polo-induced centrosome maintenance in vivo is dependent on the presence of ANA1". I think this should be re-phrased to "at least partially dependent", as the data suggest that other mechanisms could be involved.

We thank the reviewer for this comment.

In this experiment we have induced the ectopic maintenance of centrioles by using the expression of GFP-Polo-PACT to maintain centrioles in stages where they normally are starting to disappear. This allowed us to deplete ANA1 in this context and ask if the maintenance of centrioles is dependent on Polo.

The ideal experimental setup would be using the DeGradFP to degrade ANA1, as was done in Fig.2 of our manuscript. Unfortunately, we could not use such an approach as the DegradFP would also degrade the GFP-POLO-PACT which is also being expressed in this experiment. We could also not use ANA1 mutants or germ-line clones, as centriole duplication would be affected at the germline, not allowing us to address centriole maintenance.

As an alternative we induced the depletion of ANA1 by using a RNAi strategy. The RNAi however does not completely deplete ANA1. This is consistent with previous reports showing that ANA1 at centrosomes has a dynamic and a stable pool (Saurya et al., 2016). It is possible that the RNAi only affects the dynamic pool of ANA1 upon RNAi in the germline. We had previously tested the efficiency of the RNAi by using a line expressing ANA1-Tomato (under ANA1 endogenous promoter) and using the RNAi line for ANA1. We found that in stages 10 of oogenesis (where centrioles are still normally present), although the signal was reduced, we could always identify ANA1-tdTomato in these stages, showing that the depletion of ANA1, is not 100% efficient using this strategy (see new Fig. EV4 A, B). This partial RNAi effect is likely to explain why there is a reduction to 60% of oocytes showing centrioles and not 30%, as pointed out by the reviewer. We now discuss this result in the manuscript in page 7 and have changed the text to accommodate the reviewer's suggestion to rephrase the text.

3) When the authors tether Ana-1 to centrioles and show that this prevents centriole loss in oocytes they then state that "Given these centrioles are naked (no PCM), our data also shows that ANA1 is critical for maintaining centriole integrity, independently of its known role in PCM recruitment during centriole-to-centrosome conversion in cycling cells" but I'm not sure they can say that the centrioles are naked. In Figure 4A they show that γ -tubulin levels are lower than when tethering Polo to the centriole and similar to controls, but there is still some γ -tubulin present (i.e. controls still have a γ -tubulin signal). It's the same for Cnn in Figure EV4AC. It's likely that these low signals of γ -tubulin and Cnn reflect a fraction of the proteins that associate tightly with the PCM that is different from PCM recruited during mitosis, but have the authors ruled out that these tightly associated PCM fractions don't contribute to Ana-1's ability to maintain the centrioles? This is especially relevant given there appears to be some PCM signal left in cultured cells after PCM depletion (see point above). It is clear that the tethering of Ana-1 does not lead to the recruitment of additional PCM, so the authors can claim that Ana-1 functions in centriole stability independent of recruiting any additional PCM to the centrioles. Perhaps it should be re-phased like this.

As suggested by the reviewer we have rephrased the text (page 8 , written in green).

4) In terms of the conclusion that the PCM may exert its function in centriole stability by recruiting and maintaining Ana-1, I still think the authors need to reword the last paragraph of the results to help the reader. The result is that over-expressing Ana-1 can partially rescue the "all PCM" RNAi phenotype. For this to be true, and for the result to match the conclusion, there MUST be some PCM left for the over-expressed Ana-1 to be stabilised by, unless the authors are suggesting that the addition of more Ana-1 into the cytoplasmic pool somehow overrides the need for the PCM to recruit/stabilise Ana-1. Can the authors explain more clearly their idea in the last paragraph?

We thank the reviewer for this comment and have changed the text also in reply to reviewer 1. In page 9 we now state that expressing Ana-1 rescues centriole loss in the "all PCM" RNAi condition. Overexpression of Ana-1 is thus likely to override the need for the PCM to recruit/stabilize ANA-1 (as suggested by this reviewer) and potentially also other components of the centriole, which are important for maintenance (as suggested by reviewer 1).

REFERENCES

Saurya, S., Roque, H., Novak, Z.A., Wainman, A., Aydogan, M.G., Volanakis, A., Sieber, B., Pinto, D.M.S. and Raff, J.W. (2016), "Drosophila Ana1 is required for centrosome assembly and centriole elongation", *Journal of Cell Science*, Vol. 129 No. 13, pp. 2514 LP – 2525

Ana Pimenta-Marques
NOVA Medical School Research Center
Faculdade de Ciências Médicas, Universidade Nova de Lisboa
Rua Câmara Pestana nº 6, 6-A
Edifício CEDOC II
Lisbon, Lisboa 1150-082 Lisboa
Portugal

Dear Ana,

I am very pleased to accept your manuscript for publication in the next available issue of EMBO reports. Thank you for your contribution to our journal.

Best wishes,
Esther
